# In vivo single-cell CRISPR uncovers distinct TNF programmes in tumour evolution

Peter F. Renz[1,6], Umesh Ghoshdastider[1,6], Simona Baghai Sain[2,6], Fabiola Valdivia-Francia[1,3], Ameya Khandekar[4,5], Mark Ormiston[1], Martino Bernasconi[1], Clara Duré[1,3], Jonas A. Kretz[2], Minkyoung Lee[2], Katie Hyams[1], Merima Forny[1], Marcel Pohly[2], Xenia Ficht[2], Stephanie J. Ellis[4,5], Andreas E. Moor[2✉] & Ataman Sendoel[1✉]

The tumour evolution model posits that malignant transformation is preceded by randomly distributed driver mutations in cancer genes, which cause clonal expansions in phenotypically normal tissues. Although clonal expansions can remodel entire tissues[1–3], the mechanisms that result in only a small number of clones transforming into malignant tumours remain unknown. Here we develop an in vivo single-cell CRISPR strategy to systematically investigate tissue-wide clonal dynamics of the 150 most frequently mutated squamous cell carcinoma genes. We couple ultrasound-guided in utero lentiviral microinjections, single-cell RNA sequencing and guide capture to longitudinally monitor clonal expansions and document their underlying gene programmes at single-cell transcriptomic resolution. We uncover a tumour necrosis factor (TNF) signalling module, which is dependent on TNF receptor 1 and involving macrophages, that acts as a generalizable driver of clonal expansions in epithelial tissues. Conversely, during tumorigenesis, the TNF signalling module is downregulated. Instead, we identify a subpopulation of invasive cancer cells that switch to an autocrine TNF gene programme associated with epithelial–mesenchymal transition. Finally, we provide in vivo evidence that the autocrine TNF gene programme is sufficient to mediate invasive properties and show that the TNF signature correlates with shorter overall survival of patients with squamous cell carcinoma. Collectively, our study demonstrates the power of applying in vivo single-cell CRISPR screening to mammalian tissues, unveils distinct TNF programmes in tumour evolution and highlights the importance of understanding the relationship between clonal expansions in epithelia and tumorigenesis.

Tumour evolution begins with the transformation of single or multiple cells that expand to form a tumour mass. Malignant transformation is preceded by randomly distributed driver mutations in cancer genes, which cause clonal expansions in phenotypically normal tissues. These clonal expansions increase the pool of cells that can accumulate additional driver mutations until a sufficient set of mutations and gene-expression changes are acquired to enable malignant transformation. Notably, driver mutations in cancer genes occur with a surprisingly high frequency in normal human epithelia such as the skin, oesophagus or colon[1–3]. In addition, rather than only being viewed as the underlying basis for transformation, highly competitive driver mutation clones can also have a tumour-suppressive role by eliminating new emerging tumours through clonal competition[4]. These observations raise the question of the fundamental relationship between clonal expansions in normal epithelia and their role in driving malignant transformation.

Although pooled CRISPR screens are powerful tools for studying clonal growth in vivo, they are limited to simple readouts such as proliferation, averaged over all tissue cell types. The monitoring of genetically perturbed clones in normal epithelia and tumorigenesis requires scalable, tissue-wide platforms that can profile the transcriptomic consequences of gene perturbations across all cell types. Here, we combine ultrasound-guided in utero microinjections into embryonic stage 9.5 (E9.5) mouse embryos with a single-cell CRISPR strategy to systematically document clonal expansions at single-cell transcriptomic resolution and outline the molecular processes that distinguish them from tumorigenesis.

## In vivo single-cell CRISPR screening

To monitor tissue-wide clonal expansion using an in vivo single-cell CRISPR strategy, we adjusted the CRISPR droplet sequencing (CROP-seq) system[5,6] by substituting the puromycin cassette with mCherry to enable selection of infected cells by fluorescence-activated cell sorting (FACS) (Fig. 1a). We selected the 150 most frequently altered

[1]Institute for Regenerative Medicine (IREM), University of Zurich, Schlieren-Zurich, Switzerland. [2]Department of Biosystems Science and Engineering, ETH Zurich, Basel, Switzerland. [3]Life Science Zurich Graduate School, Molecular Life Science Program, University of Zurich and ETH Zurich, Zurich, Switzerland. [4]Max Perutz Labs, Vienna BioCenter Campus (VBC), Vienna, Austria. [5]Center for Molecular Biology, Department of Microbiology, Immunobiology and Genetics, University of Vienna, Vienna, Austria. [6]These authors contributed equally: Peter F. Renz, Umesh Ghoshdastider, Simona Baghai Sain. ✉e-mail: andreas.moor@bsse.ethz.ch; ataman.sendoel@uzh.ch

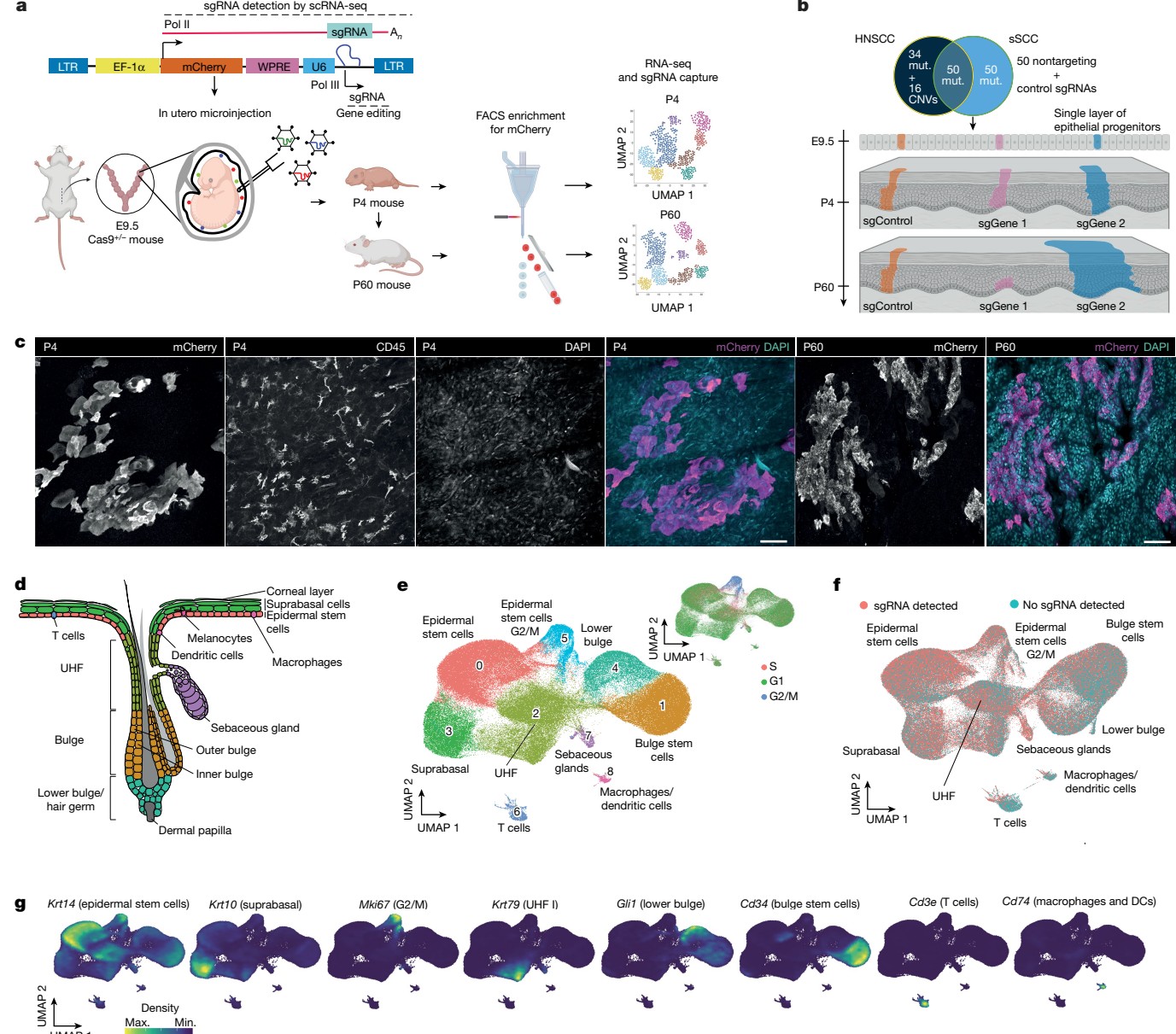

**Fig. 1 | In vivo CRISPR screening to monitor clonal expansion in the mouse skin. a**, Schematic of the in vivo CROP-seq strategy to couple pooled CRISPR screening with single-cell transcriptomic readout. LTR, long terminal repeat; U6, U6 RNA polymerase III promoter; UMAP, uniform manifold approximation and projection; WPRE, woodchuck hepatitis virus post-transcriptional regulatory element. **b**, The sgRNA library targets the 150 most frequently mutated genes (3 sgRNAs per gene and 50 control sgRNAs) in patients with HNSCC or sSCC. sgControl, sgGene1 and sgGene2 refer to non-targeting control sgRNAs or sgRNAs targeting gene 1 and gene 2, respectively. Mut., mutations. **c**, Whole-mount immunofluorescence staining of mCherry-positive clonal expansions in P4 and P60 mouse skin. Immunofluorescence images provide a planar view across the epidermis. CD45 staining marks haematopoietic cells. Scale bars, 50 μm. **d**, Schematic representation of the anatomical localization of the different cell populations in the P60 mouse skin. UHF, upper hair follicle. **e**, UMAP of the 9 cell populations identified by the in vivo single-cell CRISPR strategy for cells in skin from P60 mice ($n = 183,084$ cells from 10 mice, following filtering and sgRNA annotation). Top right, cell cycle phase superimposed on the UMAP. **f**, sgRNAs are uniformly detected in all major cell populations. The UMAP shows total mCherry-sorted P60 cells. **g**, Visualization of cell-type-specific marker gene expression by Nebulosa density plots. DC, dendritic cell (Langerhans cell); max., maximum; min., minimum.

genes in human head and neck squamous cell carcinoma (HNSCC) and skin squamous cell carcinoma (sSCC), comprising 134 mutated genes and 16 copy number variations (CNVs) (15 of which are amplifications) (Fig. 1a,b and Extended Data Fig. 1a–c). Leveraging the ultrasound-guided in utero microinjection system[7,8], we delivered the lentiviral library into the amniotic cavity of E9.5 mouse embryos to infect the single-layered surface ectoderm of Cas9-expressing embryos and introduce loss-of-function mutations in each of the selected cancer genes[9]. The library carrying the 500 single guide RNAs (sgRNAs)

(3 sgRNAs per gene) is stably propagated by the surface ectoderm, which gives rise to the epidermis and different skin appendages[7,8,10]. At postnatal day 4 (P4) and 60 (P60), we collected epidermal tissues, sorted for mCherry-positive infected cells, and analysed them by single-cell RNA sequencing (scRNA-seq) to assess gene-expression profiles and capture sgRNA identity (Fig. 1a). We generated scRNA-seq libraries comprising a total of 477,773 single cells. Following stringent filtering and sgRNA annotation, we obtained 120,077 P4 and 183,084 P60 single cells, providing an average coverage of 240 P4 cells and 366 P60 cells per

guide. To reduce double sgRNA infections[7], we aimed at infection rates between 1% and 12.5%, resulting in multiple infection rates under 1% (Extended Data Fig. 1d). Whole-mount immunofluorescence imaging of the P4 and P60 skin confirmed the clearly separated mCherry-positive clusters within the epidermis (Fig. 1c and Supplementary Fig. 1).

To validate our approach, we first confirmed the efficiency of selected sgRNAs and found a 67–86% editing efficiency in keratinocytes (Extended Data Fig. 1e). Second, the control library of 50 sgRNAs was overall stably propagated across the time points (Extended Data Fig. 1f). Third, to validate the accuracy of our sgRNA assignments and single-cell transcriptomic data, we examined *Notch1* sgRNA cells, which exhibited a significant downregulation of *Notch1* target genes such as *Hes1*, *Fzd1* and *Tcf7l2* (Extended Data Fig. 1g). Fourth, we assessed the reproducibility of our results and found that the enrichment and depletion of the top and bottom 20 sgRNAs correlated well over 8 replicates in P60 mice (Extended Data Fig. 1h,i and Supplementary Fig. 2).

Our strategy enabled the identification of 9–10 distinct sgRNA-containing cell types in the P4 and P60 mouse skin, including epidermal stem cells (EpSCs), suprabasal cells, hair follicle cells, sebaceous glands, T cells and macrophages (Fig. 1d,e and Extended Data Fig. 2a–d). The classic marker genes, such as *Krt14* for EpSCs, *Krt10* for suprabasal stem cells and *Cd34* for bulge stem cells, showed specific expression in the corresponding clusters[11] (Fig. 1g and Extended Data Fig. 2e,f). Moreover, sgRNAs were uniformly detected throughout these clusters (Fig. 1f). Together, the in vivo single-cell CRISPR strategy that we established provides a powerful tool to study tissue-wide gene function in the epidermis by coupling single-cell transcriptomics with sgRNA capture.

## Monitoring clonal expansion

Previous ultradeep sequencing efforts in phenotypically normal human skin revealed somatic mutations in cancer driver genes, with the highest burden of *NOTCH1* and *NOTCH2* family, *FAT1* and *TP53* mutations[3]. To determine clonal expansions in the mouse epidermis, we computed the total sgRNA enrichment compared to the initial library (T0) across the P4 and P60 time points. Consistent with findings in normal human skin, the P60 epidermis exhibited a strong enrichment of guides targeting *Fat1*, *Notch1*, *Trp53* and *Notch2*, alongside less-characterized genes such as those encoding the ryanodine receptor (*Ryr3*) and the actin-binding protein (*Xirp2*) (Fig. 2a, Extended Data Fig. 3a and Supplementary Tables 1 and 2). At P4 stage, following embryonic stratification and differentiation, *Fat1*, *Notch1* and *Trp53* mutant clones were already expanded (Extended Data Fig. 3b). Between the P4 and P60 time points, whereas *Notch2*, *Notch1* and *Trp53* continued to demonstrate clonal expansion, myosin heavy chain (*Myh2*) and cadherin (*Cdh10*) emerged as the top enriched mutant genes (Extended Data Fig. 3c). By contrast, sgRNAs targeting the 15 amplified CNVs in cancer, such as *Trp63* (which encodes an essential transcription factor for epidermal development and maintenance[12]), were among the most strongly depleted guides (Fig. 2a).

The interfollicular epidermis is maintained by EpSCs that divide stochastically, with balanced fate outcomes between self-renewing daughter EpSCs and differentiating cells[13–15] (Fig. 1d and Extended Data Fig. 3d). For clonal expansion to occur, EpSCs must either proliferate faster than their neighbours or shift the balance towards self-renewal[13,14,16]. To examine these two scenarios, we computed an anti-differentiation and a proliferation index (Fig. 2b–e). *Notch1* sgRNA cells showed, while remaining low in proliferation, the most potent differentiation block, which aligns with the known role of NOTCH1 in inducing suprabasal cell fate[17,18] (Fig. 2c). Thus, *Notch1*-perturbed clonal expansion is driven primarily by the differentiation block, an aspect that reflects the differentiation imbalance, as demonstrated in elegant studies in the oesophagus[16]. This observation was further supported by assessing the total percentages of EpSCs, in which *Notch1* sgRNA cells showed the highest EpSC percentage (Fig. 2b). We also analysed other cell types, such as T cells, and found

sgRNAs targeting *Kmt2d*, *Nsd1* and *Reln* to be the most strongly enriched (Fig. 2d). *KMT2D* is recurrently mutated in Sézary syndrome[19], whereas *RELN* is recurrently mutated in early T cell precursor acute lymphoblastic leukaemia[20], indicating that our in vivo CRISPR strategy may also uncover genes involved in immune cell-related disorders. Together, these findings demonstrate that our single-cell CRISPR screening strategy enables cell-type-specific monitoring of clonal expansion.

## Clonal expansion converges on TNF

Unlike traditional single-mutation characterization, our systems-level strategy enables us to study gene programmes shared by multiple cancer gene mutations. We first individually computed differential gene expression induced by each of the 150 perturbations in EpSCs. Notably, many of the top expansion-driving perturbations exhibited gene-expression changes that were highly enriched for tumour necrosis factor (TNF) signalling (Extended Data Fig. 4a and Supplementary Table 3). To systematically examine shared signalling pathways, we used weighted gene correlation network analysis (WGCNA) in the P60 clusters, a powerful tool for uncovering gene regulatory networks[21–24] (Extended Data Fig. 4b). We identified a total of 44 WGCNA modules across P60 cell populations, which were dominated by 1 major gene module present in 6 clusters (Fig. 2f, Extended Data Fig. 4c and Supplementary Table 4). This module comprised 25 genes in EpSCs (Fig. 2g) and showed a marked enrichment for TNF signalling (adjusted *P* value ($P_{adj}$) = $1.21 \times 10^{-27}$; Extended Data Fig. 4d). Besides this major gene module, we found many modules that reflected subclusters present within cell populations, such as a *Dct* and *Sox10* module, representing melanocytes, and an Lce1 module, reflecting the corneal layer within the suprabasal compartment (Extended Data Fig. 4c).

To explore whether TNF signalling correlates with expansion rates, we first clustered perturbations according to TNF gene module expression (Fig. 2g). We observed that clusters 1 and 2, which were characterized by high module gene expression, displayed higher expansion rates than cluster 3 (Fig. 2h). Second, we fitted a linear model[25] to estimate how the 25 module genes predict clonal expansions and found a Pearson correlation of 0.68 ($P = 1.4 \times 10^{-9}$), indicating that the TNF signalling module explains a substantial portion of the variance in clonal expansion rates (Extended Data Fig. 4e–g and Supplementary Figs. 3 and 4). Third, we expressed the TNF module genes *Ccn1*, *Jun* and *Fos* in keratinocytes and observed that *Jun* or *Fos* expression was sufficient to enhance their proliferative rate (Extended Data Fig. 4h).

To corroborate this convergence on TNF signalling, we first compared the gene expression of the top 5, 10 or 20 expanded perturbations to the corresponding average or bottom perturbed groups. In each of these comparisons, TNF signalling was the most enriched pathway (Fig. 2i and Extended Data Fig. 4i). By contrast, genes involved in epithelial–mesenchymal transition (EMT) were strongly enriched among downregulated genes. In a second approach, we performed an unbiased P60 clustering of transcriptomically similar perturbations and a minimum-distortion embedding to place perturbations with correlated EpSC transcriptomes near each other[23] (Extended Data Fig. 5a,b). *Fat1*, *Notch1* and *Notch2* clustered together along with other enriched perturbations such as *Myh1* and *Fgf3*, providing additional evidence that enriched perturbations lead to similarly altered gene-expression pathways and activate TNF signalling (Extended Data Fig. 5a). Collectively, despite the wide range of biological processes encompassed by our cancer gene cohort, our findings suggest that clonal expansion in phenotypically normal epithelia converges on a shared pathway that induces a TNF signalling module.

## TNF receptor 1 and macrophage dependency

TNF signalling is mainly activated by binding of the TNF ligand to the TNF receptor 1 (TNFR1), which induces several signal transduction

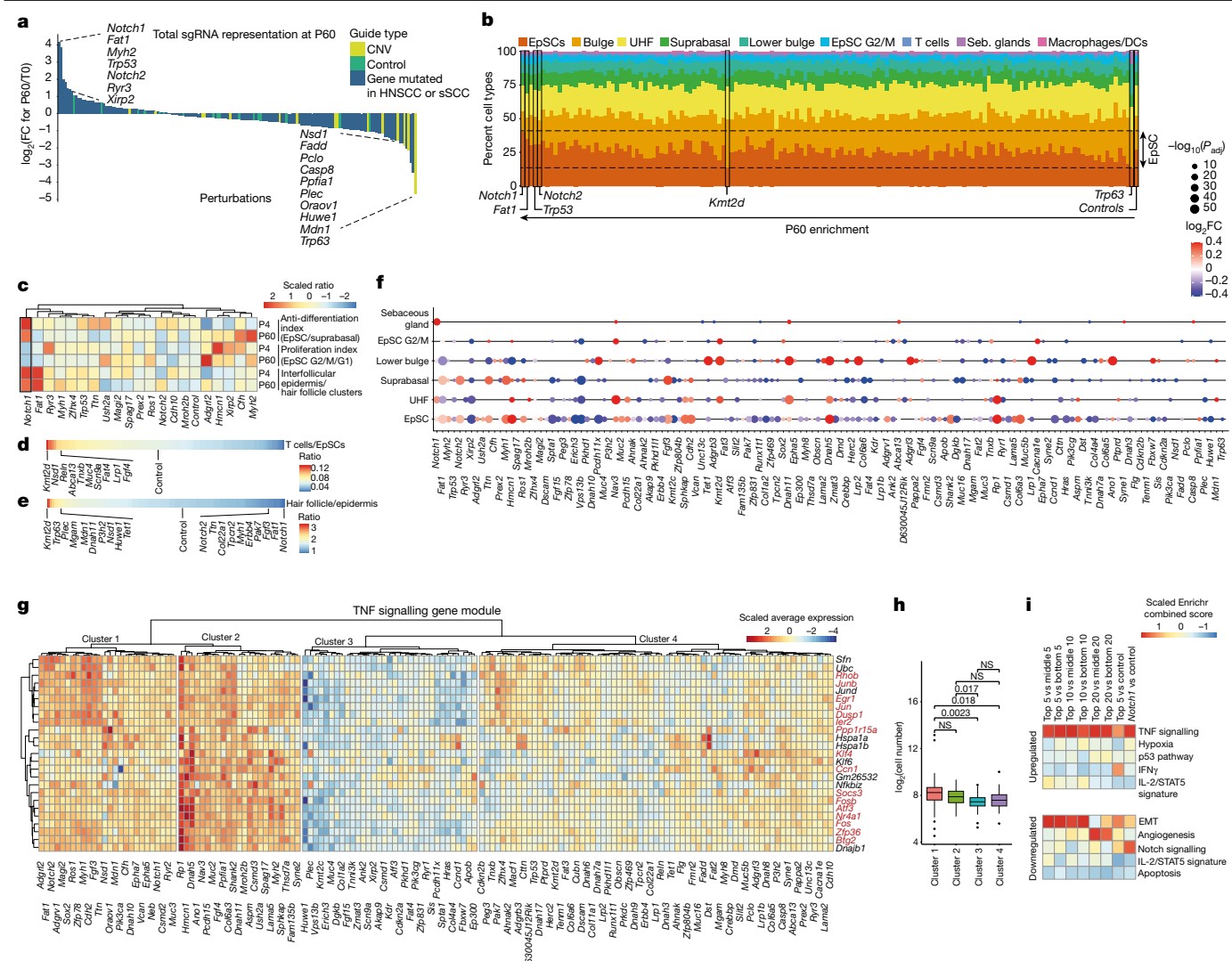

**Fig. 2 | Clonal expansion in epithelia converges on TNF signalling. a**, *Notch1*, *Fat1* and *Trp53* sgRNAs are the top enriched perturbations in P60 epidermis. Waterfall plot displaying the $\log_2$ fold change (FC) in total P60 cell numbers compared to the library T0 representation. Data represent the weighted average of eight P60 replicates from ten mice. **b**, Cell-type distribution of the 150 P60 skin perturbations, sorted by P60 enrichment. Dashed lines indicate minimum and maximum EpSC percentages. Seb. sebaceous. **c**, Heat map of the top 20 enriched perturbations in P60 skin, displaying anti-differentiation, proliferation and interfollicular epidermis indices. **d**,**e**, Linear heat maps illustrating the cell-type specificity of the perturbations as the ratios of T cells to EpSCs (**d**) and hair follicle cells/epidermis cells (**e**). **f**, Average perturbation

effect on the TNF signalling gene module. WGCNA identified a major gene module in six different P60 clusters. Dot plots show the average $\log_2$ fold change of module genes. Adjusted *P* values with Benjamini–Hochberg correction. **g**, Heat map displaying the average expression of 25 genes in the TNF EpSC module across 150 perturbations. Red text highlights 17 TNF signalling genes (MsigDB Hallmark). **h**, Clusters 1 and 2, which exhibit increased TNF module expression (**g**), demonstrate higher expansion rates than cluster 3. *P* values by Wilcoxon rank-sum test. **i**, TNF signalling is the top enriched pathway in enriched P60 perturbations. Comparisons of top 5, 10 and 20 versus middle or bottom 5, 10 and 20 gene perturbations by transcriptomic GSEA (Enrichr, MSigDB Hallmark). NS, not significant.

arms, including JUN and NF-κB[26,27]. These transcription factors are responsible for diverse biological processes, including cell growth, immune and stress responses. *Tnfrsf1a* (also known as *Tnfr1*), *Jnk* (also known as *Mapk8*) and *NF-κB* (also known as *Nfkb1*) act as proliferative signals for controlling epidermal growth[28]. TNF further serves as a proinflammatory signal in clonal haematopoiesis[29,30], and *Tnfrsf1a*- or *Tnf*-knockout mice are resistant to skin carcinogenesis[31,32]. In P60 skin, the receptor *Tnfrsf1a* is mainly expressed in EpSCs and bulge stem cells (Fig. 3a). *Tnfrsf1b* (also known as *Tnfr2*) is restricted to T cells, whereas TNF is primarily secreted by macrophages (Fig. 3a). Examining the ligand–receptor interactions[33] supported TNF signalling from macrophages and T cells to EpSCs (Fig. 3a).

To experimentally test whether the TNF pathway directly mediates clonal expansion in phenotypically normal epithelia, we crossed

*Cas9* mice with *Tnfrsf1a*$^{+/-}$ mice and used ultrasound-guided in utero microinjection to target the 150 cancer genes in E9.5 embryos. We then tested which cancer gene perturbations showed TNFR1-dependent clonal expansion by comparing the expansion rates in wild-type versus *Tnfrsf1a*$^{+/-}$ P60 skin (Fig. 3b,c). Among the top 20 enriched sgRNAs at P60, all but 3 perturbations were highly dependent on TNFR1 for clonal expansion, indicating that TNF signalling has a widespread role in mediating clonal expansions in normal epithelia (Fig. 3c, Supplementary Figs. 5 and 6 and Supplementary Table 5). As expected, the dependency on TNFR1 decreased as the perturbations became less enriched (Fig. 3c, red weighted regression curve).

Immunofluorescence analyses of epidermal sections confirmed TNFR1 expression in E11.5 embryos and EpSCs and highlighted immune cells as a potential source of TNF in the epidermis (Fig. 3d,e

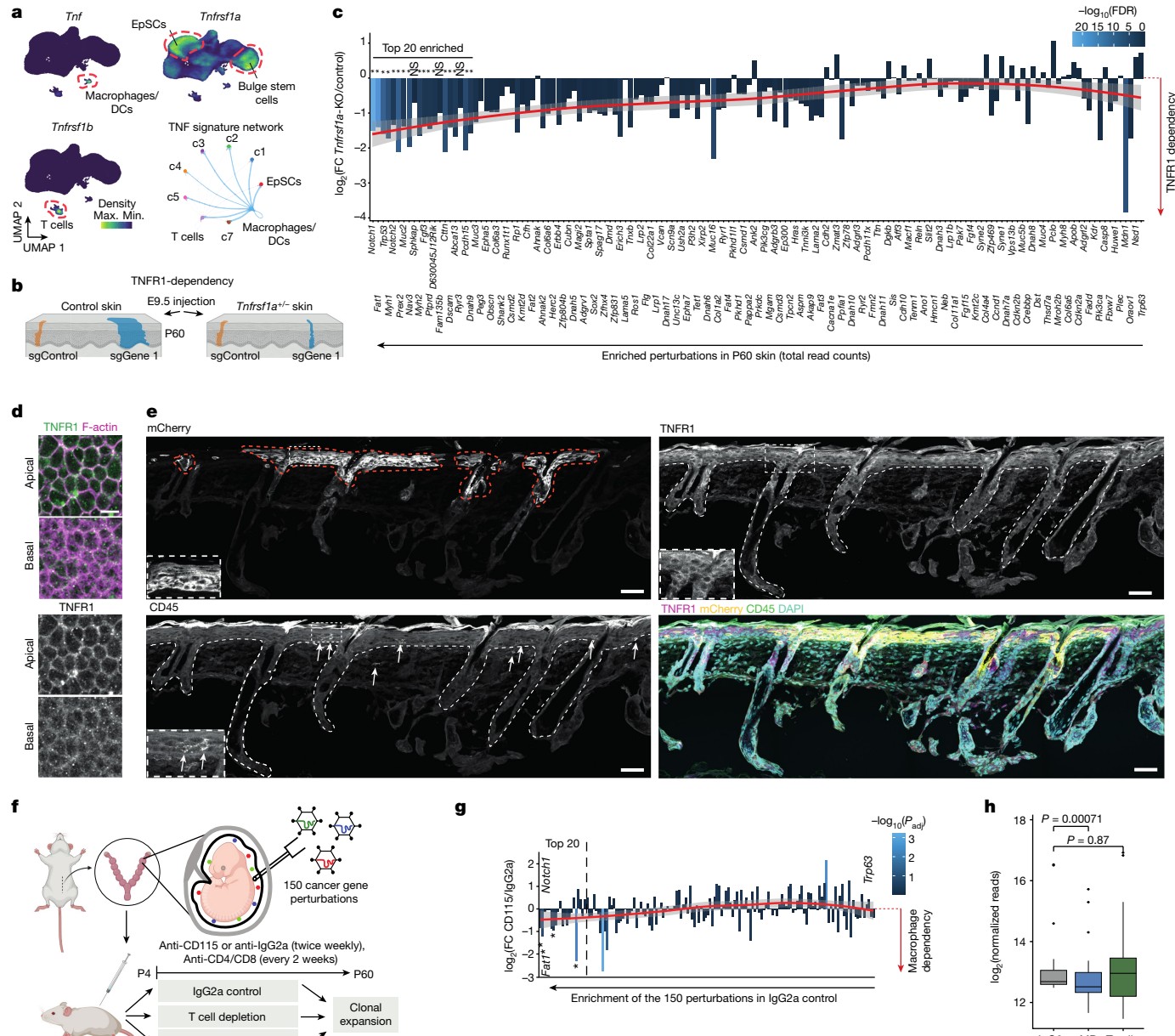

**Fig. 3 | Clonal expansion is mediated via TNF signalling. a**, Nebulosa plots showing expression in P60 skin. Bottom right, CellChat ligand–receptor analysis of P60 skin. **b**, Schematic of the experimental setup to test the role of TNFR1 in clonal expansions. **c**, Clonal expansion is dependent on TNFR1. The 500 sgRNA library was microinjected into control and *Tnfrsf1a*-knockout (*Tnfrsf1a*$^{+/-}$) E9.5 embryos and sgRNA representation was quantified in P60 mice. Using MAGeCK, perturbations were normalized to the 50 control sgRNAs and TNFR1-dependency was calculated by the ratio of *Tnfrsf1a*-knockout (KO) and control embryos. The red line is a locally weighted regression curve and the grey area shows the 95% confidence interval. *False discovery rate (FDR) < 0.05. Data represent the average of 8 control (10 mice) and 6 *Tnfrsf1a*$^{+/-}$ (3 mice) replicates (Methods). **d**, Whole-mount immunofluorescence across the basal and apical E11.5 epidermis shows cell membrane expression of TNFR1. Scale bar, 10 µm. **e**, Immunofluorescence of P60 back skin shows mCherry-positive expanded clones and TNFR1 expression in EpSCs alongside CD45-positive immune cells. Scale bars, 50 µm. **f**, Schematic of the immune-depletion experiment.

**g**, Macrophage depletion reduces clonal expansion of the top enriched cancer gene perturbations. sgRNA representation was assessed at P60, normalized to 50 control sgRNAs and compared to IgG2a control samples using DESeq2. The red line is a locally weighted regression curve. *P < 0.05 within top 20 perturbations. The grey area shows the 95% confidence interval. IgG2a: 14 replicates, 7 mice; CD115: 14 replicates, 7 mice; CD4 and CD8 (CD4/8): 12 replicates, 6 mice. Macrophage and T cell quantification upon depletion is shown in Supplementary Fig. 7c. **h**, Depletion of macrophages (MP), but not of T cells, reduces clonal expansion of the top 20 enriched cancer gene perturbations. DESeq2 normalized reads of the top 20 enriched perturbations in control (IgG2a)-depleted, macrophage (CD115)-depleted or T cell (CD4/8)-depleted mice. In box plots, the centre line is the median, box edges delineate the first and third quartiles, and whiskers extend to the highest or lowest value no further than 1.5× the inter-quartile range from the box edge. *P* values by two-sided Wilcoxon rank-sum test.

and Extended Data Fig. 5c,d). To investigate the contribution of immune cells as a source of TNF in driving clonal expansion, we then focused on macrophages and T cells, both of which are known to express TNF (Fig. 3a). Following targeting of the 150 cancer genes in

E9.5 embryos, we selectively depleted macrophages or T cells from P4 to P60 and examined sgRNA representations at P60[34,35]. Whereas T cell depletion did not reduce clone sizes, macrophage depletion led to a small yet consistent reduction in the expansion rate of the top 20

enriched perturbations (Fig. 3f–h, Supplementary Fig. 7a–d and Supplementary Table 5). Given that immune cells were depleted only postnatally, when considerable expansion is already evident (Extended Data Fig. 3b), our findings suggest that macrophages contribute to clonal expansion in epithelia, but are probably complemented by other sources of TNF.

Together, the TNFR1 dependency and macrophage-depletion data provide strong experimental evidence supporting the functional role of the TNF signalling module in clonal expansions in normal epithelia and establish the module as an integral pathway downstream of cancer gene mutations.

## Autocrine TNF signalling in cancer

We next tested whether TNF signalling contributes to the transition from clonal expansions to tumour initiation. We injected our sgRNA library targeting the 150 cancer genes at E9.5. At P60, we induced chemical carcinogenesis with 7,12-dimethylbenz[*a*]anthracene (DMBA) and 2-*O*-tetradecanoylphorbol-13-acetate (TPA) on the clonally expanded mouse skin for 12 weeks (Extended Data Fig. 6a,b). We collected 168 tumours (papillomas and sSCCs) for individual tumour sgRNA capture and sorted mCherry-positive and negative tumour cells from 30 tumours for scRNA-seq analysis (without sgRNA capture).

The scRNA-seq analysis of the tumours unveiled distinct tumour cell clusters, including basal, differentiated and cycling tumour cells (Fig. 4a,b and Extended Data Fig. 6c,e). Consistent with reports in human sSCCs, we also observed a tumour-specific keratinocyte (TSK) population, located at the invasive front[36], that expressed marker genes such as *Mmp9* and *Mmp10* (Fig. 4a,b and Extended Data Fig. 6d). We then compared P60 EpSCs and tumour cells using reciprocal principal components analysis[37] (PCA) and observed that the large majority of TNF module genes were downregulated in tumour clusters (Fig. 4c,d and Extended Data Fig. 6f). Gene set enrichment analysis (GSEA) comparing basal tumour cells with P60 EpSCs underscored TNF signalling as the most prominently downregulated pathway (Extended Data Fig. 6g and Supplementary Table 6). Consistent with this observation, TNF signalling is also strongly downregulated in human sSCCs[38] compared with normal skin, including an overlap with 14 TNF signalling module genes (Extended Data Fig. 6h, Supplementary Fig. 8 and Supplementary Table 7).

We subsequently examined the TSK cluster, as its localization at the invasive front hints at its involvement in tumour progression[36]. Surprisingly, we observed that 8.5% of the TSK cluster cells robustly expressed endogenous TNF (Fig. 4e). TNF expression overlapped with TSK markers such as *Mmp9* and *Mmp10* (Fig. 4c,d). Moreover, TNF was co-expressed with TGFα and interleukin-1α (IL-1α), which were previously implicated in human epithelial cells as part of an autocrine cascade induced by TNF signalling[39], raising the possibility of a switch towards an autocrine TNF mode in tumorigenesis. We confirmed TNF protein expression in distinct KRT14-positive basal tumour cells in squamous cell carcinomas (SCCs) (Fig. 4g). Probing ligand–receptor interactions further underscored a potential autocrine TNF loop within TSKs (Fig. 4f and Extended Data Fig. 6i).

TNF signalling and EMT were the top upregulated pathways in TNF-positive versus negative TSKs (Extended Data Fig. 6j). However, only 3 out of 26 TNF signalling genes overlapped with the TNF signalling gene module, indicating that the cancer-associated TNF gene programme (hereafter referred to as the 'autocrine TNF gene programme') is distinct from the TNF module that is operative in clonal expansions. Moreover, whereas EMT was downregulated in clonal expansion (Fig. 2i), the autocrine TNF gene programme acted in concert with EMT. We corroborated these findings by co-culturing TNF-expressing and receiving keratinocytes, which confirmed that TNF-expressing keratinocytes induce EMT-related genes, irrespective of whether extracellular TNF is present (Supplementary Fig. 9).

Together, these findings suggest that a distinct, autocrine TNF gene programme is induced in TSKs, promoting EMT in this subpopulation that resides at the leading edges[36].

To chart the spatial organization of tumour clusters, we then performed Visium spatial transcriptomics across 7 tumours (Extended Data Fig. 7a–e). *Mmp10*-positive TSKs more closely resembled the human TSK signature (Extended Data Fig. 7f) and resided at the invasive front, forming invading nests and marking invasive fronts at the tumour base (Fig. 4h–j). Conversely, *Mmp10*-negative TSKs localized to differentiated regions (Fig. 4h–j and Extended Data Fig. 7b–d). Co-occurrence analysis[40] confirmed a cellular neighbourhood of *Mmp10*-positive TSKs with fibroblasts and endothelial cells, in line with a fibrovascular niche[36] (Fig. 4k). Furthermore, *Mmp10*-positive TSKs correlated with G2/M clusters, suggesting that they reside in proliferative tumour areas (Fig. 4k and Extended Data Fig. 7g).

Collectively, our findings imply a dual role for TNF and suggest that in the transition from clonal expansion to tumorigenesis, cancer cells downregulate the TNF module that is exploited in the clonal expansion phase and instead switch to an autocrine TNF–MMP9 and MMP10 axis at the invasive front.

## Clonal expansion and tumour initiation

To determine the relationship between clonal expansions and their role in cellular transformation, we individually sequenced the sgRNA amplicons in each of the 168 tumours and calculated the representation of each sgRNA (Fig. 5a). Overall, we found that *Notch1*, *Fat1*, *Trp53*, *Zmat3* and *Ahnak* had the highest representation in tumours (Fig. 5b). We then computed a selection score based on the comparison of sgRNA representation in tumours, normalized to the P60 skin representation. Focusing on the top 60 perturbations with robust P60 or tumour coverage, we observed that *Zmat3*, *Prkdc*, *Myh8*, *Kdr*, *Dnah3* and *Ahnak* showed the highest positive selection rates (Fig. 5c and Supplementary Table 8). By contrast, *Dscam*, *Zfhx4*, *Ttn* and *Fgf3* were among the perturbations that displayed the strongest negative selection, suggesting that these perturbations instead suppress tumour initiation (Fig. 5c).

Across all 150 perturbations, *Trp63* emerged as the top positively selected perturbation for tumour formation (Extended Data Fig. 8a), which was surprising considering the major depletion at P60 (Fig. 2a). However, beyond its well-established role in epidermal development, *Trp63*[+/−] mice exhibit spontaneous tumour formation, including sSCCs[41]. Notably, mice with heterozygous mutations in both *Trp63* and *Trp53* show a much higher incidence of SCCs and metastases compared with mice with single *Trp53*[+/−] mutations[41], implying that loss of *Trp63* can cooperate with other mutations in tumorigenesis.

To evaluate whether shared pathways confer a predisposition for transformation, we next examined gene-expression changes in positively versus negatively selected perturbations in P60 EpSCs (Fig. 5c). Our analysis revealed a significant enrichment of pathways related to EMT, suggesting that positively selected perturbations exhibit a predisposition for invasive features already at P60 (Fig. 5d and Extended Data Fig. 8b). Second, we used SEACells[42], which groups single cells into metacells to enable the identification of rare cell states. This analysis revealed that select metacells strongly associated with EMT gene expression, in line with a rare EpSC subpopulation being able to orchestrate EMT (Extended Data Fig. 8c,d). Third, we found a notable overlap between the genes upregulated in positively selected perturbations and TSK markers, with a subset also being induced by TNF (Extended Data Fig. 8e,f and Supplementary Fig. 10), consistent with reports that epithelial TNF induces Mmp genes during tumour cell migration[43]. Collectively, our analyses of how clonal expansions contribute to tumour initiation indicate that a TNF–TSK-like gene programme is already present in positively selected clones, which may partly mediate cancer predisposition of expanded clones.

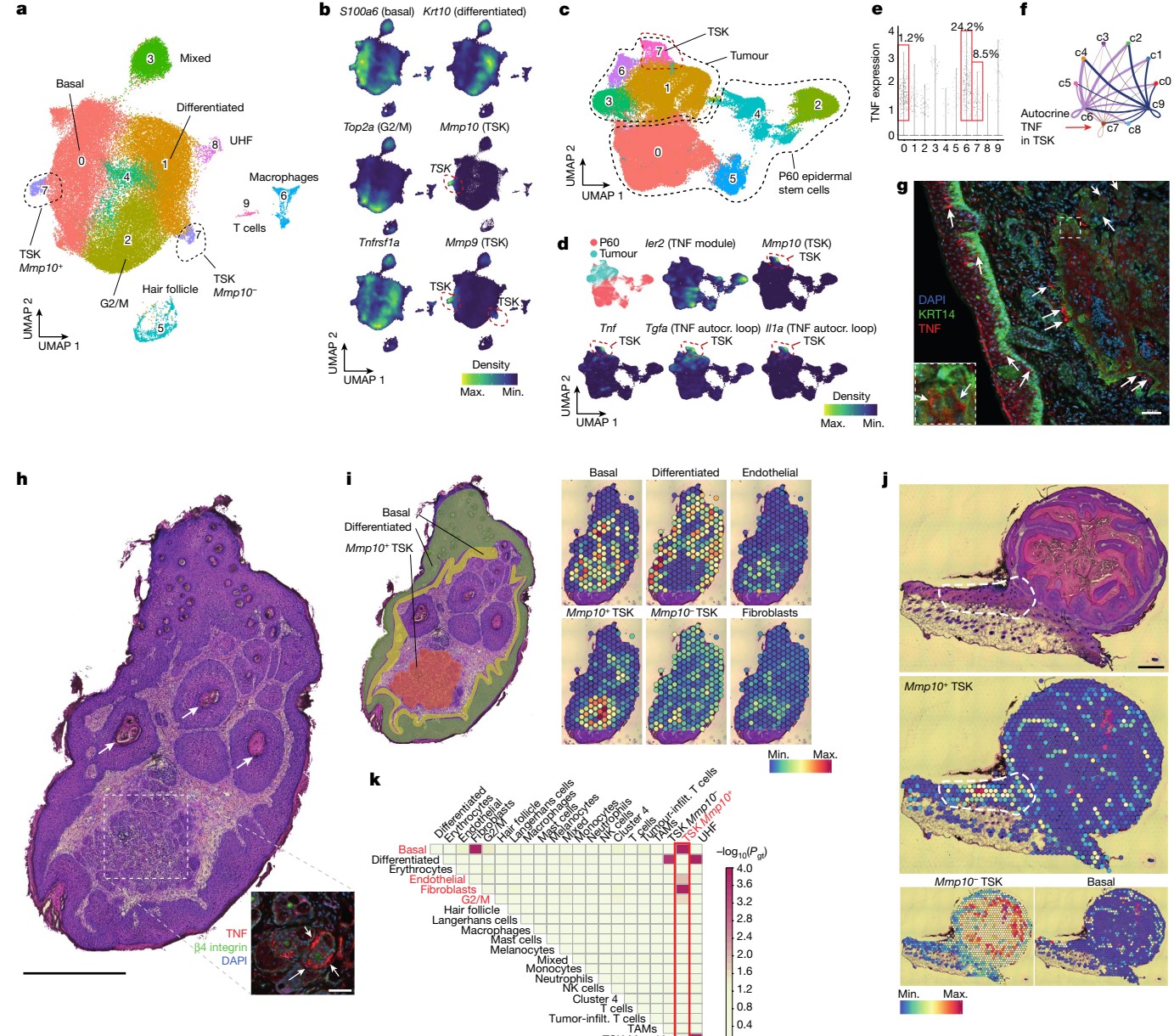

**Fig. 4 | Cancer cells switch to an autocrine TNF gene programme. a**, UMAP representation of single-cell RNA sequencing data from mCherry-positive tumour cells (*n* = 61,303 cells from 30 tumours, 2 mice). **b**, Visualization of tumour cell-type-specific marker gene expression by Nebulosa density plots on the mCherry-positive tumour UMAP in **a**. **c**, UMAP representation of integrated P60 EpSCs and mCherry-positive tumour clusters 0 and 7 using reciprocal PCA in Seurat. **d**, Visualization of marker gene expression by Nebulosa density plots on the integrated P60 and tumour UMAP in **c**. Autocr., autocrine. **e**, TNF expression and frequency within each cluster of mCherry-positive tumours. Red boxes indicate cells expressing *Tnf* in the corresponding clusters (>0.5 normalized counts). **f**, CellChat circle plot showing TNF signalling ligand–receptor interactions across mCherry-positive tumour clusters. **g**, Immunofluorescence of a squamous cell carcinoma section shows regions with TNF-expressing tumour cells (arrows). KRT14 is used as a basal cell marker. Inset, KRT14-positive TNF-expressing basal tumour cells. Scale bars, 50 μm. **h**, Haematoxylin and eosin (H&E)-stained section of a tumour used for Visium

spatial transcriptomics. Arrows indicate keratin pearls (histological hallmark of SCCs). The outlined region shows the *Mmp10*-positive TSK cluster. The inset shows an immunofluorescence image of TNF-expressing tumour cells embedded within cells expressing the basal membrane marker β4-integrin. Scale bars: 0.5 mm (main image); 50 μm (inset). **i**, Right, Visium spatial transcriptomics and spatial maps show tumour cluster localization, deconvoluted from the single-cell reference (Extended Data Fig. 7e,f). Left, spatial distribution of cell types in an H&E-stained tumour section based on the histology and spatial maps. **j**, H&E-stained sections and spatial maps show *Mmp10*-positive TSKs at the invasive front (delineated by the white dashed line). Scale bar, 0.5 mm. **k**, Co-occurrence analysis shows a positive correlation between the localization of *Mmp10*-positive TSKs, basal tumour cells, endothelia and fibroblasts (marked in red). *P*(gt) is the probability of the observed co-occurrence being greater than the expected co-occurrence. Infilt., infiltrating; NK cells, natural killer cells; TAM, tumour-associated macrophages.

## Autocrine TNF induces invasion

We hypothesized that the transition from clonal expansion to early alterations in tissue architecture might be triggered by epithelial TNF.

To test this hypothesis in vivo, we designed a lentiviral TNF expression construct (Fig. 5e). We confirmed TNF expression in keratinocytes and detected both secreted and membrane-bound TNF (Fig. 5f, Extended Data Fig. 9a and Supplementary Fig. 9). TNF also promoted cellular

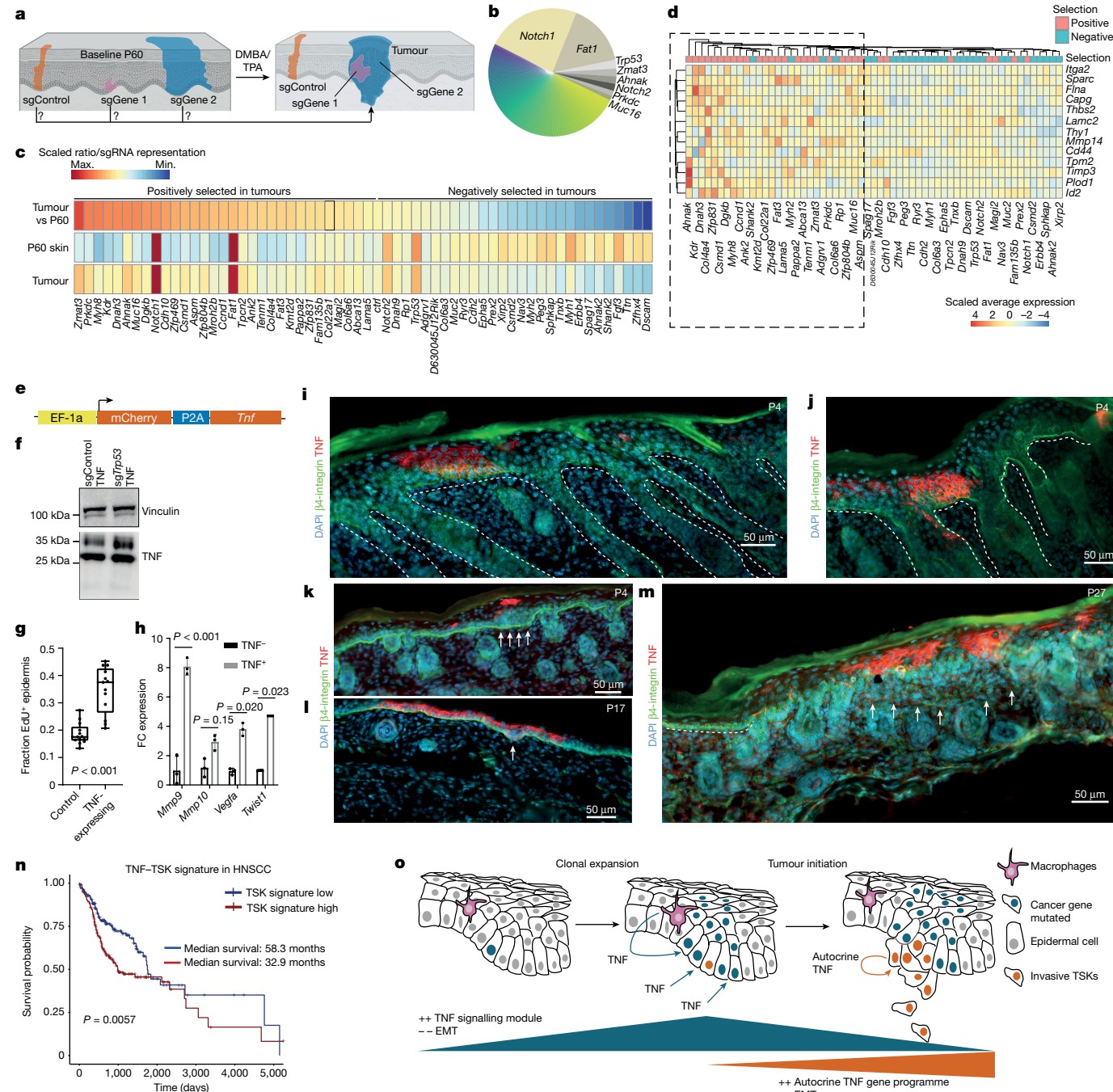

**Fig. 5 | Epithelial TNF induces invasive properties. a**, Schematic representation of the selection rate as a proxy for the predisposition of each perturbation for transformation. **b**, Pie chart of total sgRNA representation across 168 sequenced tumours. **c**, Heat map showing selection score between P60 skin and tumours, highlighting positively and negatively selected perturbations. Full list in Extended Data Fig. 8a. **d**, EMT is induced in positively selected perturbations in the P60 skin, identified through differential gene expression between positively and negatively selected perturbations. **e**, Schematic of lentiviral vector for microinjection into E9.5 embryos to express TNF in epithelial cells. **f**, Western blot showing the 26 kDa membrane-bound TNF in keratinocytes transduced with the lentiviral TNF expression construct. P2A, porcine teschovirus-12A; sgControl, control sgRNA; sg*Trp63*, sgRNA targeting *Trp63*. **g**, Increased proliferation in TNF-expressing P4 epidermis shown by EdU incorporation 1 h after injection. In box plots, the centre line represents the median, box edges delineate first and third quartiles, and whiskers extend to minimum and maximum values. Twelve independent TNF-expressing and adjacent control areas were quantified (Extended Data Fig. 9e). *P* value by one-way ANOVA. **h**, Epithelial TNF expression induces EMT markers. P4 mCherry-positive and mCherry-negative cells were sorted and analysed by reverse transcription–quantitative PCR (RT–qPCR). Data are mean ± s.d. from three technical replicates from two mice. *P* values by one-sided Welch's *t*-test. **i,j**, Immunofluorescence of P4 back skin sections depicting two different clones of TNF-expressing keratinocytes. Dashed lines indicate basal membrane. **k–m**, Immunofluorescence of P4, P17 and P27 epidermal sections reveals regions of epidermal hyperproliferation (**k**), epithelial invaginations into the ear dermis (**l**) and breakdown of the basal membrane (**m**; arrows). **n**, TSK signature mRNA levels correlate with shorter overall survival in human patients with HNSCC, stratified by upper tertile and bottom tertile TSK signature mRNA expression. *P* value by standard log-rank test. Multivariate analysis is shown in Extended Data Fig. 10. **o**, Working model illustrating the distinct TNF programmes in tumour evolution.

motility in scratch assays and enhanced keratinocyte proliferation (Extended Data Fig. 9b–d).

Having validated our construct, we then injected high-titre lentivirus into E9.5 embryos to express TNF in epithelial cells. P4 analysis confirmed distinct TNF-expressing clones displaying defined basal membranes (Fig. 5i,j). However, in time, TNF-expressing clones became associated with various abnormalities, including epidermal hyperproliferation, confirmed by a strong increase in EdU incorporation (Fig. 5g,k and Extended Data Fig. 9e), and epithelial invaginations (Fig. 5l). Furthermore, TNF-expressing clones induced EMT markers such as *Vegfa*, *Mmp9* and *Twist1* already in P4 skin (Fig. 5h and Extended Data Fig. 9g). Finally, in P27 mice, we observed various examples of TNF-expressing clones in which the basal membrane exhibited partial or complete dispersion, leading to invasion into the surrounding tissue (Fig. 5m and Extended Data Fig. 9f). Together, these findings provide direct in vivo evidence that the epithelial TNF gene programme is sufficient to induce invasive properties of EpSCs.

Finally, to explore the significance of the TNF–TSK axis in human SCCs, we tested whether TSK markers could stratify the overall survival of patients with SCC. Notably, several TSK marker genes such as *Mmp9*, *Inhba* and *Nt5e* strongly correlated with shorter overall survival across different SCC types (Extended Data Fig. 10a–c). We then created a TSK signature, which stratified the overall survival of patients with HNSCC, with a median survival difference of more than two years (Fig. 5n). Multivariate analysis confirmed that the TSK signature is an independent prognostic factor for survival (Extended Data Fig. 10d,e). Together, these findings underscore the potential relevance of the invasive TSK signature in driving human cancer and position the TNF–TSK gene programme as a target for cancer therapy.

## Discussion

Here we established a scalable in vivo single-cell CRISPR platform to systematically dissect tissue-wide clonal expansions of 150 cancer gene perturbations at single-cell transcriptomic resolution. We provide multiple lines of evidence that a TNF signalling gene module in EpSCs, stimulated in part by macrophage-derived TNF, is a generalizable driver of clonal expansions in phenotypically normal skin. By contrast, during the transition from clonal expansion to tumour initiation, we identified a switch to a distinct autocrine TNF gene programme at invasive tumour fronts[36], which is associated with ECM-remodelling factors such as MMP9 and MMP10 to facilitate tissue invasion. Given that autocrine TNF is sufficient to induce invasive properties in EpSCs and in human cancer[44–47], our findings suggest that the switch to autocrine TNF represents a key step in tumorigenesis (Fig. 5o).

The distinction of two separate TNF programmes is relevant in the context of TNF inhibitors used in chronic inflammatory diseases. As TNF inhibitors have shown efficacy in only a subset of patients with cancer[26,48,49], our findings underscore the need for a better understanding of how TNF inhibitors affect the autocrine TNF gene programme. Whereas TNF antibodies effectively targeted stromal-derived TNF, they did not suppress the autocrine TNF gene programme (Supplementary Fig. 9). Exploiting specific vulnerabilities of autocrine TNF-expressing tumours, as demonstrated with SMAC mimetics[50], could serve as a potential therapeutic avenue and guide the development of targeted autocrine TNF inhibitors.

Collectively, our study demonstrates the power of applying in vivo single-cell CRISPR to mammalian tissues, highlights the multifaceted roles of clonal expansions in epithelia and unveils a switch from a TNF gene module to an autocrine TNF programme during tumour evolution. Given the strong correlation between TSK signature expression and shorter survival in human cancer, our findings provide a foundation for developing novel strategies for cancer prevention and therapy.

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

# Methods

## Selection of target genes and sgRNA design

Five hundred sgRNAs were selected as initial targeting library. Fifty out of the 500 sgRNAs were non-targeting control guide RNAs. Three sgRNAs per gene targeted the top 150 most frequently mutated or CNV genes in human HNSCC and sSCC. We selected genes that have a mutation frequency of ≥6% in HNSCC, ≥30% in sSCC and ≥15% of copy number alterations in both. Mutation frequency was assessed using the cBioPortal for Cancer Genomics[51,52] with the datasets for sSCC[53,54] and HNSCC[55–59]. The exact selection of genes and their mutation frequency can be found in Extended Data Fig. 1. The overlap and differences between HNSCC and sSCC genes is graphically represented in Fig. 1b (Venn diagram). Individual guides were designed with the Broad Institute sgRNA design tool (sgRNA Designer: CRISPRko (https://portals.broadinstitute.org/gpp/public/analysis-tools/sgrna-design)). The three top-ranking guides per target were chosen. Fifty non-targeting control sgRNAs were randomly picked from the Brie library[60] (Addgene #73633). All 500 sgRNA sequences are listed in Supplementary Table 9.

## CROP-mCherry vector and library cloning

The puromycin resistance cassette in the original CROP-seq-Guide-Puro vector (Addgene #86708) was replaced with a mCherry sequence, amplified from pAAVS1-NDi-CRISPRi (Gen1) (Addgene #73497) and cloned in via Pfl23II and MluI restriction sites for FACS. The 500 sgRNA sequences were ordered as an oligonucleotide pool from IDT, cloned in batch via Gibson assembly or via BsmbI restriction sites for sgRNAs as previously described[5]. For all single-cell experiments, three different library batches were cloned and individually sequenced to confirm homogeneous sgRNA representation.

## High-titre lentivirus production

Production of vesicular stomatitis virus (VSV-G) pseudotyped lentivirus was performed by calcium phosphate transfection of Lenti-X 293 T cells (TaKaRa Clontech, 632180) with CROP-mCherry and helper plasmids pMD2.G and psPAX2 (Addgene plasmids 12259 and 12260). Sixteen hours after transfection, medium was changed to viral production medium (DMEM (Gibco 11965092), 1% penicillin-streptomycin-glutamine (Gibco 10378016), 1% 100 mM sodium pyruvate (Gibco 11360070), 1% sodium bicarbonate 7.5% solution (Gibco 25080094), 5 mM sodium butyrate (Sigma-Aldrich B5887)) and incubated at 37 °C/7.5% $CO_2$. Viral supernatant was collected 46 h after transfection and filtered through a 0.45-μm filter (Millipore Stericup Quick Release Durapore S2HVU02RE). For in vivo lentiviral transduction, the viral supernatant was concentrated ~2,000-fold using a 100 kDa MW cut-off Millipore Centricon 70 Plus (Merck Millipore; UFC710008). The virus was further concentrated by ultracentrifugation using the SW 55 Ti rotor for the Beckman Coulter Optima L-90 Ultracentrifuge at 45,000 rpm and 4 °C. Final viral particles were resuspended in viral resuspension buffer (20 mM Tris pH 8.0, 250 mM NaCl, 10 mM MgCl₂, 5% sorbitol) and stored at −80 °C until used for titration or injection. For low-titre virus, production of lentivirus was performed similarly as described above, and viral supernatant was collected and filtered using a 0.45-μm syringe filter (Sarstedt 83.1826) and stored at −80 °C until used.

## In vivo experiments

All animal experiments were conducted in strict accordance with the Swiss Animal Protection law and requirements of the Swiss Federal Office of Food Safety and Animal Welfare (BLV). The Animal Welfare Committee of the Canton of Zurich approved all animal protocols and experiments performed in this study (animal permits ZH074/2019, ZH196/2022).

Genetically modified mice of the strain *Tg(B6J.129(Cg)-Gt(ROSA)26Sor^tm1.1(CAG-cas9*,-EGFP)Fezh/J* (denoted as B6.Cas9) were purchased from the Jackson Laboratory (strain 026179). B6.129 S *Tnfrsf1a^tm1Imx Tnfrsf1b^tm1Imx* (denoted as *Tnfrsf1a*-KO, originally from Jackson

Laboratory 003243) were acquired through the Swiss Immunology Mouse Repository (SwImMR). Wild-type mice of the strain CD1-IGS (denoted as CD1) were purchased from Charles River. B6.Cas9 were crossbred either with *Tnfrsf1a*-knockout or CD1 mice to provide embryos heterozygous for the Cas9 allele, suitable for lentivirus injection at embryonic development stage E9.5.

Ultrasound-guided in utero injections were conducted as previously described[61]. In brief, females at E9.5 of gestation were anaesthetized with isoflurane. Each embryo was injected with 0.5 μl of high-titre lentivirus, and up to 8 embryos were injected per litter. Surgical procedures were limited to 30 min to ensure fast recovery.

All mice were housed at the Laboratory Animal Services Center (LASC) of the University of Zurich in individually-ventilated cages in a humidity- and light-controlled environment (22 °C, 45–50%, 12 h light/dark cycle), and had access to food and water ad libitum. B6.Cas9 males used for crossbreeding and pregnant CD1 females were housed individually. All other mice were group-housed. Successful infection of mice was controlled by eye through excitation with a dual fluorescent protein flashlight (NIGHTSEA, DFP-1). Positively infected mice were then euthanized by decapitation (P4) or with $CO_2$ (P60), shaved and treated with hair removal cream if necessary.

P4 back skin was processed as previously described[8], with minor modifications. After being surgically removed and scraped for fat, the back skin was washed once in cold PBS and then placed in dispase (Corning 354235) for 35 min at 37 °C, dermis-side facing down, on an orbital shaker. Next, epidermis was separated from dermis with fine forceps, torn into smaller pieces and placed in 4 ml 0.25% Trypsin-EDTA (1×, Gibco; 25200056) for 15 min at 37 °C with orbital shaking. Epidermis was washed with cold PBS and pipetted vigorously to achieve a single-cell suspension. Suspension was then afterwards filtered through a 70-μm and a 40-μm strainer (Corning; 431750, 431751) consecutively, then centrifuged for 10 min at 400g and resuspended in FACS buffer (PBS + 2% chelexed FBS (FBS(-)) + DAPI).

P60 back skin was processed as previously described[62]. In brief, back skin was harvested with surgical scissors and placed dermis-facing up on a styrofoam tray with pins. Fat and subcutis was scraped off with a scalpel. Fat-free skin was then washed, dermis-side facing down in 1× PBS. Skin was then placed in the same orientation in 0.5% Trypsin-EDTA (10×, Gibco; 15400054) and incubated at 37 °C on an orbital shaker for 25 (females) or 50 min (males). Using a glass microscopy slide, skin was secured to the bottom of the dish and scraped again with a scalpel until it started to break. Excess cold PBS with 2% FBS(-) was added to neutralize trypsin. Cell suspension was strained on ice first through a 70-μm strainer, washed with an additional 15 ml of cold 1× PBS with 2% FBS(-) and then filtered through a 40-μm strainer. Cell suspension was spun down at 400g for 10 min and resuspended in FACS buffer.

P4: a total of 58 injected P4 mice were collected and assessed in 7 independent single-cell RNA-seq runs. P60: a total of 10 injected mice were collected and assessed in 8 independent single-cell RNA-seq runs. Because the P4 skin contains less cells, more mice had to be pooled to sort a sufficient number of mCherry-positive cells.

mCherry-positive cells were sorted on a BD FACSAria III using a 70-μm nozzle and processed for single-cell capture with a BD Rhapsody Single-Cell Analysis System using BD Rhapsody Cartridge Kit (633733). For P4, 7 separate cartridges with an average of 50,037 cells were loaded. For P60, 8 cartridges with an average of 64,625 cells were loaded. Reverse transcription from beads and sequencing library production was carried out according to the manufacturer's instructions (BD Biosciences, doc ID: 210967 rev. 1.0).

## Dial-out nested PCRs for sgRNA amplification

The sgRNA-containing region was amplified from BD Rhapsody beads in a separate nested PCR reaction. In PCR1, we used 100 μl KAPA HiFi HotStart ReadyMix (Roche 07958935001), 6 μl forward primer (5′-ACACGACGCTCTTCCGATCT-3′, 10 μM), 6 μl reverse primer

(5′-TCTTGTGGAAAGGACGA-3′, 10 μM), 12 μl Bead RT/PCR Enhancer reagent from BD Biosciences and 72 μl nuclease-free water for a total volume of 200 μl. Rhapsody beads from each separate cartridge preparation were resuspended in the PCR mix, aliquoted in 4 × 50 μl reactions and quickly moved into a pre-heated PCR machine without allowing beads to settle. The following conditions were used: Initial denaturation at 95 °C for 5 min, followed by 25 cycles of denaturation 95 °C for 30 s, annealing 53 °C for 30 s, extension 72 °C for 20 s and a final extension of 10 min at 72 °C.

PCR1 product was pooled, beads were removed magnetically and amplicons were cleaned up with Agencourt AMPure XP beads (Beckman Coulter A63881) according to the manufacturer's instructions before the second PCR. For PCR 2, 3 μl of cleaned up PCR1 was used as template along with 2 μl forward primer (5′-ACACGACGCTCTTCCGATCT-3′, 10 μM), 2 μl reverse primer (CAGACGTGTGCTCTTCCGATCTCTTGTGGA AAGGACGAAACA*C*C*G-3′, 10 μM), 18 μl nuclease-free water, along with 25 μl of KAPA HiFi HotStart ReadyMix for a total volume of 50 μl. PCR2 was performed under the following conditions: Initial denaturation at 95 °C for 3 min, followed by 10 cycles of denaturation 95 °C for 30 s, annealing 60 °C for 3 min, extension 72 °C for 60 s, and a final extension of 5 min at 72 °C. PCR2 product was cleaned up with Agencourt AMPure XP beads according to the manufacturer's instructions before the indexing PCR. Indexing PCR was performed according to BD Biosciences 'mRNA Targeted Library Preparation' protocol (doc ID: 210968 rev. 3.0).

## sgRNA enrichment and depletion calculation

To calculate enrichment and depletion of sgRNAs, we used sgRNA cell counts for P4 and P60. For T0, read counts from the sequenced pre-injection libraries A, B and C originating from separate batch plasmid preparations were used. As each P$x$ (P60 or P4) sample received one of the three T0 library batches, we normalized each sample individually as follows. Counts at P$x$ and T0 were first transformed to proportions over the total counts, either of each entire sample or library batch. Then, each P$x$ sample guide proportions were normalized by dividing against its T0 proportion (that is, its initial representation in the matching pre-injection library). This yields a fold change (of ratios). Fold changes from triplets of guides targeting the same gene were then averaged together (and indicated with the symbol of the targeted gene), and so were the 50 control guides (grouped for a total of 17 pseudo-triplet controls: ctrl_1 for control sgRNA from 1 to 3, ctrl_2 for control sgRNA from 4 to 6, etc, and ctrl_17 which comprised only control sgRNA 49 and 50). This resulted in 167 fold change values in each P$x$ sample: 150 for the target genes and 17 for the pseudo-triplet controls. Ranks were defined for each P$x$ sample based on these fold changes in decreasing order, therefore with the highest fold change ranking first. Ties (typically, fold changes with value zero) were assigned to the same lowest rank (see min_rank function from R package dplyr). These ranks were used to compute the correlation of the top 20 and bottom 20 perturbations (guide triplets) among the eight P60 samples (biological replicates), as shown in Extended Data Fig. 1h.

Finally, the fold changes obtained were also averaged across the same-time point samples (either P4 or P60). For this average, a weighted mean was used, with the weights being the total number of guide-positive cells for each sample. These results are reported in $\log_2$ scale in Fig. 2a and in Extended Data Figs. 1i and 3. For the P60 versus P4 guide representation (Extended Data Fig. 3c), the fold changes shown in Extended Data Fig. 3a were divided by those in Extended Data Fig. 3b (by matched perturbation identity) and re-ranked. Epidermal stem cell-specific sgRNA representations (as shown in Extended Data Fig. 3d) were computed with the same approach, after subsetting the P4 or P60 datasets to include only epidermal stem cells (cluster 0, in Fig. 1e,h).

## DMBA/TPA chemical carcinogenesis

Two-stage cutaneous chemical carcinogenesis was performed as previously described[63]. In brief, the back skin of 6- to 8-week-old female mice were shaved with electric hair clippers and treated once with 400 nmol DMBA (Sigma-Aldrich D3254) per 100 μl (dissolved in acetone) as an initiator mutagen. After a 2-week rest period, 40 nmol TPA (Sigma-Aldrich 79346) dissolved in 100 μl 100% ethanol was applied twice a week. Tumour formation was monitored regularly by visual inspection and mice were euthanized after 12 weeks of treatment. The maximum cumulative tumour size of 2 cm, as permitted by the Animal Welfare Committee of the Canton of Zurich (animal permits ZH074/2019, ZH196/2022), was not exceeded.

## Tumour preparation

Tumours and 1–2 mm of surrounding tissue from DMBA/TPA treated mice were excised and finely minced with a scalpel. Tumour pieces were submerged in 5 ml pre-warmed DMEM with 0.25% Trypsin/EDTA and 3 U ml$^{-1}$ DNAse I at 37 °C on an orbital shaker with rigorous resuspension every 5 min for a total of 30 min. The reaction was quenched with 1 ml FBS and strained through a 70-μM mesh, pelleted at 400$g$ for 10 min with excess PBS and resuspended in 0.5 ml FACS buffer before sorting. Single-Cell capture was performed with BD Rhapsody as described above using Rhapsody Enhanced Cartridge Kit (BD Biosciences 664887). Reverse transcription from beads and sequencing library production was carried out according to manufacturer's instructions (BD Biosciences, doc ID: 210967 rev. 1.0).

## Tumour preparation for sgRNA amplification

DMBA/TPA-induced tumours that were not used for either scRNA-seq or spatial analysis were minced. Genomic DNA was extracted using the Qiagen DNeasy Blood and Tissue Kit. sgRNA region was amplified in a nested PCR. In PCR1, we used 12.5 μl KAPA HiFi Hot-Start ReadyMix (Roche 07958935001), 0.75 μl forward primer (5′-CTTGTGGAAAGGACGAAACACCG-3′, 10 μM), 0.75 μl reverse primer (5′-GTGTCTCAAGATCTAGTTACGCCAAGC-3′, 10 μM), 1 μl of extracted genomic tumour DNA (with a concentration between 50–100 ng μl$^{-1}$) and 10 μl nuclease-free water for a total volume of 25 μl per tumour. PCR1 was performed using the following conditions: Initial denaturation at 98 °C for 2:30 min, followed by 22 cycles of denaturation 98 °C for 20 s, annealing 62 °C for 30 s, extension 72 °C for 30 s. PCR1 products for each tumour were individually column-purified with a GenElute PCR Clean-Up Kit (Sigma-Aldrich NA1020-1KT). For PCR2, 1 μl of cleaned up PCR1 was used as template along with 0.75 μl forward barcoded i5 primer (see Supplementary Table 1, 10 μM), 0.75 μl reverse barcoded i7 primer (see Supplementary Table 1, 10 μM), 10 μl nuclease-free water, along with 12.5 μl of KAPA HiFi HotStart ReadyMix for a total volume of 25 μl. Eight different i5 and i7 primers allowed for a total of 64 different combinations. PCR2 was performed under following conditions: initial denaturation at 95 °C for 3 min, followed by 32 cycles of denaturation 98 °C for 30 s, annealing 62 °C for 30 s, extension 72 °C for 30 s. PCR2 products for a maximum of 64 individually barcoded tumour amplifications were pooled and cleaned up with Agencourt AMPure XP beads (Beckman Coulter A63881) according to manufacturer's instructions before sequencing.

## Single-cell RNA sequencing

With the exception for whole-transcriptome amplification analysis 7 (sent to Novogene and sequenced on a S4 flow cell of an Illumina Novaseq), all single-cell or amplicon sequencing was prepared as ready-made libraries and sequenced either on Illumina NovaSeq or Nextseq 500 instruments at the Functional Genomics Center Zurich. Sequencing libraries were checked for peak size and concentration on a Bioanalyzer 2100 using DNA High Sensitivity Chip. Concentrations were additionally measured using a Qubit Fluorometer. Whole-transcriptome amplification analyses from Rhapsody Cartridges were sequenced in full SP flow cells at 1.8 nM concentration with 20% PhiX and the following read configuration: Read 1 60, I1 8, Read 2 62 cycles. Tumours were sequenced with 10% PhiX, dual indexing i5, 8 bp; i7, 8 bp; and Read 1, 64 cycles.

## Bulk RNA sequencing and processing

mCherry-positive or negative keratinocytes were FAC-sorted on a BD FACSAria III directly into TriZol LS and column-purified with a Direct-zol RNA Miniprep kit (Zymo Research). Sequencing libraries from 1,000 ng of RNA per sample were prepared with the TrusSeq mRNA library kit and sequenced on a NovaSeq X 2 × 150 bp at the Functional Genomics Center Zurich. Quality control of the paired-end bulk RNA-seq data was performed using the FastQC program. Adapter trimming of the Illumina Universal Adapter (AGATCGGAAGAG) was carried out by Cutadapt. Transcript level quantification was performed by Salmon[64] using the Gencode Mouse M25 reference transcriptome. DESeq2 was utilized to identify the differentially expressed genes between different conditions[65]. Initial quality control of the count data was carried out by PCA and hierarchical clustering using top 500 variable genes after variance stabilizing normalization. The gene level count data were normalized by the median of ratios method and dispersion was estimated. Next a generalized linear model was fitted to identify differentially expressed genes. The *P* values were obtained by the Wald test and corrected for FDR using the Benjamini–Hochberg procedure. FDR cut-off <0.05 was used to detect significant differentially expressed genes for subsequent analysis.

## Immunofluorescence

Back skin from P4 and P60 mice was scraped to remove fat, placed on Whatman paper and cut into strips. Sections were embedded in OCT (Tissue-Tek; 4583), frozen by placing them in a liquid nitrogen-cooled isopentane filled metal beaker, stored at −80 °C and sectioned at 10–12 µm thickness on a Leica CM1900. Sections were immobilized on Superfrost glass slides, fixed for 10 min at room temperature with 4% paraformaldehyde (PFA), washed twice in 1× PBS before blocking in blocking solution (1% BSA, 1% gelatin, 2.5% normal goat serum, 2.5% normal donkey serum, 0.30% Triton X-100, 1× PBS). Primary antibodies (TNF [D2D4] XP[R] Rabbit anti-mouse monoclonal antibody, CST 11948; 1:300) and purified rat anti-mouse CD104 (also known as β4-integrin) (BD Pharmingen 553745, 1:300) were incubated overnight at 4 °C. After washing twice with 1× PBS, slides were incubated with secondary antibodies (Alexafluor488, Cy3, or AlexaFluor647, Jackson ImmunoResearch Laboratory; 1:500–1:1,000) and 0.5 µg ml⁻¹ 4′,6-diamidino-2-phenylindole (DAPI) at room temperature for 1 h. Sections were then washed again with 1× PBS, dried, covered with home-made mounting medium and sealed with nail polish before image acquisition. Pictures were acquired with a 20× objective on a ZEISS Axio Observer microscope controlled by the ZEN microscopy software (version 3.1).

## Whole-mount immunofluorescence and antibodies

Dissected back skin from P4 and P60 mice injected with CROP-mCherry library were fixed in 4% PFA for 1 h at room temperature. Following fixation, samples were permeabilized in 0.8% PBS-Triton overnight. All steps during staining were carried out at room temperature. Primary antibodies were diluted into blocking buffer (5% donkey serum, 2.5% fish gelatin, 1% BSA, 0.8% Triton in PBS) and were incubated for at least 16–20 h at room temperature. Samples were then washed for 3–4 h in 0.8% PBS-Triton, and incubated with secondary antibodies (in blocking buffer) together with DAPI (to label nuclei, 0.25 mg ml⁻¹) for at least 16–20 h. After staining, samples were extensively washed with 0.8% PBS-Triton every hour for 4–6 h. Samples were then dehydrated in increasing concentrations of Ethanol: 30%, 50% and 70% in doubled-distilled water (with the pH adjusted to 9.0 with NaOH/HCL) for 1 h each, and finally in 100% ethanol for 1 h. For tissue clearing, samples were transferred to Eppendorf tubes with 500 µl ethyl cinnamate (ECi, Sigma 112372) and shaken overnight at room temperature under dark conditions. Fresh ECi was used for mounting for imaging.

For fixed sections, back skin was cut into strips, embedded and frozen in OCT (Leica), and sectioned with a Leica cryostat (10-µm sections). Sections were permeabilized in 0.3% PBS-Triton for 15 min at room temperature, blocked in blocking buffer (5% donkey serum, 2.5% fish gelatin, 1% BSA, 0.3% Triton in PBS) for 15 min at room temperature. Primary antibodies in blocking buffer were incubated on slides overnight at 4 °C, washed off with 0.3% PBS-Trition before adding secondary antibodies, with DAPI for 1 h at room temperature. After washing off the secondary, samples were mounted for imaging using ProLong Diamond Antifade Mountant (P36961, Invitrogen).

Antibodies used were as follows: rat anti-RFP (Chromotek, 5F8; 1:200), rabbit anti-RFP (MBL, PM005; 1:200), chicken anti-GFP (Abcam, ab13970; 1:200), rat anti-CD45-biotin (Biolegend, 103104; 1:200), goat anti-TNFR1 (R&D, AF-425-PB; 1:200). All secondary antibodies used were raised in a donkey host and were conjugated to Alexa Fluor 488, Cy3 or Alexa Fluor 647 (Jackson ImmunoResearch Laboratory; 1:500-1:1000). 4′,6-diamidino-2-phenylindole (DAPI) was used to label nuclei (0.25 mg ml⁻¹).

## Whole-mount microscopy

Whole-mount images were acquired using an LSM980/Airyscan module combined with Tiling (where indicated) 40× multi-immersion objective (LD LCI Plan-Apochromat 40×/1.2 Imm Corr DIC (silicone oil, glycerol or water immersion), WD 0.41 mm) with glycerine as an immersion medium for cleared samples, or a 40× oil immersion objective (Plan-Apochromat 40×/1.4 Oil DIC, WD 0.13 mm) for sections.

## Whole-mount image processing and analysis

All processing was done in Zen Blue, with a custom batch processing macro (Thomas Peterbauer). All images depicted are maximum intensity projections.

## In vivo EdU labelling

One hour before they were euthanized, mice were injected intraperitoneally (50 µg per g body weight) with 5-ethynyl-2′-deoxyuridine (EdU, Targetmol, TMO-T17341-500mg) dissolved in PBS at a concentration of 10 mg ml⁻¹. After cryo-sectioning, click-chemistry was performed for staining with the In Vivo EdU Click Kit 488 (Merck, BCK488-IV-IM-S) according to manufacturer's instructions. In brief, 10–12 µm thin frozen tissue sections were fixed with 4% PFA in PBS for 15 min at room temperature. Cells were washed twice with 1× PBS before permeabilization with 0.3% Triton X-100 in PBS for 15 min at room temperature. Five hundred microlitres of EdU reaction cocktail was added to each section and incubated for 30 min at room temperature, protected from light. Afterwards, sections were washed in PBS and antibody staining was carried out as described above.

## Keratinocyte culture

Newborn, primary mouse epidermal keratinocytes derived from B6-LSL-Cas9-eGFP mice were isolated as previously described[66]. In brief, isolated epidermal keratinocytes were cultured on 3T3-S2 feeder layer previously treated with Mitomycin-C in 0.05 mM Ca²⁺ E-media supplemented with 15% serum. After 3 passages on 3T3-S2 feeder layer, cells were cultured in 0.05 mM Ca²⁺ E-media, made in house as previously described[14]. The B6-LSL-Cas9-eGFP keratinocytes were transiently transfected with 1 µg of a Cre-expressing plasmid in 6 wells using Lipofectamine 2000 (Invitrogen 11668-027) to activate Cas9–GFP. Cells were then sorted for GFP-positive signal on a BD Aria III and put back in culture using 0.05 mM Ca²⁺ E-media. For this project, all further modified cell lines derived from the Cas9–GFP activated cell line and were grown at standard conditions, 37 °C and 5% CO₂. The SCC cell line was previously isolated from DMBA/TPA chemically induced skin tumours and brought into culture. Cell lines were tested for mycoplasma using the Mycoplasma PCR detection kit (Sigma; D9307).

## In vitro lentivirus infections

For lentiviral infections in culture, B6.Cas9 keratinocytes (see above) were plated in 6-well plate (Thermo Scientific Nunclon TM Delta Surface 140675) at $1.5 \times 10^5$ cells per well and infected with 100–300 µl of low-titre virus in the presence of infection mix (1 in 10 dilution of polybrene (10 mg ml$^{-1}$ Sigma 107689-100MG in PBS) in FBS[-]), by centrifuging plates at 1,100g for 30 min at 37 °C in a Thermo Heraeus Megafuge 40 R centrifuge. Infected cells were sorted by FACS for mCherry on a BD FACSAria III. Viruses used for generation of knockout keratinocytes carried the following sgRNAs: *Trp53*_sgRNA.3, *Fat1*_sgRNA.3, *Notch1*_sgRNA.2, ctrl_44. Additionally, we used constructs carrying sgRNAs and *Tnf* cDNA: *Trp53*_sgRNA.3 + *Tnf*, *Notch1*_sgRNA.2 + *Tnf*, ctrl44-*Tnf*.

## Enzyme-linked immunosorbent assay

Mouse TNF uncoated ELISA Kit from Invitrogen (88-7324-22) was used according to the manufacturer's instructions to quantify TNF concentration in supernatants from confluent 6 wells of keratinocyte cultures after 24 h of incubation. Measurements of 96-well plates in technical and biological triplicates from separate wells were taken on a Tecan Infinite M1000Pro plate reader. Concentrations in supernatant were calculated according to included recombinant TNF standard curve.

## RT–qPCR

In vitro-infected keratinocytes and SCC cells were lysed in 6 wells with 1 ml TRIzol Reagent and the RNA was purified using the Direct-zol RNA Miniprep kit (Zymo Research). The procedure was performed according to the manufacturer protocol except for an additional 1 min centrifugation after the last washing step to completely remove residual ethanol. cDNA was synthesized using the Promega GoScript Reverse Transcription Mix, Oligo Protocol. In this procedure, 500–1,000 ng of RNA were converted into oligo(dT)-primed first-strand cDNA. iTaq Universal SYBR Green Supermix was used according to the manufacturer's protocol for RT–qPCR reaction in a Quant Studio 7 Flex (Applied Biosystem by Life Technologies). RT–qPCR primers can be found in Supplementary Table 9. The delta-$C_t$ method in Quant Studio Real-Time PCR software (v1.3) was used for analysis and to calculate fold changes based on $C_t$ values.

## Western blot

Keratinocytes were cultured as described above, washed with PBS and lysed in protein sample buffer (100 mM Tris-HCl pH 6.8, 4% SDS, 20% glycerol, and 0.2 M DTT). The lysate was heated for 5–10 min to 95 °C and vortexed briefly to shear genomic DNA. Isolated proteins were stored at −20 °C for later use or loaded directly (10–20 µg). Proteins were separated by 4–12% Bis-Tris SDS–PAGE electrophoresis (at 180 V, 300 W, 55 min) and transferred onto a nitrocellulose membrane (GE Healthcare). Membranes were blocked in 5% BSA (Sigma A3059-100G) in TBS-Tween (0.1%) for 1 h at room temperature, incubated with primary antibodies in 3% BSA at 4 °C overnight and washed with TBS-Tween. The secondary antibodies were added at room temperature in 3% BSA for 2 h. Western blots were developed with freshly mixed ECL solutions (GE Healthcare). The following antibodies were used: anti-TNF (Cell Signaling Technology 11948), anti-vinculin (Abcam, ab129002). Secondary Goat Anti-Rabbit IgG HRP Linked Antibody (1:1,000 dilution, Cell Signaling Technology 7074S).

## Scratch assay

Cells were seeded in an ibidi Culture-Insert 2-well (81176) according to the manufacturer's recommendations. This insert creates a clean cell-free gap in a 6- or 12-well plate (Thermo Scientific, 150239). The assay was performed according to the manufacturer's protocol (ibidi). In brief, once confluency was reached, the insert was removed, and detached cells were washed away with PBS. New medium (with or without 10 ng ml$^{-1}$ recombinant TNF (R&D Systems 410-MT) was added, and the wells were imaged every hour for 22 h in an incubation chamber at 37 °C and 5% $CO_2$. The images were taken with a Zeiss Axio Observer controlled via the ZEN software. The data analysis was performed with an ImageJ plugin[67] (version 1.54 h) and default settings.

## TNF and TNF module overexpression constructs

Full-length mouse *Tnf* was amplified by PCR from GFP-TNF-alpha plasmid (Addgene #28089) via Phusion PCR according to the manufacturer's protocol with the following primer: forward: 5′-agcctgctgaagcaggcc ggcgacgtggaggagaacccccggccccatgagcacagaaagcatgatccgcgacgt-3′, reverse: 5′-ATATaacgcgtCTAcagagcaatgactccaaag′. Primers created a 5′-P2A-site overhang with the opposing site added to the 3′ end of mCherry. mCherry-P2A-TNF was then ligated into the CROP-seq vector via Pfl23 and MluI restriction sites. Expression was tested with quantitative PCR, western blot and immunofluorescence (see Fig. 5).

Full-length cDNA for *Jun, Ccn1* and *Fos* was Phusion PCR amplified from Addgene plasmids #47443, #12519 and #193089 with the following primers: *Fos* fw (5′-ATGGATCCGGGgccaccaacttcagcctgctg aagcaggccggcgacgtggaggagaacccccggccccATGATGTTCTCGGGTT-3′), *Fos* rv (5′-taacgcgtTCACAGGGCCAGCAGCGTG-3′); *Jun* fw (GGATCCG GGgccaccaacttcagcctgctgaagcaggccggcgacgtggaggagaacccccggcccc atgactgcaaagatggaaac-3′), *Jun* rv (5′-ataacgcgttcaaaacgtttgcaact-3′); *Ccn1* fw (5′-ataGGATCCGGGgccaccaacttcagcctgctgaagcaggccgg cgacgtggaggagaacccccggccccATGAGCTCCAGCACCTTC -3′), *Ccn1* rv (5′-taaacgcgtTTAGTCCCTGAACTTGTGGATGTC-3′).

PCR amplicons were ligated via BamHI and MluI restriction site overhangs into the same EF1-mCherry-2A expression vector as TNF.

## MTT assay

Proliferation rates for different treatments and overexpression constructs were tested using the MTT Cell Growth Assay Kit (Merck CT01). In short, 10,000 keratinocytes were seeded per 96-well of a flat-bottom tissue culture plate and incubated in 100 µl regular E-low medium for 24 h at 37 °C, 5% $CO_2$ before adding 10 µl of freshly prepared MTT solution for four hours. The resulting formazan crystals were dissolved by adding 100 µl isopropanol with 0.04 N HCl to each well with vigorous pipetting. Absorbance at 570 nm with a reference wavelength of 630 nm was measured directly on a Tecan Infinite M1000Pro plate reader.

## CRISPR sgRNA efficiency

Eleven sgRNAs representing top, middle and bottom enrichment cohorts were selected and sgRNA efficiency was assessed by infecting B6.Cas9 keratinocytes. Lentivirus was derived previously with low-titre lentiviral preparations of the CROP-Seq-Puro vector and single guides. Cells were harvested after 10 days of puromycin selection (1 µg ml$^{-1}$, Gibco A11138-03). Genomic DNA was extracted using the DNeasy Blood and Tissue Kit (QIAGEN 69504), followed by Taq (NEB M0273) PCR following the manufacturer's instructions. Amplicons were 800–1,000-bp-long and amplified by primers, designed with primer 3 listed in Supplementary Table 1. T7 endonuclease assay was performed as described in the ALT-R genomic editing detection kit (IDT). Samples were quantified on a Bioanalyzer 2100 with a DNA high-sensitivity chip. Primers for amplification of genomic target regions of guides can be found in Supplementary Table 1.

## Visium spatial gene expression

The workflow of Mirzazadeh et al.[68] for RNA-rescue spatial transcriptomics was followed to overcome low RNA integrity number score. A microarray of fresh-frozen DMBA/TPA-induced squamous cell carcinoma samples was cryo-sectioned with 10 µm thickness (Leica, CM3050S) and placed on the capture area of a spatial gene-expression slide (10X Genomics, 1000188). Samples were stored in −80 °C before processing. The samples were fixed with 4% PFA and H&E staining was performed. The spatial libraries were then generated from the probe hybridization step (10X Genomics, 1000365) according to

Visium Spatial Gene Expression Reagent Kits for FFPE (10X Genomics, User Guide CG000407 rev D, 1000361). The resulting libraries were sequenced by the Genomics Facility Basel. The sequencing was performed in a paired-end manner with dual indexing (10X Genomics, 1000251) on a NovaSeq 6000 (Illumina) with a S4 or SP (PE 51) flow cell. S4 libraries were then sequenced with the following cycle settings: Read 1 101 cycles, i7 10 cycles, i5 10 cycles, Read 2 101 cycles.

### Visium spatial expression data processing

Sequenced libraries were processed using Space Ranger software (version 2.0.1, 10X Genomics). Reads were aligned with the 'spaceranger count' pipeline to the pre-built mouse reference genome provided by 10X Genomics version 2020-A (comprising the STAR-indexed mm10 mouse genome and the GTF gene annotation from GENCODE version M23 (Ensembl 98)), and the probe set reference CSV file (Visium Mouse Transcriptome Probe Set v1.0) was provided for filtering of valid genes. The resulting filtered count matrix of features (that is, genes) per spot barcodes were employed for the downstream analysis. Proportions of cell-type mixtures in each Visium spot were deconvoluted with the robust cell-type decomposition (RCTD)[69] method from the spacexr package (version 2.2.1; https://github.com/dmcable/spacexr), leveraging as single-cell reference the union of mCherry-positive (Fig. 4c) and mCherry-negative (Extended Data Fig. 6c) tumour cells (merged dataset in Extended Data Fig. 7e). Seurat R package (version 4.9.9.9041) was used to plot these cell-type deconvolution results. Spatial co-occurrence of cell types was analysed with the ISCHIA package[40] (version 1.0.0.0), and resulting probabilities for positive ($P_{gt}$) or negative ($P_{lt}$) co-occurrence were plotted in −log10 scale.

### Bioinformatic processing of the scRNA-seq data

The raw sequencing data comprising the BCL files were demultiplexed using the Illumina bcl2fastq program v2.20 with default options, allowing one mismatch of the sample barcode sequences. The resulting fastq files were processed by the BD Rhapsody Pipeline version 1.9.1 hosted at the Seven Bridge's cloud platform (https://www.sevenbridges.com/). A custom mouse reference genome based on the Gencode version 25 was utilized and combined with our 500 sgRNA sequences. Since the BD pipeline uses STAR aligner[70] in the backend, the custom STAR reference genome was generated by the genomeGenerate command in STAR. The Refined Putative Cell Calling option was disabled while running the pipeline. The output unique molecular identifier (UMI) counts data corrected by the BD Genomics RSEC (Recursive Substitution Error Correction) method were used for further downstream analysis.

### Processing of the PCR sgRNA dial-out data

The PCR dial-out data was processed by a custom in-house Python script. In brief, the script extracted three 9-nucleotide (nt) long cell barcodes out of possible 97 barcodes at specific locations from the Read 1, allowing one mismatch per barcode (Hamming distance of 1). It also obtained 8-nt UMI sequences from the Read 1, as provided in the BD manual. For valid barcodes, it counted the presence of sgRNAs in Read 2 by an exact match of the 20-nt sgRNA sequence. Next, the UMI counts per cell were deduplicated and sgRNA UMI counts per cell were obtained. If multiple sgRNAs were detected in one cell, the cell was only assigned to a specific sgRNA if the sgRNA UMI count was higher than the quantile 0.99.

### Downstream processing of the scRNA-seq data

To remove the doublets, the Scrublet[71] Python package was utilized. Doublets were removed from each dataset separately with initial filtering of genes expressed in minimum 5 cells and expected doublet rate of 20% as detected by the BD Rhapsody. Next, 7 and 8 scRNA-seq datasets for P4 and P60, respectively, were merged in Seurat 4[72]. Cells with detected sgRNAs and that were not doublets were selected for

further analysis. Cells with UMI counts >500, UMI counts lower than quantile 0.99 of total UMI counts (to remove outliers) and cells <20% mitochondrial genes were filtered. Afterwards, the data were processed by the standard Seurat pipeline of normalizing the data with NormalizeData function, which scales the counts to 10,000 per cell followed by natural log transform using log1p. Two thousand variable genes were detected by the FindVariableGenes function. The normalized data was scaled by the ScaleData function followed by PCA analysis with the RunPCA function. The contribution of each principal component in explaining the variance of the data with the ElbowPlot function and selected top principal components for each dataset of P4, P60 and tumours. The selected principal components were used for finding nearest neighbours with the FindNeighbors function. The clustering of the data with FindClusters function with modularity optimization by Louvain algorithm. UMAP dimensional reduction technique was used to visualize the data in two dimensions. No batch effect in the P4 and P60 datasets was observed in the UMAP projections. Hence no batch effect correction was performed. The marker genes for each cluster were obtained by the FindAllMarkers function and the clusters were annotated with cell types based on the expression of known marker genes. The differential expression analysis of the scRNA-seq data for each perturbation was carried out using the MAST package[73] and Wilcoxon rank-sum test, which yielded comparable results. The kernel density estimates for gene expression were inferred by the Nebulosa package.

### Analysis of average expression and perturbation-perturbation matrix

To obtain an overview of the transcriptional phenotype of each perturbation, the average gene expression of cells for each perturbation in basal cells and all cells of P60 was calculated. Next the top 2,000 most variable genes in those perturbations were selected and heat maps of correlation matrices were computed using the pheatmap package with ward.D2 method for clustering. Moreover, we utilized the preserve_neighbors function from PyMDE package (pymde.org) for minimum-distortion embedding of the data in two dimensions.

### sgRNA counts in tumour data

One-hundred and sixty eight individual tumours were sequenced to quantify sgRNA representation in each tumour. The raw sequencing data were demultiplexed with bcl2fastq function with corresponding i5 and i7 indexing barcodes. The number of sgRNAs in the sample was counted by the count command from the MAGECK[74] package. The resulting sgRNA representations in tumours were used for total representation and selection scores in tumours.

### Immune cell depletion

CD4+ or CD8+ T cells were depleted by intraperitoneal injection of a cocktail of monoclonal rat anti-mouse CD4 (YTS 191.1, Hoelzel Diagnostika LEIN-C3210) and rat anti-mouse CD8 (YTS169.4 Hoelzel Diagnostika LEIN-C2850) neutralizing antibodies at 10 mg per kg body weight. Antibodies were injected every two weeks from P4 to P60. Macrophages were depleted by intraperitoneally injecting rat anti-mouse CD115 (CSF-1R, Hoelzel Diagnostika LEIN-C2268) at an initial dose of 0.5 mg followed by 25 mg per kg body weight intraperitoneally twice per week from P4 to P60. Control mice received rat IgG isotype control (Hoelzel Diagnostika, LEIN-I-1177). We used 6 mice for T cell depletion, 7 mice for macrophage depletion and 7 control IgG mice. At P60, the epidermis was prepared as described above and split into two parts (anterior back skin and posterior back skin) for separate amplicon sequencing runs. Following genomic DNA isolation, PCR amplification of sgRNA cassettes was carried out with the same protocol as the sgRNA amplification protocol from tumours. Amplicons were sequenced on a Novaseq 6000 SP flow cell with 10% PhiX, dual indexing i5, 8 bp; i7, 8 bp; and R1 for 64 cycles. The number

of sgRNAs in the fastq files was counted by MAGECK. DESeq2 was utilized for differential enrichment analysis of the sgRNAs at gene level[65]. The count data for the samples were normalized using the median of ratios method using the 50 control sgRNAs and the dispersion across samples was estimated. A negative binomial generalized linear model was fitted for each guide to detect differentially enriched or depleted guides. The P values obtained by the Wald test were corrected by the Benjamini–Hochberg multiple testing procedure.

## WGCNA analysis

For the WGCNA[24] analysis, the dataset was downsampled to a maximum of 500 cells per perturbation using the subset function with downsample = 500 option in Seurat. Next, the data were normalized and 2,000 most variable genes were selected with NormalizeData and FindVariableFeatures functions. The power calculation was performed by the pickSoftThreshold function from the WGCNA package for powers 1 to 30. Next, we ran WGCNA analysis with blockwiseModules function with signed network, minimum module size of 10 and mergeCutHeight value 0.15 and correlation function bicor for robust correlations using the power estimate from the previous step. The cells with control sgRNA were not included for WGCNA analysis unless specified otherwise. WGCNA analysis was carried out for each cluster separately. To calculate the perturbation score, moduleEigengenes function was run to obtain the module eigengenes (first principal component) of the given module. Next, the values of the first principal component were normalized using the scale function in R. Next, linear modelling was performed in R to obtain perturbation score for each perturbation over control with the formula: normalized gene score ~ perturbation. The P values, standard error and effect sizes were extracted from the linear model for plotting. This approach was similar to the one described in Jin et al.[25]. The P values of differentially expressed module genes over control by the Fisher's method were combined using the metap R package with the average of log fold change to compute the average perturbation effect.

## Cell–cell signalling analysis

In order to decipher cell-to-cell signalling between each cluster, the Cellchat[33] package with truncated mean approach with minimum ten cells for inferring signalling was used.

## Integration of scRNA-seq data

The integration of P60 cluster 0 and tumour mCherry-positive clusters 0 and 7 was performed using Seurat IntegrateData function. First, each dataset was normalized and 2,000 variable genes were obtained. Next SelectIntegrationFeatures was used to select features that are repeatedly variable across datasets for integration. Next, the datasets were scaled and PCA was performed using the common features. After that, Seurat FindIntegrationAnchors function was run with the RPCA algorithm utilizing 30 principal components followed by IntegrateData to integrate the datasets. Results were compared with the Harmony[75] algorithm, which yielded similar integration.

## SEACells analysis

To identify meta cell states in the scRNA-seq data, the SEACells approach was employed, which identifies a group of cells defined as metacells to uncover cell states. First, the P60 EpSC dataset was downsampled to a maximum of 200 cells per sgRNA to obtain a similar number of metacells for each perturbation. Around 50 cells were chosen to define one meta cell (SEACell) and used the PCA to build the kernel. The SEACells function was run with the option n_waypoint_eigs= no of SeaCell +1 and convergence_epsilon =1e-5 for each perturbation separately resulting in 1–4 metacells per perturbation. The SEACells model was fitted with a maximum of 100 iterations. Furthermore, the average normalized expression was calculated for each meta cell by the AverageExpression function in Seurat for plotting the gene-expression heat maps.

## General data analysis

The analysis was carried out using in-house Python 3.9 and R 4.1 scripts. Data wrangling was done with the pandas library in Python and tidyverse library in R. Heat maps were created using the ComplexHeatmap and pheatmap packages. Other plots were made using the ggplot2 library in R and seaborn library in Python.

## Pathway enrichment

The overrepresentation analysis for the pathway enrichment was performed by EnrichR[31] (https://maayanlab.cloud/Enrichr), which is based on Fisher's exact test, and with custom R scripts utilizing the enrichR library on MSigDB Hallmark 2020 and other gene sets. A cut-off of FDR < 0.05 was set to select differentially expressed genes. Pathway enrichment analysis was carried out using the GSEA[31] pre-ranked method using the GSEApy[76] (https://github.com/zqfang/GSEApy), which enables the analysis of up- and downregulated genes simultaneously. 10,000 permutations were performed at a FDR cut-off value of 0.05 to determine enriched gene sets.

## Survival analysis

TCGA data was obtained from UCSC Xena Toil[77] (http://xena.ucsc.edu). Survival analysis was conducted for HNSCC and other TCGA tumour types in R utilizing the survival and survminer library. The Cox Proportional Hazards regression model was used to fit the gene-expression data to survival to obtain the Hazard Ratio. Kaplan–Meier analysis was performed on samples with lower and upper tertiles (top 1/3 versus bottom 1/3 mRNA) of gene expression (in transcripts per million (TPM)) and the P values were computed by log-rank test. To obtain the association of survival with the TSK signature genes, the average expression levels of *MMP9*, *MMP10*, *PTHLH*, *FEZ1*, *IL24*, *KCNMA1*, *INHBA*, *MAGEA4*, *NT5E*, *LAMC2* and *SLITRK6* were computed. Kaplan–Meier analysis was performed with lower and upper tertiles of average expression of these genes.

## Statistics and reproducibility

Whole-mount stainings shown in Figs. 1c and 3e are representative images from three fields of view from two mice each. Immunofluorescence as shown in Fig. 4 are representative images from six fields of view from four different tumours. Figure 5j–n are representative images from five fields of view from two mice each.

## Reporting summary

Further information on research design is available in the Nature Portfolio Reporting Summary linked to this article.

## Data availability

The complete single-cell RNA sequencing and CROP-seq data for P4, P60 and tumours, P60 *Tnfrsf1a*-knockout and immune cell depletion data are available at the Gene Expression Omnibus (GEO) under accession GSE235325.

## Code availability

The main scripts used for the analyses in this paper and the R markdown for the main figures are available on the Sendoel laboratoy GitHub repository: https://github.com/sendoellab/single-cell_CRISPR.

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

**Acknowledgements** The authors thank F. Quiroz, M. Jovanovic, R. Barricarte, and members of the Sendoel and Moor laboratories for critical input on the manuscript; M. van den Broek and M. Greter for advice with the immune-depletion experiments; C. Aquino and the FGCZ for sequencing and H.-U. Schwarz for assistance. The project was supported by the European Research Council (ERC) under the European Union's Horizon 2020 research and innovation programme (grant agreement no. 759006), by an SNSF Professorship grant (grant no. 176825), by the Swiss Cancer Research foundation (KFS-5023-02-2020-R), by the Helmut Horten Foundation and the National Center of Competence in Research (NCCR) on RNA and Disease funded by the SNSF (grant number 205601). Schematics in Figs. 1a,b, 3b,f and 5a,o, Extended Data Figs. 1d and 6a and Supplementary Figs. 7a and 9b were partly created using BioRender.com.

**Author contributions** P.F.R. conducted the experiments and collected the data. P.F.R., U.G., S.B.S. and A.S. performed data analysis and interpretation. A.K. and S.J.E. performed whole-mount stainings and image analysis. M.O. carried out in utero lentiviral injections and mouse handling, assisted by M.B., K.H. and C.D. M.F. and F.V.-F. assisted with lentiviral preparations and single-cell capture experiments. J.A.K., M.L. and M.P. performed RNA-rescue spatial transcriptomics on frozen tumour samples. P.F.R., A.S. and A.E.M. conceived the project. A.S. and A.E.M. supervised the project. X.F. provided input on the manuscript. P.F.R. and A.S. wrote the manuscript. All authors reviewed and approved the final version.

**Competing interests** The authors declare no competing interests.

**Additional information**
**Correspondence and requests for materials** should be addressed to Andreas E. Moor or Ataman Sendoel.

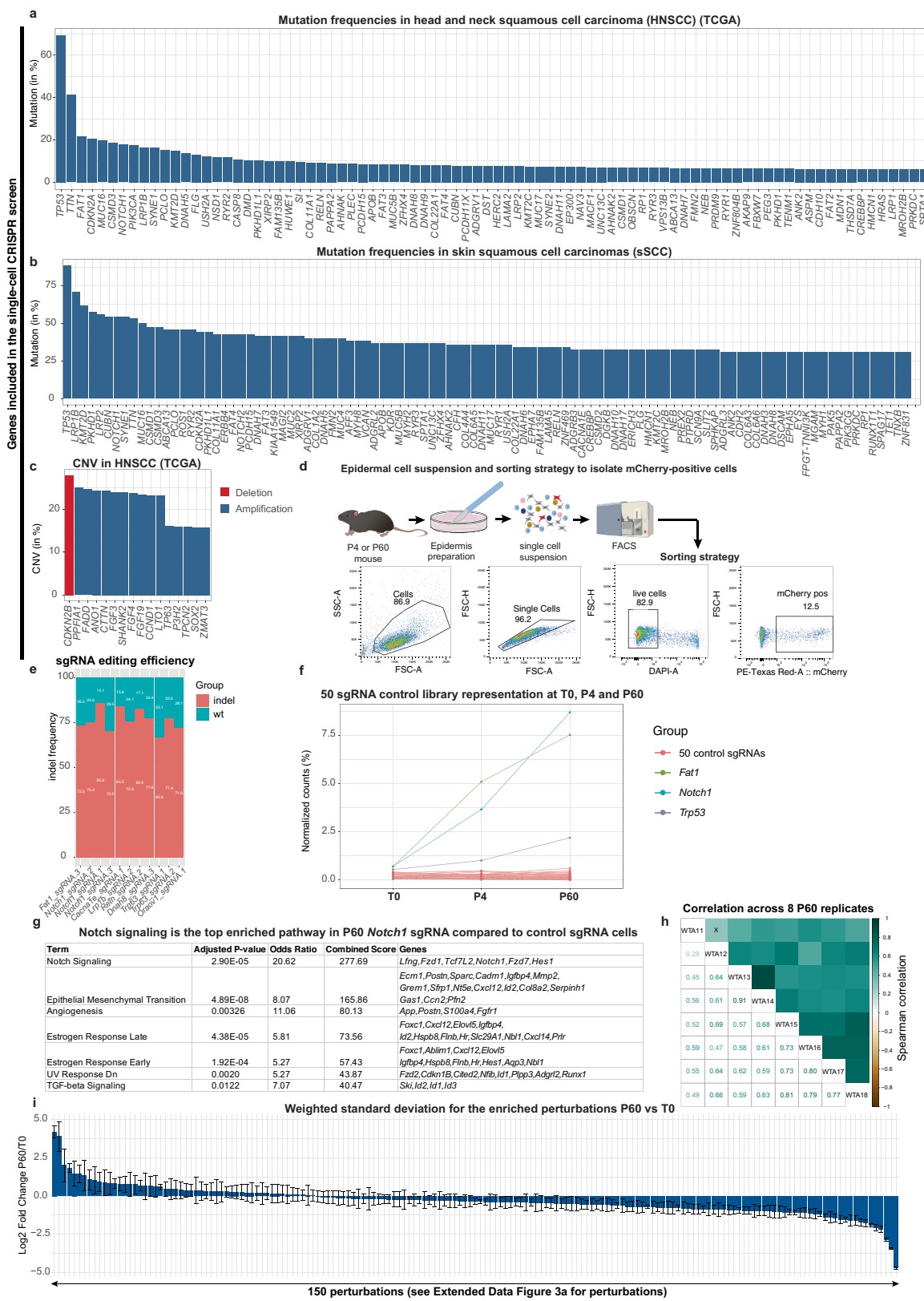

**Extended Data Fig. 1** | See next page for caption.

**Extended Data Fig. 1 | Establishing an in vivo single-cell CRISPR screen.**
**a-b**, Perturbations included in the single-cell CRISPR library. Genes were ranked based on their mutation frequency in head and neck squamous cell carcinoma (HNSCC) and skin squamous cell carcinoma patients available from the cBioPortal[51]. **c**, CNVs included in the single-cell CRISPR library. Genes were ranked by their copy number variation frequency (CNV) in head and neck squamous cell carcinoma (HNSCC)[51]. While *CDKN2B* is deleted in HNSCC, the other 15 CNVs represent amplifications. **d**, Experimental workflow to prepare single-cell suspensions from back skin followed by fluorescence-assisted cell sorting (FACS) with representative gating strategy. An example preparation with 12.5% mCherry-positive epidermal cells is depicted. Median infection rate at P4, 2.4%. Median infection rate at P60, 16.3%. **e**, sgRNA editing efficiency represented as the percentage of indel frequency. *Cas9*-positive keratinocytes were infected with the CROP-seq vector containing single sgRNAs. Indel frequency was measured by the Surveyor nuclease assay and corresponding DNA fragments were quantified using the Bioanalyzer. 11 sgRNAs representing top, middle and bottom enriched guides were assessed. **f**, Stable propagation of the control library. Non-targeting control sgRNA representation compared to the enriched *Notch1, Fat1* and *p53* (3 sgRNAs each) across the T0, P4 and P60 time points. The graph displays the percentage of cells containing the sgRNA relative to the total number of cells at each time point. **g**, Gene set enrichment analysis using Enrichr of P60 *Notch1* sgRNA cells compared to control sgRNA cells reveals *Notch* signaling as the top enriched pathway (combined score), validating the bioinformatics sgRNA identification pipeline. **h**, P60 sgRNA enrichment and depletion correlate well across the 8 replicates, as shown by the Spearman correlation analysis of the top/bottom 20 perturbations in the P60 skin. WTA11-18 refer to the different replicates and "X" indicates FDR > 0.05. **i**, The enrichment and depletion patterns and weighted standard deviation of total sgRNAs across the 8 replicates in the mouse P60 skin (related to Fig. 2a). The waterfall plot displays the enrichment and depletion of 150 perturbations at P60, represented as the log2 fold change between total cell numbers at P60 compared to the same ratio in the respective library T0. The standard deviation is calculated as weighted approach, where the weight corresponds to the number of cells in each replicate. The specific perturbation details can be found in Extended Data Fig. 3a.

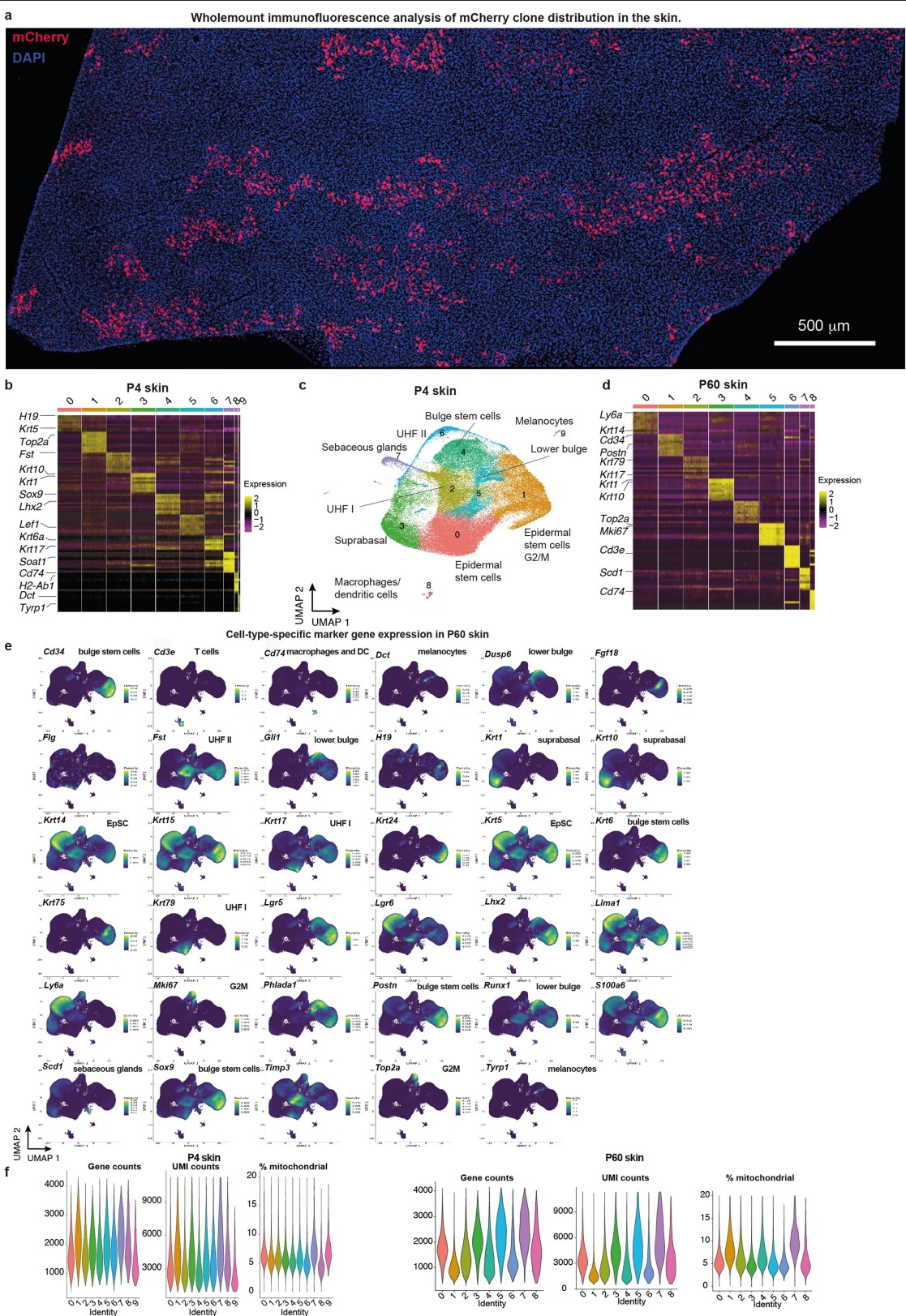

**Extended Data Fig. 2 | Marker gene expression in P4 and P60 skin. a**, Whole-mount immunofluorescence staining of mCherry clones in the E16.5 skin, injected with the library of 500 sgRNAs. **b**, Heatmap displaying the expression profiles of the top marker genes across individual mCherry-positive cells in P4 animals. **c**, Uniform manifold approximation and projection (UMAP) of the ten major cell populations identified by the in vivo single-cell CRISPR strategy of P4 animals (n = 120,077 cells, from 58 animals, following filtering and sgRNA annotation). UHF, upper hair follicle. **d**, Heatmap displaying the expression

profiles of the top marker genes across individual mCherry-positive cells in P60 animals. **e**, Visualization of cell-type specific marker gene expressions by Nebulosa density plots, highlighting their specific localization in the P60 UMAP. EpSCs, epidermal stem cells. UHF, upper hair follicle. DC, dendritic cell. **f**, Characterization of the P4 and P60 single-cell RNA sequencing data sets, displaying gene counts, unique molecular identifier (UMI) counts and % mitochondrial genes as violin plots in each cluster.

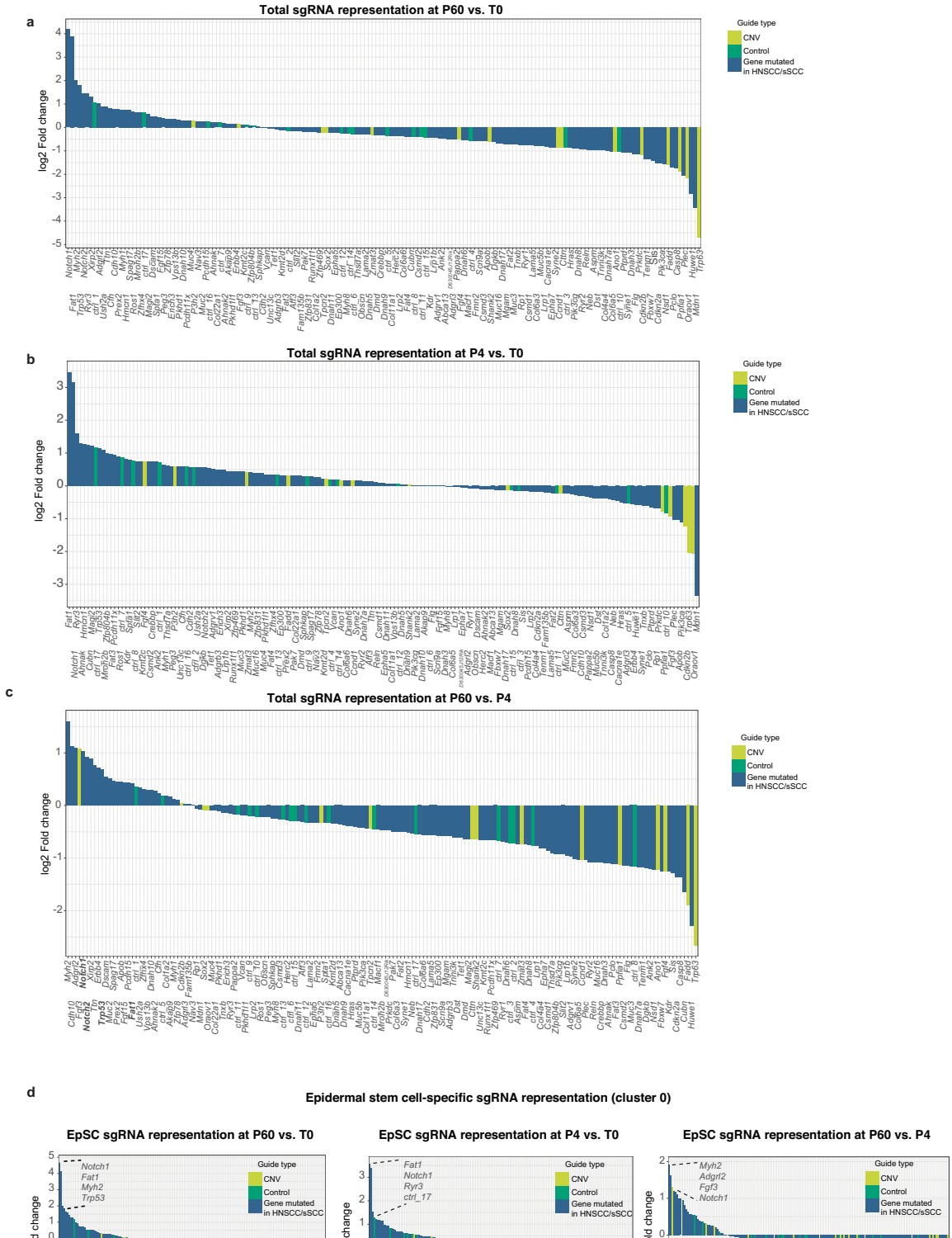

**Extended Data Fig. 3 | Waterfall plots showing enrichment and depletion of the 150 perturbations. a-c**, The waterfall plots display the enrichment and depletion of 150 perturbations at P60 compared to the library T0 (**a**), P4 compared to the library T0 (**b**), or P60 compared to P4 (**c**). Yellow indicates copy number variation genes, green represents control sgRNAs, and blue signifies genes mutated in head and neck (HNSCC) and skin squamous cell carcinoma (sSCC) patients. Data represent the weighted average of seven P4 and eight P60 replicates (7/8 single-cell RNA sequencing runs, 58 P4 animals, 10 P60 animals).

**d**, Epidermal stem cell (EpSC)-specific enrichments. The waterfall plot displays the enrichment and depletion of 150 perturbations, represented as the log2 fold change between EpSC numbers (cluster 0) compared to T0 or P4 representation. Yellow indicates copy number variation genes, green represents control sgRNAs, and blue signifies genes mutated in head and neck (HNSCC) and skin squamous cell carcinoma (sSCC) patients. Data represent the weighted average of seven P4 and eight P60 replicates (7/8 independent single-cell RNA sequencing runs, 58 P4 animals, 10 P60 animals).

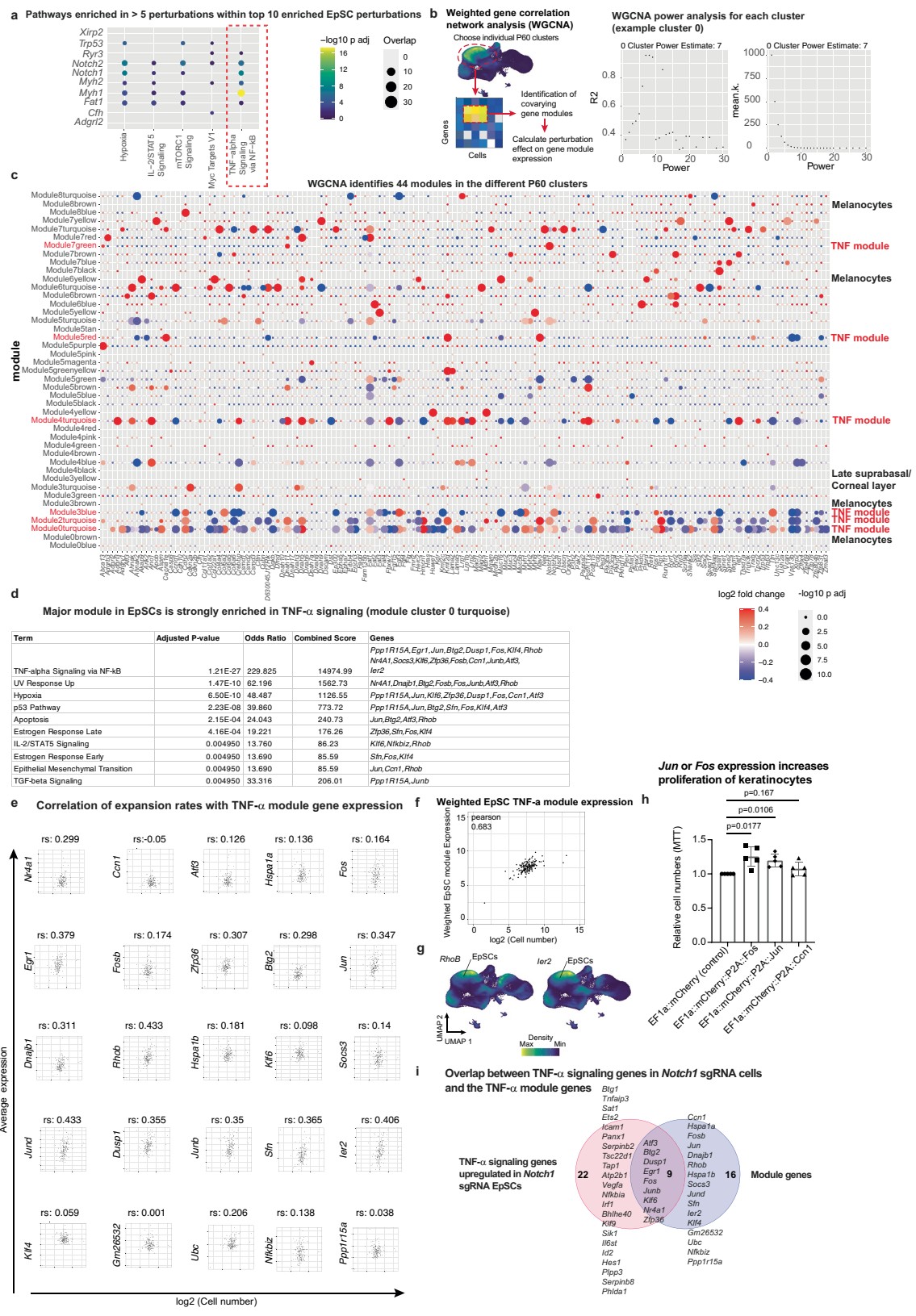

**a** Pathways enriched in > 5 perturbations within top 10 enriched EpSC perturbations

**b** Weighted gene correlation network analysis (WGCNA)

WGCNA power analysis for each cluster (example cluster 0)

**c** WGCNA identifies 44 modules in the different P60 clusters

**d** Major module in EpSCs is strongly enriched in TNF-α signaling (module cluster 0 turquoise)

| Term | Adjusted P-value | Odds Ratio | Combined Score | Genes |
|---|---|---|---|---|
| TNF-alpha Signaling via NF-kB | 1.21E-27 | 229.825 | 14974.99 | Ppp1R15A,Egr1,Jun,Btg2,Dusp1,Fos,Klf4,Rhob Nr4A1,Socs3,Klf6,Zfp36,Fosb,Ccn1,Junb,Atf3, Ier2 |
| UV Response Up | 1.47E-10 | 62.196 | 1562.73 | Nr4A1,Dnajb1,Btg2,Fosb,Fos,Junb,Atf3,Rhob |
| Hypoxia | 6.50E-10 | 48.487 | 1126.55 | Ppp1R15A,Jun,Klf6,Zfp36,Dusp1,Fos,Ccn1,Atf3 |
| p53 Pathway | 2.23E-08 | 39.860 | 773.72 | Ppp1R15A,Jun,Btg2,Sfn,Fos,Klf4,Atf3 |
| Apoptosis | 2.15E-04 | 24.043 | 240.73 | Btg2,Atf3,Rhob |
| Estrogen Response Late | 4.16E-04 | 19.221 | 176.26 | Zfp36,Sfn,Fos,Klf4 |
| IL-2/STAT5 Signaling | 0.004950 | 13.760 | 86.23 | Klf6,Nfkbiz,Rhob |
| Estrogen Response Early | 0.004950 | 13.690 | 85.59 | Sfn,Fos,Klf4 |
| Epithelial Mesenchymal Transition | 0.004950 | 13.690 | 85.59 | Jun,Ccn1,Rhob |
| TGF-beta Signaling | 0.004950 | 33.316 | 206.01 | Ppp1R15A,Junb |

**e** Correlation of expansion rates with TNF-α module gene expression

**f** Weighted EpSC TNF-a module expression

**g**

**h** *Jun* or *Fos* expression increases proliferation of keratinocytes

**i** Overlap between TNF-α signaling genes in *Notch1* sgRNA cells and the TNF-α module genes

**Extended Data Fig. 4** | See next page for caption.

**Extended Data Fig. 4 | WGCNA analysis of the P60 clusters. a**, TNF signaling is the overall most strongly enriched pathway in the top 10 expanded perturbations in P60 epidermal stem cells. Differential gene expression was computed for each of the top 10 enriched perturbations in EpSCs and pathway enrichment was performed for the significantly upregulated genes. The pathway enrichment for upregulated and downregulated genes is shown in Supplementary Table 3. Dot plot shows the pathways that are significantly enriched in at least 5 perturbations. Overlap, total number of genes overlapping with the pathway. **b**, Left panel, schematic of the weighted gene correlation network analysis (WGCNA) pipeline to identify covarying gene modules in P60 EpSCs. For each cluster's weighted gene correlation network analysis (WGCNA), a power analysis was performed to find the power for the scale-free topology and then used for all 150 perturbations to unveil gene expression changes that are common to multiple gene perturbations. An example for cluster 0, EpSCs, and its power analysis is displayed. **c**, Analysis of gene modules in the different cell populations of the P60 skin. The panel depicts 44 gene modules identified within nine distinct cell clusters at P60. Using Weighted Gene Correlation Network Analysis (WGCNA), we explored correlation patterns among genes to identify modules (groups of genes) that co-vary within these clusters. The dot plot visualizes the average log2 fold change for genes within each WGCNA module relative to control sgRNA cells, offering insights into the perturbation effects on different cell types. A notable finding is the identification of a significant TNF gene module, which is consistently present across five different cell clusters in P60 skin, comprising largely the same set of genes. Additionally, the plot highlights modules corresponding to specific cell types, such as

melanocytes (characterized by genes like *Sox10, Dct, Tyrp1* and *Kit*) or cells of the stratum corneum within the suprabasal cluster. The color of each dot indicates the log2 fold change, while the size of the dot represents the adjusted p value, calculated using the Benjamini-Hochberg procedure for controlling the false discovery rate. Detailed lists of genes in each module can be found in Supplementary Table 4. The nomenclature for modules follows a module + cluster number in P60 + color format, referencing the specific modules within each cluster. **d**, The major identified gene module in epidermal stem cells (EpSCs) is strongly enriched in TNF signaling, with 17 out of 25 genes overlapping with the pathway. Gene set enrichment analysis was performed using Enrichr. **e**, Spearman correlation (rs) of the expression of the 25 epidermal stem cell WGCNA module genes with clonal expansions of the 150 perturbations. P values indicate a two-tailed t-test. **f**, A linear model predicts clone size with average expression of 25 TNF signaling module genes in EpSCs, explaining a significant proportion of variance (p < 0.001, two-tailed t-test, Methods). **g**, Nebulosa density plots visualize expression of the two TNF module genes *RhoB* and *Ier2*. **h**, *Jun* and *Fos* expression increase proliferation rates in keratinocytes. Keratinocytes were infected with lentiviral constructs expressing a control, *Ccn1, Jun* or *Fos* construct. Proliferation was assessed by a MTT assay in 5 independent experiments. Error bars represent standard deviation of the mean. P values indicate a one-sided Welch's t-test. **i**, Overlap of TNF signaling genes differentially expressed in P60 *Notch1* sgRNA cells compared to the TNF module genes. Besides 9 genes overlapping with the TNF module, an additional 22 genes are differentially expressed in *Notch1* sgRNA compared to control sgRNA epidermal stem cells. Pathways, MSigDB Hallmark.

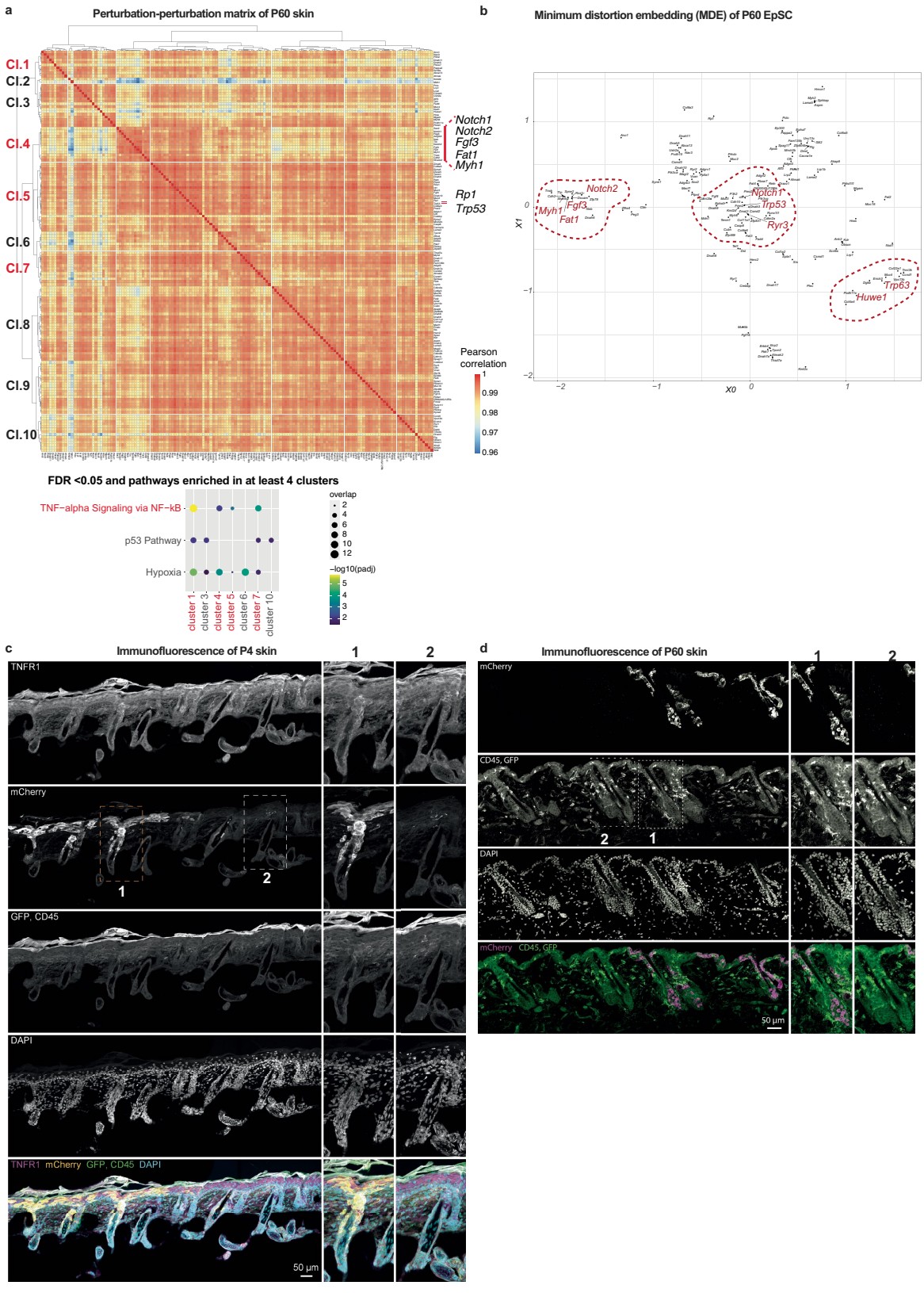

**Extended Data Fig. 5** | See next page for caption.

**Extended Data Fig. 5 | Clustering perturbations to condense expression programs. a**, Clustering of perturbations to condense expression programs and phenotypes in P60 skin. P60 perturbation-perturbation matrix based on gene expression shows clustering of enriched perturbations such as *Notch1, Notch2, Fat1* and *Myh1*. Lower panel shows pathways that are enriched in 4 or more clusters, suggesting that the differentially expressed genes in epidermal stem cells of cluster 1, 4 (including *Notch1, Notch2, Fat1* and *Myh1* perturbations), 5 and 7 are strongly enriched in TNF signaling. Included in this analysis were differentially expressed genes for each perturbation in the cluster compared to control sgRNA cells (FDR < 0.05) and the pathways enriched (FDR < 0.05) in at least 4 clusters. **b**, Minimum distortion embedding (MDE) was used for the visualization and dimensionality reduction of the perturbation effect of P60 epidermal stem cells (EpSCs). MDE depicts each perturbation as a dot. Some clusters are highlighted and consist of enriched and depleted perturbations in EpSCs. **c-d**, Immunofluorescence of mouse P4 and P60 back skin sections exhibit mCherry-positive expanded clones, TNFR1 expression in epidermal stem cells and the presence of CD45-positive immune cells in the epidermis. Insets show higher magnification of select areas. Scale bars, 50 μm.

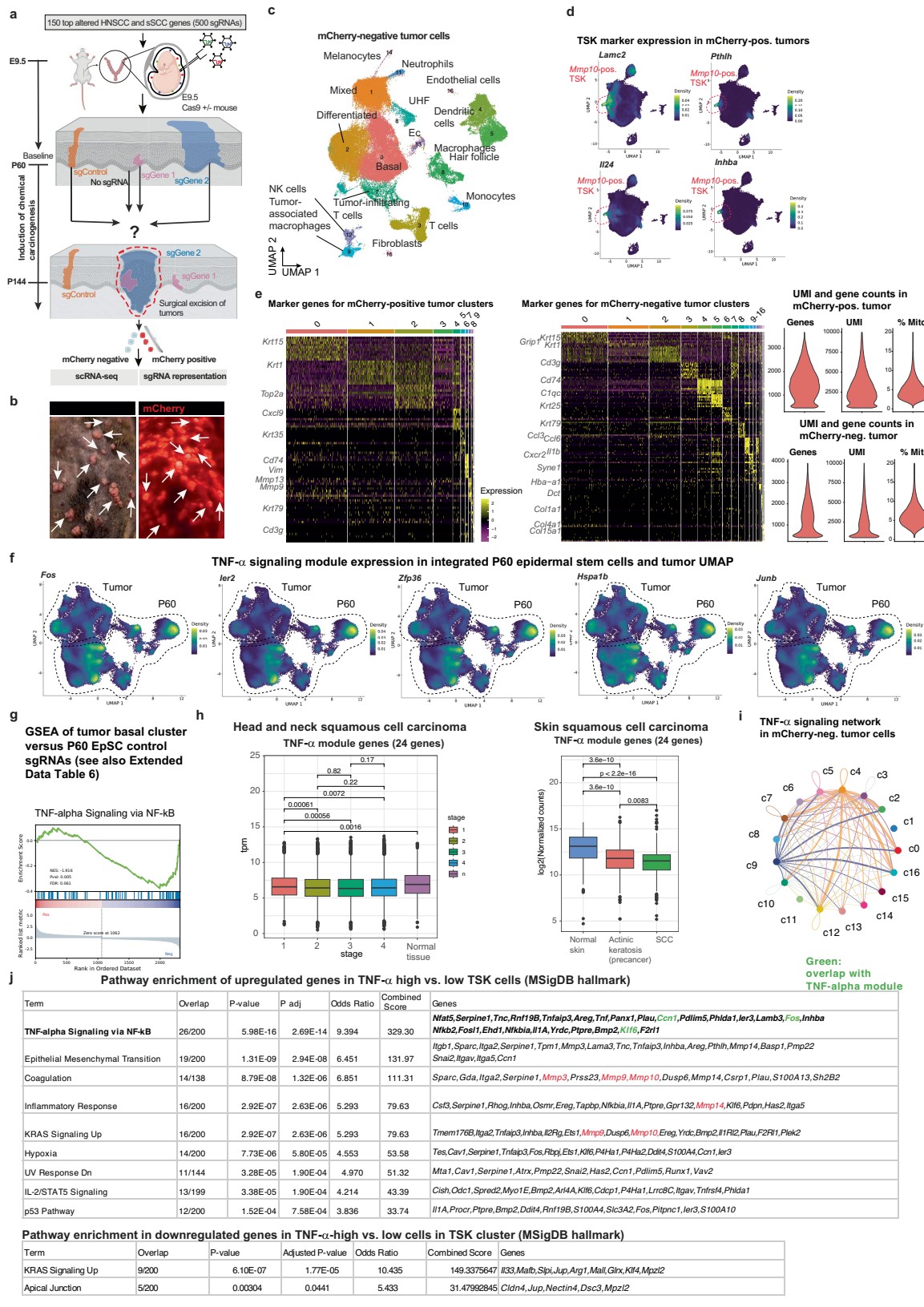

**Extended Data Fig. 6** | See next page for caption.

**Extended Data Fig. 6 | Single-cell RNA sequencing of mCherry-positive and negative tumors. a**, Schematic outline of the experimental strategy to test the role of clonal expansions in tumor initiation. The high-titer lentiviral library consisting of 500 sgRNAs is introduced into *Rosa26-Cas9* +/− E9.5 embryos via ultrasound-guided *in utero* microinjections. At P60, we induced chemical carcinogenesis by DMBA/TPA for 12 weeks and tumor single-cell suspensions were sorted for mCherry-positive and -negative cells by FACS. Single-cell RNA sequencing was performed on 30 mCherry-positive tumors (from 2 animals), and another 168 tumors (from 14 animals) were collected for sgRNA amplicon sequencing. **b**, Representative macroscopic image of the mouse back skin after 12 weeks of chemical carcinogenesis by DMBA/TPA treatment show mCherry-positive and mCherry-negative tumors. Left panel, normal light. Right panel, fluorescent light. Arrows indicate individual mCherry-positive tumors. **c**, UMAP representation of single-cell RNA sequencing data from mCherry-negative tumor cells (n = 79,821 cells, from 30 tumors, 2 animals). **d**, Visualization of tumor-specific keratinocyte (TSK)-specific marker gene expressions by Nebulosa density plots, highlighting their specific localization in the mCherry-positive tumor UMAP. **e**, Heatmap displaying the expression profiles of the top marker genes across individual mCherry-positive and negative tumor cells (left two panels). Right panels show the quality control of the tumor single-cell RNA sequencing data sets, displaying gene counts, unique molecular identifier (UMI) counts and % mitochondrial genes as violin plots in both mCherry-positive and mCherry-negative tumor cells. **f**, TNF signaling module genes are strongly downregulated in tumors. Visualization of select TNF signaling gene expressions by Nebulosa density plots, highlighting their specific localization in the integrated P60 EpSC-tumor UMAP. **g**, Gene Set Enrichment Analysis (GSEA) comparing tumor cluster 0 cells versus P60 epidermal stem cells reveals enrichment of E2F and G2M-related pathways, with TNF signaling identified as the top downregulated pathway. Right panel, enrichment plot for TNF signaling. **h**, TNF module gene expression is decreased in human skin and head and neck squamous cell carcinoma patients. The decrease in TNF module gene expression correlates with disease stage in head and neck squamous cell carcinoma patients (also see Supplementary Fig. 8). Box plots indicate the interquartile range with median drawn as line and Tukey-style whiskers. Sample size per stage: stage 1 = 48, stage 2 = 133, stage 3 = 96, stage 4 = 173, normal tissue = 44. P values indicate a two-tailed t-test. **i**, CellChat circle plot showing ligand-receptor interactions of the TNF signaling pathway across mCherry-negative tumor clusters. **j**, Gene set enrichment analysis comparing TNF-positive tumor-specific keratinocyte (TSK) versus TNF-negative TSKs (cluster 7, mCherry-positive tumors) uncovers TNF signaling and epithelial-mesenchymal transition (EMT) as top enriched pathways in TNF-expressing TSKs, suggesting an autocrine TNF cascade associated with invasive features in TSKs. Green, only 3 genes overlap with the TNF module, suggesting the presence of a distinct TNF gene program. Red, *Mmp* genes. Two-tailed Fisher's exact test was used to examine whether a gene set was significantly enriched.

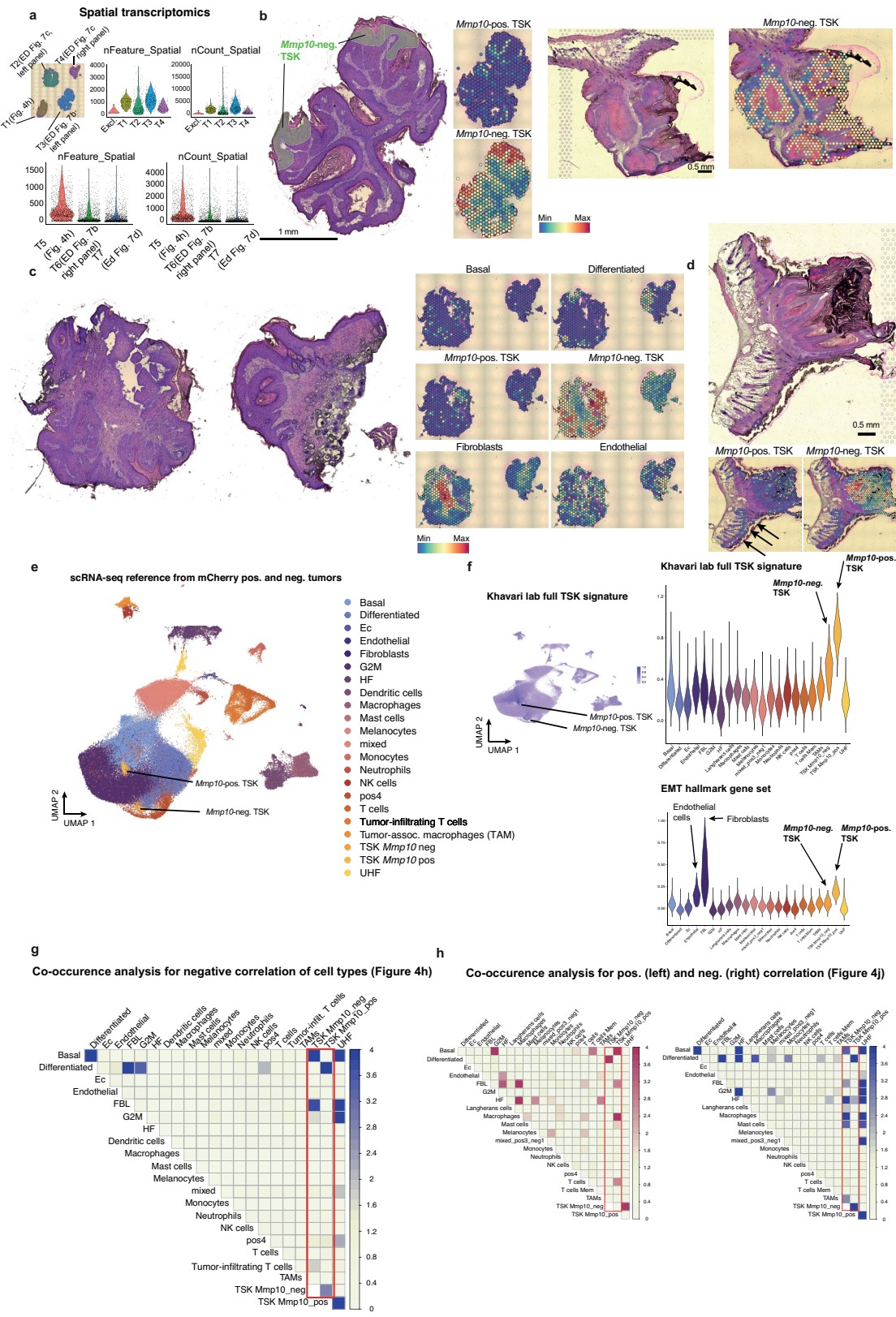

**Extended Data Fig. 7** | See next page for caption.

**Extended Data Fig. 7 | Spatial transcriptomics of tumors. a**, Characterization of the tumor spatial transcriptomics datasets. Violin plots displaying gene counts (nFeature) and unique molecular identifier (UMI) counts (nCount) of the 7 processed tumor sections. **b-d**, Hematoxylin and Eosin (H&E) stained images of five additional sections utilized for Visium spatial transcriptomics. A total of seven sections were processed to obtain spatial transcriptomics data. Spatial maps highlight the basal and differentiated tumor areas, as well as *Mmp10*-positive and negative tumor-specific keratinocytes (TSK), fibroblasts and endothelial cells. While *Mmp10*-negative TSKs localize to differentiated regions at the surface of the tumor, distant from the invasive front (b, d), *Mmp10*-positive TSKs reside at the invasive front of the tumor (d, arrows). **e**, UMAP of the single-cell reference employed for the cell-type deconvolution of the Visium spots (see Methods section). Obtained by merging the scRNAseq datasets of mCherry-positive and mCherry-negative tumors (datasets from Fig. 4c,d). Ec, erythrocytes. HF, hair follicle. Pos4, cluster 4 in mCherry-positive UMAP. UHF, upper hair follicle. **f**, *Mmp10*-positive TSKs more closely resemble the human TSKs identified by the Khavari lab[36] and show higher EMT hallmark gene set expression than *Mmp10*-negative TSKs. The score results for the TSK signature were obtained using Seurat's *AddModuleScore* function for the human TSK signature genes (extended marker list[36]). EMT score results were obtained using the hallmark gene set (Molecular Signature Database). **g**, Co-occurrence analysis demonstrating negative correlations between different cell types within the tumor in Fig. 4h. The probability (P(lt)) indicates the likelihood of the observed co-occurrence being less than the expected co-occurrence. Ec, erythrocytes. HF, hair follicle. Pos4, cluster 4 in mCherry-positive UMAP. FBL, fibroblasts. TAM, tumor-associated macrophages. UHF, upper hair follicle. **h**, Co-occurrence analysis of the tumor in Fig. 4j shows a positive correlation between the localization of *Mmp10*-positive TSKs, basal tumor cells, macrophages and fibroblasts, in line with *Mmp10*-positive TSKs residing in a fibrovascular niche. In contrast, *Mmp-10*-negative TSKs are positively correlated with differentiated tumor cells. P(gt), the probability of the observed co-occurrence to be greater than the expected co-occurrence. The probability (P(lt)) indicates the likelihood of the observed co-occurrence being less than the expected co-occurrence. HF, hair follicle. Ec, erythrocytes. Cluster 4, cluster 4 in mCherry-positive tumor. UHF, upper hair follicle. TAM, tumor-associated macrophages.

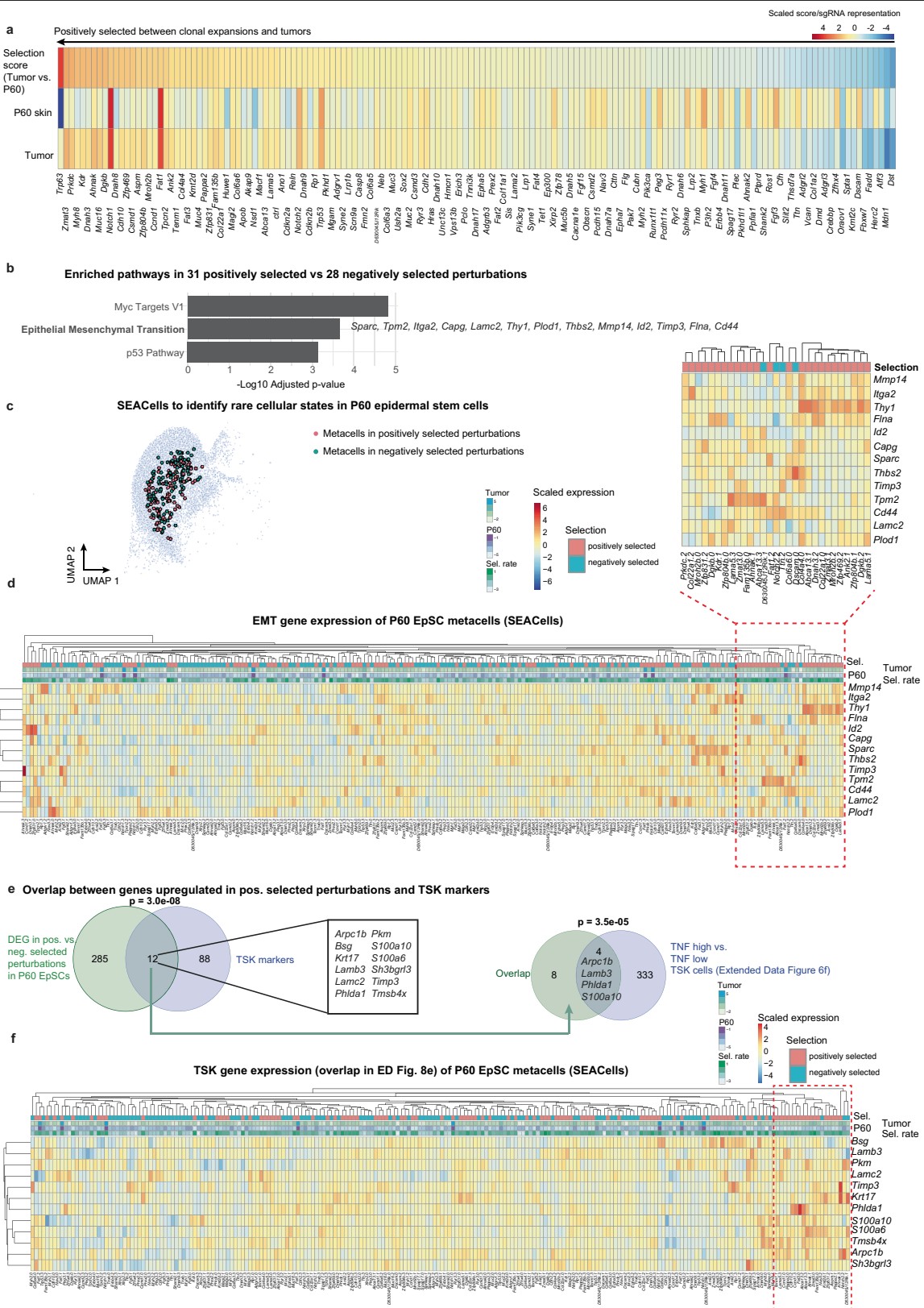

**a** Positively selected between clonal expansions and tumors

Scaled score/sgRNA representation

**b** Enriched pathways in 31 positively selected vs 28 negatively selected perturbations

Myc Targets V1

Epithelial Mesenchymal Transition — *Sparc, Tpm2, Itga2, Capg, Lamc2, Thy1, Plod1, Thbs2, Mmp14, Id2, Timp3, Flna, Cd44*

p53 Pathway

-Log10 Adjusted p-value

**c** SEACells to identify rare cellular states in P60 epidermal stem cells

Metacells in positively selected perturbations
Metacells in negatively selected perturbations

**d** EMT gene expression of P60 EpSC metacells (SEACells)

**e** Overlap between genes upregulated in pos. selected perturbations and TSK markers

p = 3.0e-08

DEG in pos. vs. neg. selected perturbations in P60 EpSCs

285  12  88  TSK markers

*Arpc1b  Pkm*
*Bsg  S100a10*
*Krt17  S100a6*
*Lamb3  Sh3bgrl3*
*Lamc2  Timp3*
*Phlda1  Tmsb4x*

p = 3.5e-05

Overlap  8  *Arpc1b Lamb3 Phlda1 S100a10*  333  TNF high vs. TNF low TSK cells (Extended Data Figure 6f)

**f** TSK gene expression (overlap in ED Fig. 8e) of P60 EpSC metacells (SEACells)

**Extended Data Fig. 8** | See next page for caption.

**Extended Data Fig. 8 | SEACells identifies rare cell states with invasive properties. a**, The heatmap shows the selection score for all 150 perturbations between P60 and tumors, revealing both positively and negatively selected perturbations during tumor initiation. Because of the n = 168 tumors, we focused our main analysis on the top 30 perturbations in P60 EpSCs and the top 30 perturbations in tumors to ensure the robustness and high coverage of the selection score, as presented in Fig. 5c. The selection score (sgRNA tumor/ sgRNA P60) is compared to the selection score of the median control sgRNA library. P60 total representation of perturbation in P60 skin (dial-out data). Tumor, representation of sgRNA in tumors, calculated as the sum of the percentage in each of the 168 tumors. **b**, Pathway enrichment of differentially expressed genes between 31 positively selected perturbations and 28 negatively selected perturbations (as shown in Fig. 5c) uncovers significant enrichments in genes associated with epithelial-mesenchymal transition (EMT), Myc targets and the p53 pathway. Two-tailed Fisher's exact test was used to examine whether a MsigDB Hallmark gene set was significantly enriched. **c**, Identification of rare cell states in P60 epidermal stem cells by single-cell aggregation of cell states (SEACells)[42]. SEACells algorithm was applied to identify metacells in positively and negatively selected perturbations, upper panel. Positively selected and negatively selected metacells are highlighted as red and green large cells.

**d**, Heatmap shows average gene expression of epithelial-mesenchymal transition (EMT) genes that were differentially expressed between positively and negatively selected perturbations metacells (Extended Data Fig. 8b). The number next to the perturbation refers to the metacell number. Dashed box highlights clusters of metacells expressing high levels of EMT genes, in line with a small subpopulation of P60 epidermal stem cells with invasive features. A closeup of the red dashed region is provided in the upper panel. **e**, Positively selected perturbations also induce 12 TSK markers. The Venn diagram shows the overlap between the genes induced in the 31 positively selected perturbations and the 100 TSK markers[36]. In contrast, the same comparison with the genes induced in negatively selected perturbations only resulted in an overlap of 2 genes (p = 0.46). Right panel, Venn diagram showing the overlap of 12 TSK markers and the genes upregulated in TNF high vs. low TSKs, suggesting that a subset of these TSK markers is also induced by TNF. P values indicate a hypergeometric test. **f**, Rare cell states induce TSK marker genes even in P60 clonally expanded epidermal stem cells. Heatmap shows scaled average expression of the TSK genes identified as the overlap of genes induced by the positively selected perturbations and the TSK markers (Extended Data Fig. 8e). Metacells were identified by SEACells as outlined in Extended Data Fig. 8c. The number next to the perturbation refers to the metacell number.

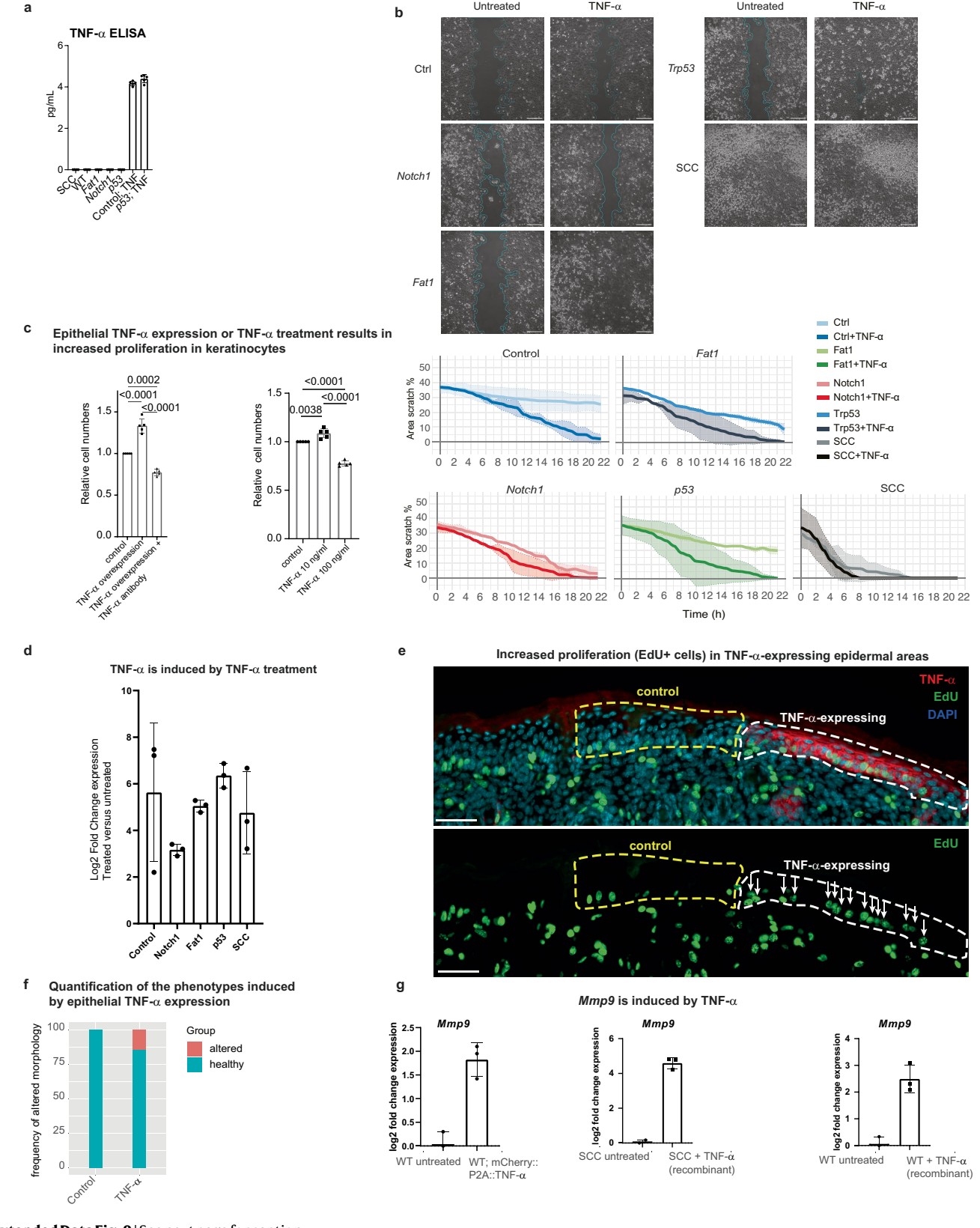

**Extended Data Fig. 9** | See next page for caption.

**Extended Data Fig. 9 | TNF induces motility in keratinocytes. a**, TNF secretion in keratinocytes transduced with the TNF expression construct, as demonstrated by an Enzyme-linked Immunosorbent Assay (ELISA) of the cell culture supernatant. WT, wild type. SCC, squamous cell carcinoma cells. *Fat1, Notch1* and *p53* indicate sgRNA-infected knockout keratinocytes. Error bars indicate standard deviation of the mean from n = 5 independent experiments. **b**, In vitro scratch-wound assay shows that TNF can induce cellular motility in keratinocytes. Different perturbations in keratinocytes isolated from *Rosa26-Cas9* mice (and infected and selected with *Notch1, Fat1* and *p53* sgRNAs) and squamous cell carcinoma cells (SCC) were treated with recombinant murine TNF and wound closure was tracked over time in 1 h intervals. Scale bar, 200 μm. Data indicate the average ± s.d. of 3 experiments. **c**, Epithelial TNF expression or TNF treatment result in increased proliferation. Left panel, Methylthiazolyldiphenyl-tetrazolium bromide (MTT) assay in either control, TNF-expressing keratinocytes or TNF-expressing keratinocytes treated with TNF antibodies. Right panel, MTT assay in keratinocytes either mock-treated or treated with ectopic TNF. As previously reported, high TNF concentration (100 ng/ml) leads to cell death. Error bars indicate standard deviation of the mean from n = 5 independent experiments. P values indicate one-sided Welch's t-test. **d**, TNF treatment induces epithelial TNF expression. RT-qPCR was performed for treated and untreated keratinocytes and TNF induction was calculated. Error bars represent standard deviation of the mean from n = 3 individual experiments. **e**, Increased proliferation in the TNF-expressing epidermal areas. Representative section of a TNF-expressing and control skin area (see quantifications in Fig. 5h). EdU incorporation 1 h post-injection in P4 animals. E9.5 embryos were microinjected with the TNF::P2A::mCherry lentivirus. Scale bars, 50 μm. **f**, Quantification of the TNF expression phenotype was performed using immunofluorescence staining of P4, P17, and P27 epidermal sections obtained from animals injected with a TNF overexpression construct. Infected regions were identified based on TNF staining, while adjacent negative regions served as controls. Quantified as altered were regions that displayed hyperproliferation of epidermal stem cells, epithelial invaginations or breakdown of the basal membrane. N = 42 TNF-positive regions and n = 42 control regions. **g**, Epithelial TNF expression or recombinant murine TNF treatment induces epithelial *Mmp9* mRNA. RT-qPCR for *Mmp9* was performed for treated/infected and untreated keratinocytes. The lentiviral construct for TNF expression is displayed in Fig. 5e. Error bars indicate standard deviation of the mean from n = 3 independent experiments.

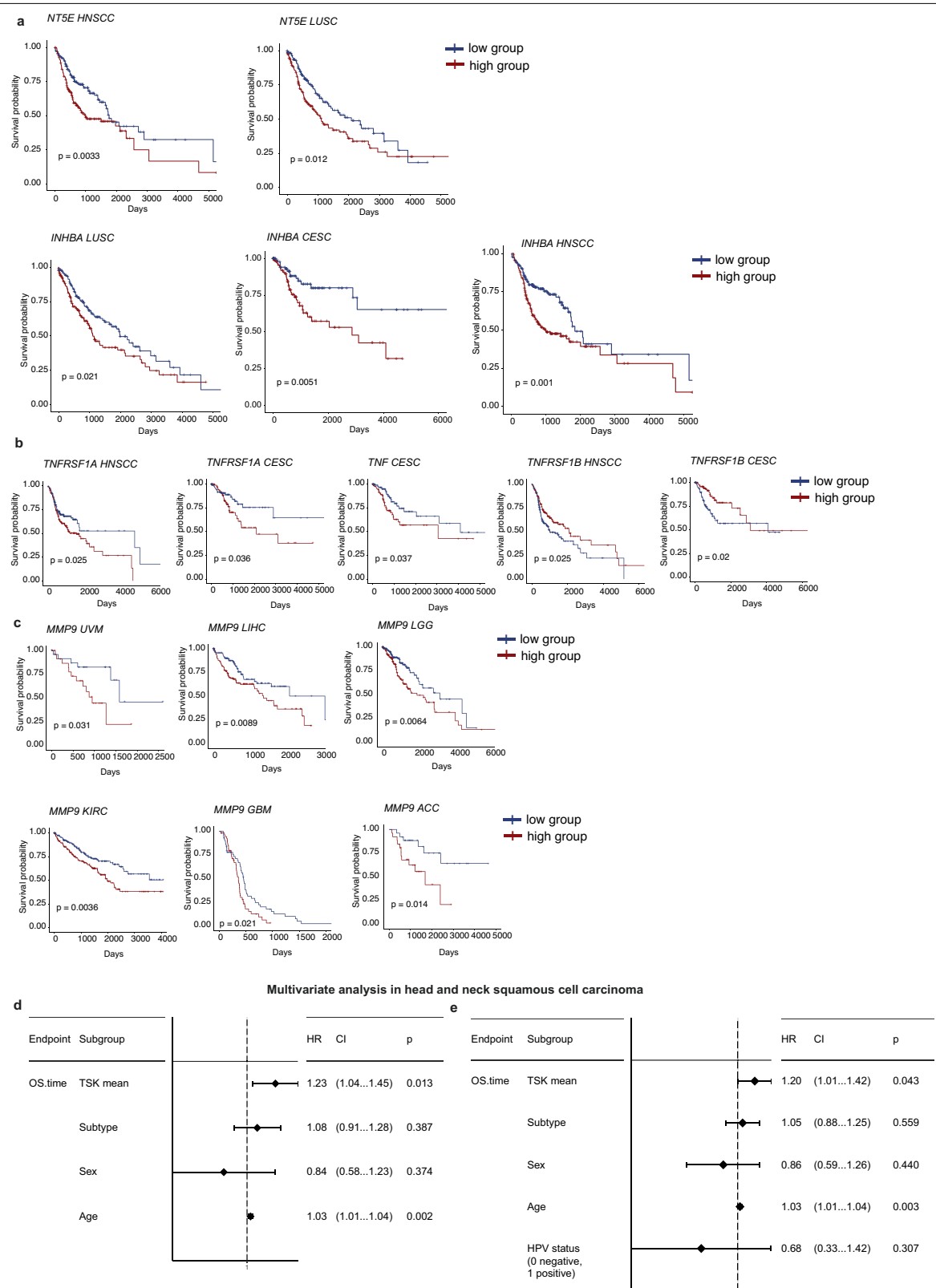

**Extended Data Fig. 10 |** See next page for caption.

**Extended Data Fig. 10 | TSK mRNA levels correlate with overall survival in human SCC patients. a**, TSK gene mRNA levels (*INHBA* and *NT5E*) correlate with shorter overall survival in human squamous cell carcinoma patients. LUSC, lung squamous cell carcinoma. HNSCC, head and neck squamous cell carcinoma. CESC, cervical squamous cell carcinoma. All Kaplan-Meier plots show the overall survival, stratified by upper tertile (high) and bottom tertile (low) mRNA expression. P value indicates a standard log-rank test. **b**, TNF, TNF receptor 1 and 2 mRNA level correlation with overall survival in human squamous cell carcinoma patients. P value indicates a standard log-rank test. **c**, *Mmp9* mRNA levels correlate with shorter overall survival in a total of 6 different cancer types. UVM, Uveal Melanoma. LIHC, Liver hepatocellular carcinoma. LGG, Brain Lower Grade Glioma. KIRC, Kidney renal clear cell carcinoma. GBM, Glioblastoma multiforme. ACC, Adrenocortical carcinoma. P value indicates a standard log-rank test. **d**, The TSK signature is an independent prognostic factor for overall

survival in head and neck squamous cell carcinoma patients. Multivariate analysis in head and neck squamous cell carcinoma with TSK mean expression of 11 genes, sex, age, subtypes (atypical n = 67, classical n = 87, mesenchymal n = 49 and basal n = 75, as described by the Cancer Genome Atlas Network[78]) as covariates. TSK genes: *MMP9, MMP10, PTHLH, FEZ1, IL24, KCNMA1, INHBA, MAGEA4, NT5E, LAMC2 and SLITRK6*. HR, hazard ratio. CI, confidence interval. P values indicate a Wald test. **e**, The TSK signature is an independent prognostic factor for overall survival in head and neck squamous cell carcinoma patients. Multivariate analysis in head and neck squamous cell carcinoma with TSK mean expression of 11 genes, sex, age, subtypes (atypical n = 67, classical n = 87, mesenchymal n = 49 and basal n = 75, as described by the Cancer Genome Atlas Network[78]) and HPV status as covariates. TSK genes: *MMP9, MMP10, PTHLH, FEZ1, IL24, KCNMA1, INHBA, MAGEA4, NT5E, LAMC2 and SLITRK6*. HR, hazard ratio. CI, confidence interval. P values indicate a Wald test.

# Reporting Summary

## Statistics

For all statistical analyses, confirm that the following items are present in the figure legend, table legend, main text, or Methods section.

| n/a | Confirmed | |
|---|---|---|
| ☐ | ☒ | The exact sample size (*n*) for each experimental group/condition, given as a discrete number and unit of measurement |
| ☐ | ☒ | A statement on whether measurements were taken from distinct samples or whether the same sample was measured repeatedly |
| ☐ | ☒ | The statistical test(s) used AND whether they are one- or two-sided *Only common tests should be described solely by name; describe more complex techniques in the Methods section.* |
| ☐ | ☒ | A description of all covariates tested |
| ☒ | ☐ | A description of any assumptions or corrections, such as tests of normality and adjustment for multiple comparisons |
| ☐ | ☒ | A full description of the statistical parameters including central tendency (e.g. means) or other basic estimates (e.g. regression coefficient) AND variation (e.g. standard deviation) or associated estimates of uncertainty (e.g. confidence intervals) |
| ☐ | ☒ | For null hypothesis testing, the test statistic (e.g. *F*, *t*, *r*) with confidence intervals, effect sizes, degrees of freedom and *P* value noted *Give P values as exact values whenever suitable.* |
| ☒ | ☐ | For Bayesian analysis, information on the choice of priors and Markov chain Monte Carlo settings |
| ☒ | ☐ | For hierarchical and complex designs, identification of the appropriate level for tests and full reporting of outcomes |
| ☐ | ☒ | Estimates of effect sizes (e.g. Cohen's *d*, Pearson's *r*), indicating how they were calculated |

*Our web collection on statistics for biologists contains articles on many of the points above.*

## Software and code

Policy information about availability of computer code

| Data collection | -Microscopy images were obtained using a Zeiss Axio Observer with Zen software (version 3.1)<br>-Western blot luminesence was recorded using the GE ImageQuant LAS4000 (version 1.2)<br>-Single cell capture was performed with the BD Rhapsody Scan and Analysis software (version 1.3.0.13)<br>-ELISA readings were taken using a Tecan i-control (version 2.0)<br>-Quant Studio Real-Time PCR software (version 1.3) was used to record qPCR results.<br>-Flow cytometry data was acquired with the FACSDiva software (BD Biosciences, version 8.0.1) |
|---|---|
| Data analysis | Single-cell RNA sequencing data were processed with the following tools/packages using the commands described in the methods section:<br>BD Rhapsody Pipeline hosted at the Seven Bridge's cloud platform (https://www.sevenbridges.com/, version 1.8-1.9.1)<br>STAR aligner (version 2.7.10b)<br>Scrublet Python Package (version 0.2.3)<br>MAST package (version 1.26.0)<br>pheatmap package (version 1.0.12)<br>pyMDE package (version 0.1.15)<br>MAGECK package (version 0.5.9)<br>WGCNA package (version 1.72-1)<br>metap package (version 1.8)<br>Cellchat (version 1.6.0)<br>RPCA (version 0.2.3)<br>SEACells (version 0.3.3)<br>EnrichR (version 3.2) |

GSEApy (version 1.0.5)
DeSeq2 (version 3.18)
ImageJ (stable release 1.54h)

Spatial transcriptomics data:
Space Ranger software (version 2.0.1, 10x Genomics)
spacexr package (version 2.2.1, https://github.com/dmcable/spacexr)
Seurat R package (version 4.0.3)
ISCHIA package (version 1.0.0.0)

qPCR data was analysed with the Quant Studio Real-Time PCR software (version 1.3) and Microsoft Excel 365(Version 2302)
Immunofluoresence pictures were taken with the ZEN Blue microscopy software (version 3.1). Images shown are maximum intensity projections.
Flow cytometry data were analyzed both with the BD FACSDiva (BD Biosciences, version 8.0.1) and FlowJo (version 10.8.1).

Custom scripts developed in this study is available at https://github.com/sendoellab/single-cell_CRISPR

For manuscripts utilizing custom algorithms or software that are central to the research but not yet described in published literature, software must be made available to editors and reviewers. We strongly encourage code deposition in a community repository (e.g. GitHub). See the Nature Portfolio guidelines for submitting code & software for further information.

## Data

Policy information about availability of data

All manuscripts must include a data availability statement. This statement should provide the following information, where applicable:
- Accession codes, unique identifiers, or web links for publicly available datasets
- A description of any restrictions on data availability
- For clinical datasets or third party data, please ensure that the statement adheres to our policy

The complete single-cell RNA sequencing and CROP-seq data for P4 stage, P60 stage, tumor amplicon sequencing and RNA-sequencing data are available on the GEO GSE235325.
Gene expression data in actinic keratosis and skin SCC was extracted from Chitsazzadeh et al., Nature Communications, 2016, https://doi.org/10.1038/ncomms12601.
Molecular signature database hallmark genes for gene set enrichment analysis was accessed via https://www.gsea-msigdb.org/gsea/msigdb/mouse/collections.jsp
TCGA data was accessed via xena.ucsc.edu.

## Research involving human participants, their data, or biological material

Policy information about studies with human participants or human data. See also policy information about sex, gender (identity/presentation), and sexual orientation and race, ethnicity and racism.

| Reporting on sex and gender | not applicable |
|---|---|
| Reporting on race, ethnicity, or other socially relevant groupings | not applicable |
| Population characteristics | not applicable |
| Recruitment | not applicable |
| Ethics oversight | not applicable |

Note that full information on the approval of the study protocol must also be provided in the manuscript.

## Field-specific reporting

Please select the one below that is the best fit for your research. If you are not sure, read the appropriate sections before making your selection.

☒ Life sciences        ☐ Behavioural & social sciences        ☐ Ecological, evolutionary & environmental sciences

For a reference copy of the document with all sections, see nature.com/documents/nr-reporting-summary-flat.pdf

## Life sciences study design

All studies must disclose on these points even when the disclosure is negative.

| Sample size | Sample size was not predetermined by statistical methods. Based on previous screens and general coverage requirements for high-content CRISPR screenings (Bock, C., Datlinger, P., Chardon, F. et al. High-content CRISPR screening. Nat Rev Methods Primers 2, 8 (2022). https://doi.org/10.1038/s43586-021-00093-4; Xin Jin et al. In vivo Perturb-Seq reveals neuronal and glial abnormalities associated with autism risk |
|---|---|

genes. Science 370, eaaz6063 (2020). DOI:10.1126/science.aaz6063), we aimed for a coverage of our sgRNA library of at least 200x (cells/sgRNA) in the in vivo samples. We achieved a coverage of 240x for the P4 and 366x for the P60 skin.

| | |
|---|---|
| Data exclusions | For single cell CRISPR analysis, low quality and doublet cells were filtered out following standard filtering parameters. Details are provided in the methods section. |
| Replication | For single-cell RNA sequencing data, we performed 7 replicates for the P4 time point, resulting in a total number of 120,077 cells after filtering and sgRNA annotation. For the P60 time point, we performed 8 replicates, resulting in a total number of 183,084 cells after filtering and sgRNA annotation.<br>For TNFR1-dependency (Fig. 3c), 8 wild-type and 4 Tnfr1 replicates were performed.<br>Tumor single-cell data were derived from 30 individual tumors from 2 animals, separated into mCherry-positive and mCherry-negative cells. scRNA-seq library preparations were performed separately for mCherry-positive and mCherry-negative cells.<br>ELISA was carried out in biological and technical triplicates.<br>MTT assay was carried out in 5 biological replicates and technical triplicates.<br>in vivo EdU assay was quantified across 12 separate sections with areas of 25 cells or more<br>qPCR was carried out in biological triplicates and technical duplicates.<br>All attempts at replication were successful. |
| Randomization | For mouse experiments, mice of relevant genotype were randomly assigned to each experimental protocol. |
| Blinding | Investigators were not blinded during data acquisition. Since most of our readouts are sequencing-based, blinding would not be expected to have an effect on the experimental outcome. |

# Reporting for specific materials, systems and methods

We require information from authors about some types of materials, experimental systems and methods used in many studies. Here, indicate whether each material, system or method listed is relevant to your study. If you are not sure if a list item applies to your research, read the appropriate section before selecting a response.

## Materials & experimental systems

| n/a | Involved in the study |
|---|---|
| ☐ | ☒ Antibodies |
| ☐ | ☒ Eukaryotic cell lines |
| ☒ | ☐ Palaeontology and archaeology |
| ☐ | ☒ Animals and other organisms |
| ☒ | ☐ Clinical data |
| ☒ | ☐ Dual use research of concern |
| ☒ | ☐ Plants |

## Methods

| n/a | Involved in the study |
|---|---|
| ☒ | ☐ ChIP-seq |
| ☐ | ☒ Flow cytometry |
| ☒ | ☐ MRI-based neuroimaging |

## Antibodies

| | |
|---|---|
| Antibodies used | Primary antibodies for immunofluoresence:<br>rat anti-RFP (Chromotek, 5F8; 1:200), rabbit anti-RFP (MBL, PM005; 1:200), chicken anti-GFP (Abcam, ab13970; 1:200), rat anti-CD45-biotin (Biolegend, 103104; 1:200), goat anti-TNFR1 (R&D, AF-425-PB; 1:200), rabbit anti mouse TNF-alpha (CST #11948 [D2D4] XP[R, 1:300), rat Anti-Mouse CD104 aka ITGB beta 4 (BD Pharmingen 553745, 1:300), chicken anti-Keratin14 (Biolegend 906001, 1:300).<br>Secondary antibodies for immunofluoresence:<br>All secondary antibodies used were raised in a donkey host and were conjugated to Alexafluor488(Codes: 703-545-155, 712-545-150, 712-545-153), Cy3(Codes:703-165-155, 711-165-152), or AlexaFluor647(Codes:703-605-155,711-605-152,712-605-153) (Jackson ImmunoResearch Laboratory; 1:500- 1:1000).<br><br>Primary antibodies for western blot:<br>Anti-TNF-α (Cell Signaling Technology, #11948), Anti-Vinculin (Abcam, ab129002)<br>Secondary Antibodies Western Blot:<br>Goat Anti-Rabbit IgG HRP Linked Antibody (Cell Signaling Technology 7074S)<br><br>Antibodies for Immune Cell depletion:<br>rat anti-mouse CD4 (YTS 191.1, Hoelzel Diagnostika LEIN-C3210)<br>rat anti-mouse CD8 (YTS169.4 Hoelzel Diagnostika LEIN-C2850)<br>rat anti-Mouse CD115 (CSF-1R, Hoelzel Diagnostika LEIN-C2268)<br>rat IgG isotype control (Hoelzel Diagnostika, LEIN-I-1177). |
| Validation | Antibodies were validated by the manufacturer and multiple publications. Anti-TNF-α antibody was additionally validated by western blot and immunofluorescence in TNF-α-negative cells.<br><br>rat anti-RFP (Chromotek, 5F8) - Zhang Y, Rózsa M, Liang Y, et al. Fast and sensitive GCaMP calcium indicators for imaging neural populations. Nature. 2023;615(7954):884-891. doi:10.1038/s41586-023-05828-9 |

rabbit anti-RFP (MBL, PM005) - Ishida Y et al. Autophagic elimination of misfolded procollagen aggregates in the endoplasmic reticulum as a means of cell protection. Mol Biol Cell. 20, 2744-54 (2009)(PMID:19357194)

chicken anti-GFP (Abcam, ab13970) - Berg EM  et al. Brainstem circuits encoding start, speed, and duration of swimming in adult zebrafish. Neuron 111:372-386.e4 (2023)

rat anti-CD45-biotin (Biolegend, 103104) - Podd BS, Thoits J, Whitley N, et al. T cells in cryptopatch aggregates share TCR gamma variable region junctional sequences with gamma delta T cells in the small intestinal epithelium of mice. J Immunol. 2006;176(11):6532-6542. doi:10.4049/jimmunol.176.11.6532

goat anti-TNFR1 (R&D, AF-425-PB) - Knizkova D, Pribikova M, Draberova H, et al. CMTM4 is a subunit of the IL-17 receptor and mediates autoimmune pathology. Nat Immunol. 2022;23(11):1644-1652. doi:10.1038/s41590-022-01325-9

rabbit anti mouse TNF-alpha (CST #11948) - Chan L, Chung CC, Yu RC, Hong CT. Cytokine profiles of plasma extracellular vesicles as progression biomarkers in Parkinson's disease. Aging (Albany NY). 2023;15(5):1603-1614. doi:10.18632/aging.204575

rat Anti-Mouse CD104 aka ITGB beta 4 (BD Pharmingen 553745) - Giancotti FG. Signal transduction by the alpha 6 beta 4 integrin: charting the path between laminin binding and nuclear events. J Cell Sci. 1996;109 ( Pt 6):1165-1172. doi:10.1242/jcs.109.6.1165

chicken anti-Keratin14 (Biolegend 906001) - Radtke AJ, Kandov E, Lowekamp B, et al. IBEX: A versatile multiplex optical imaging approach for deep phenotyping and spatial analysis of cells in complex tissues. Proc Natl Acad Sci U S A. 2020;117(52):33455-33465. doi:10.1073/pnas.2018488117

Anti-Vinculin (Abcam, ab129002) - Hühn D, Martí-Rodrigo P, Mouron S, et al. Prolonged estrogen deprivation triggers a broad immunosuppressive phenotype in breast cancer cells. Mol Oncol. 2022;16(1):148-165. doi:10.1002/1878-0261.13083

Goat Anti-Rabbit IgG HRP Linked Antibody (Cell Signaling Technology 7074S) - Sakamoto A, Inoue H, Miyamoto S, Ito S, Soda Y, Tani K. Coxsackievirus A11 is an immunostimulatory oncolytic virus that induces complete tumor regression in a human non-small cell lung cancer. Sci Rep. 2023;13(1):5924. Published 2023 Apr 12. doi:10.1038/s41598-023-33126-x

Jackson Immuno Goat secondary for IF - Ha J, Shin J, Seok E, Kim S, Sun S, Yang H. Estradiol and progesterone regulate NUCB2/ nesfatin-1 expression and function in GH3 pituitary cells and THESC endometrial cells. Anim Cells Syst (Seoul). 2023;27(1):129-137. Published 2023 Jun 20. doi:10.1080/19768354.2023.2226735

rat anti-mouse CD4 (YTS 191.1, Hoelzel Diagnostika LEIN-C3210) - 1. Tallón de Lara, P. et al. CD39+PD-1+CD8+ T cells mediate metastatic dormancy in breast cancer. Nat. Commun. 2021 121 12, 1–14 (2021).
rat anti-mouse CD8 (YTS169.4 Hoelzel Diagnostika LEIN-C2850) - 1. Tallón de Lara, P. et al. CD39+PD-1+CD8+ T cells mediate metastatic dormancy in breast cancer. Nat. Commun. 2021 121 12, 1–14 (2021).
rat anti-Mouse CD115 (CSF-1R, Hoelzel Diagnostika LEIN-C2268) -  1. Lelios, I. et al. Monocytes promote UV-induced epidermal carcinogenesis. Eur. J. Immunol. 51, 1799 (2021).

# Eukaryotic cell lines

Policy information about cell lines and Sex and Gender in Research

| Cell line source(s) | Lenti-X(TM) 293T cells for virus production were purchased from TaKaRa Clontech (632180). Mouse keratinocyte cells were derived in-house as described in the method section. |
|---|---|
| Authentication | Commercially available cell lines were authenticated by the vendor. No additional authentication was performed. |
| Mycoplasma contamination | Cell lines were tested for mycoplasma every three months using the Mycoplasma PCR detection kit and confirmed negative (Sigma; D9307). |
| Commonly misidentified lines (See ICLAC register) | none of the misidentified lines were used in this study |

# Animals and other research organisms

Policy information about studies involving animals; ARRIVE guidelines recommended for reporting animal research, and Sex and Gender in Research

| Laboratory animals | Tg(B6J.129(Cg)-Gt(ROSA)26Sortm1.1(CAG-cas9*,-EGFP)Fezh/J (denoted as "B6.Cas9") were purchased from the Jackson Laboratory (strain #026179). B6.129S Tnfrsf1atm1ImxTnfrsf1btm1Imx (denoted as Tnfr1-ko, originally from Jackson lab #003243) were acquired through the Swiss Immunology Mouse Repository (SwImMR). Wildtype mice of the strain CD1-IGS (denoted as "CD1") were purchased from Charles River. Animals were sacrificed as indicated in the manuscript for the single-cell data at either postnatal day 4(P4), or 60(P60). Immune depleted animals also at P60. DMBA/TPA treated animals were sacrificed at age postnatal day P144. TNF-overexpression animals (Fig. 5) were sacrificed at either P4, 17 or 27 |
|---|---|
| Wild animals | No wild animals were used in this study. |

| Reporting on sex | P4 and P60 single-cell data contains both male and female animals.<br>TNFR1 dependency on P60 animals included both P60 male and female animals.<br>For the DMBA/TPA, chemical carcinogenesis tumor single-cell data, only female animals were used as recommended because male mice may fight, causing damage to the dorsal skin. (Filler, R. B., Roberts, S. J., & Girardi, M. (2007). Cutaneous two-stage chemical carcinogenesis. Cold Spring Harbor Protocols, 2007(9), pdb-prot4837.) |
| --- | --- |
| Field-collected samples | No field collected samples were used in this study. |
| Ethics oversight | The Animal Welfare Committee of the Canton of Zurich approved all animal protocols and experiments performed in this study (animal permits ZH074/2019, ZH196/2022) |

Note that full information on the approval of the study protocol must also be provided in the manuscript.

# Flow Cytometry

## Plots

Confirm that:

☒ The axis labels state the marker and fluorochrome used (e.g. CD4-FITC).

☒ The axis scales are clearly visible. Include numbers along axes only for bottom left plot of group (a 'group' is an analysis of identical markers).

☒ All plots are contour plots with outliers or pseudocolor plots.

☒ A numerical value for number of cells or percentage (with statistics) is provided.

## Methodology

| Sample preparation | Single-cell suspension was achieved with mechanical dissociation and trypsin digest as described in the method section. |
| --- | --- |
| Instrument | BD FACSAria III |
| Software | BD FACSDiva (BD Biosciences, version 8.0.1) and FlowJo (version 10.8.1) |
| Cell population abundance | mCherry-positive rates were between 2.4%-16.3%. We sorted between 300,000-400,000 cells per replicate. |
| Gating strategy | 1) FSC-A vs SSC-A was used to gate for the bulk population of cells<br>2) FSC-A vs FSC-H was used to minimize doublet sorting<br>3) DAPI-A vs FSC-H was used to gate for live cells<br>4) PE-Texas Red- A vs FSC-H was used to gate for lentivirus-infected cells<br>Please also see Extended Data Figure 1d, which exemplifies the gating strategy. |

☒ Tick this box to confirm that a figure exemplifying the gating strategy is provided in the Supplementary Information.

