## [Peer Review File · Nature]

Manuscript Title: In vivo single-cell CRISPR uncovers distinct TNF- α programs in tumor evolution

Reviewer Comments & Author Rebuttals

Reviewer Reports on the Initial Version:

Referees' comments:

Referee #1 (Remarks to the Author):

This is a beautifully written manuscript that represents an enormous amount of work. As far as I am aware this is the first experiment to perform in vivo CROP-seq/perturb-seq to interrogate the combined effect of gene disruption whilst also examining cell-type/state at the interface of clonal dynamics.

I have a number of specific comments/questions:

1. A major technical bottleneck in experiments of this kind is getting good representation of gRNAs within the population of infected cells so as to explore their biological effect. The data in Supplementary Figure 1f is opaque to me. Clearly the strong FAT1 gRNAs are enriched but the representation of the control gRNAs does not change over time and is close to zero throughout the experiment. This is a surprise to me as I would have expected gRNAs that have no biological effect to be diluted over time but supp figure 1f is not informative (or is drawn in such a way that the message is unclear – maybe plot absolute normalised read counts instead, or at least provide them). I also find the exact experimental design confusing. i.e. up to 8 mice per litter were injected but how many litters were analysed and how many embryos in total? I can glean from Supp Figure 1i that 8 “samples” were analysed but was this 8 embryos? What was the variation in these experiments? The data in Figure 2a is not presented with any statistical analysis (or error bars) which makes me wonder how to interpret this result. The top 10 enriched and depleted genes are listed but I am unclear what to make of this beyond that several known skin drivers are shown. Ryr3 (for example) is mainly expressed in muscle (according to GTEx) and has a TPM <1 in skin but is “enriched” in figure 2a. As I read further I see that Supp Fig 1i shows some of the raw data but also suggests there are significant sample/batch effects. i.e. WTA12 and 13 are (almost) always depleted while WTA18 and 17 show little variation.

2. The authors make a strong statement at the bottom of the second paragraph of page 5 that the “rates of clonal expansion in their model faithfully recapitulate...that of normal human skin”. I don't feel that they provide evidence for this. In reality the “rate” of clonal expansion is a feature of time and clone size and studies in human the lack temporal information required to make such conclusions.

3. I was surprised to see that Sox10 was pulled out as being a gene involved in a subcluster in supp Figure 4C. Sox10 (as far as I can see) is not displayed in this figure. Sox10 is required for melanocyte survival so I would expect its disruption in these cells to be lethal.

4. I would suggest the authors do a multi-variate analysis in Figure 5o and replicate the signature across other datasets. While I think the biological hypothesis is sound an analysis using a heterogenous cohort such as TCGA can be problematic. For example, what happens if the HPV-driven cases and those that are not HPV-associated are analysed separately? These subtypes are markedly different in terms of mutational load and driver landscape, the former lacking FAT1 and TP53 mutations, for example. Similarly, HNSCC can be classified as atypical, classical, mesenchymal and basal with notable differences in mutation profile and expression and by definition histopathological appearance. How does this influence the association between the TSK signature and outcome? HPV-associated disease is more common in men. How does sex influence the TSK signature?

5. The authors note by citing reference 29 that it has long been established that the TNF-alpha pathway is implicated in SCC tumour initiation and loss of TNFR1/2 severely decreases pre-malignant (papilloma) tumour initiation and growth in classical mouse carcinogen models such as DMBA/TPA. Indeed, the experiments using Tnfr1 knockouts in this paper are an extension of this original work. I think the authors should more actively acknowledge this prior work.

Minor point

I think the supplementary table "500 sgRNAs" is truncated from line 452 onwards. From this file it's not immediately clear to me which of the gRNAs are controls. Further the gRNA names are not present in the additional tabs with the counts. This makes it very hard for me to follow (or replicate what was done without re-writing some code myself). What does library_A, _B, _C represent? I can't follow from the methods. How do these counts relate back to the data in (for example) Figure 2A. Are they normalised counts or raw counts? The supp tables don't have titles or explainers to help and I can't see any table legends. I would strongly suggest that all of the analyses are presented as Jupyter notebooks (or similar) so that readers can follow what was done.

Referee #2 (Remarks to the Author):

In their manuscript, Lenz and colleagues study how clonal expansion influences malignant transformation in the epidermis. Using a targeted CRISPR/Cas-based approach, the authors delete 150 commonly mutated cancer genes (mainly tumor suppressors) in the developing epidermis. They then follow the clonal behavior of the perturbed cells in neonatal and adult skin. As expected, they find that cancer-related gene losses, such as p53, Notch1/2, and Fat1, correlate with clonal expansion. A commonly up-regulated pathway in cancer gene perturbed epidermal cells is TNF-alpha signaling that promotes clonal expansion, yet is down-regulated in early stages of tumorigenesis, and specifically up-regulated at the tumor invasive front.

The study addresses the question how clonal expansion in normal epithelia shape or drive malignant transformation. The novelty of this study is that potential driving genetic events are induced in all cell types. Targeting the commonly perturbed genes in squamous cell carcinoma in development and following the long-term effects in an interesting approach that revealed novel insights into the role of clonal expansion to malignant transformations. My major concern with this study is that the authors draw firm conclusions from largely correlative data. The authors claim that TNF-alpha signaling plays different roles during clonal expansion, early stages of tumorigenesis and at the invasive front. As a major regulator of inflammatory responses, TNF-alpha is known to regulate cell death and survival and increasing expression of angiogenic factors. Thus, it remains unclear whether changes in TNF-alpha signaling are cause or consequence of perturbed cancer gene expression.

Major Comments:

1. Why is it required to sort the mCherry positive cells as opposed to just perform single cell sequencing of the whole skin, which would have the advantage to analyse cell-cell interaction between perturbed and non-perturbed cells and quantify clone size? Presumably this is done because the number of mCherry-positive cells is too low. In that case, the authors should comment on the efficiency of the in vivo injection method.
2. The enrichment of sgRNA targeting only one gene (Fat1) is shown (EDF1f). However, the design of the experiment would predict that nearly all sgRNAs should lead be enriched at P60. The authors should show the enrichment of all sgRNAs to see how many of the targeted genes caused indeed a clonal expansion, and the authors should show which sgRNAs are significantly enriched or depleted and include the information whether the average enrichment was shown (n=3 sgRNAs) or single guide RNAs.
3. The finding that TNF-alpha signaling pathways are enriched in expansion-driven perturbations is interesting but not entirely unexpected given that pro-inflammatory cytokines are commonly induced in damaged tissues and neoplasms. TNF-alpha has also been linked to clonal expansion in myelodysplastic syndromes (Abegunde et al, 2018; Fleischman et al, 2011) and the authors should acknowledge these findings.
4. It remains unclear whether the up-regulation of the TNF-alpha signaling pathway is directly caused by the loss of the respective genes or indirectly by for instance enhanced cell death of the affected cell populations. This could be addressed by testing whether the TNF-alpha signaling is

commonly up-regulated in squamous cell carcinoma patients (with the respective mutations) and whether it correlates with disease stage.

5. The authors show that the TNF-alpha signaling pathway is up-regulated in about 30% of sgRNAs but they do not show that the respective perturbed genes indeed consistently cause clonal expansion.

6. Page 8, first paragraph: The data and analyses shown do not warrant the conclusion that 'clonal expansions may need to be coupled to EMT downregulation to ensure that the expanding clones confine to the physiological tissue architecture of the epidermis.' The authors' statement indicates that the down-regulation of EMT genes is an active process to maintain tissue architecture, yet they provide no evidence. The analyse was biased to define expanding epidermal cell populations and dividing cells cannot migrate. Thus, the down-regulation of some ECM genes might simply be the consequence of enhanced division.

7. I am not convinced that the authors unequivocally showed that TNF-alpha signaling directly confers clonal expansion. TNF-alpha could be up-regulated due to cell damage or death caused by another sgRNA targeting for example p63. The loss of cells will be compensated by neighboring cells, always expand clones and give proliferation enhancing sgRNAs a competitive advantage. In that case, the clonal expansion is not directly caused by TNF-alpha but sgRNAs causing cell death close by. This would also explain the down-regulation of TNF-alpha modulated genes in early stages of tumorigenesis.

8. Without any information about initial clone sizes, little conclusion can be drawn by the data shown in Figure 5a-e. A larger clone is per se more likely to be hit by another mutation.

Minor Comments:

1. Three sgRNAs per gene is quite low in particular for in vivo screens. The authors need to comment on how they reached the decision that three sgRNAs are sufficient in that case.

Referee #3 (Remarks to the Author):

Nature manuscript 2023-06-10719

Title: "In vivo single-cell CRISPR uncovers distinct TNF- α programs in clonal expansion and tumorigenesis" by Renz et. al.

In the last 8 years, since the seminal work by Martincorena et.al. (Science 2015), mounting evidence has shown how our seemingly normal tissues accumulate somatic mutations throughout life, becoming a patchwork of mutant clones that compete with each other. Interestingly, despite the emergence of numerous mutations, including in cancer-associated genes, tissues remain healthy at large, suggesting that the nature of the mutations present in a particular tissue does not represent the sole cause of cancer. This raises a critical question "what are the processes that determine the predisposition of particular mutant clones to transform into cancer?".

The authors of the current manuscript are framing their work in this context, studying one of the most important and timely questions in the early cancer field. In particular, the authors aim at identifying common mechanisms dictating mutant clonal expansion under normal conditions (developmental and homeostatic) in the mouse skin, and go one step further by investigating how these change during carcinogenesis. The research questions covered in this manuscript are of significant relevance in the cancer field, offering the opportunity to understand the largely unclear factors defining mutant clonal predisposition to develop cancer; an aspect of paramount importance to develop more effective strategies to prevent and treat cancer.

What makes this study stand out is the elegant in vivo model used to address the ambitious question at task. The authors adapt the in utero lentiviral transduction method to develop an in vivo single-cell CRISPR screen that enables them to investigate clonal dynamics of the 150 most frequently mutated squamous cell carcinoma genes at the same time and in vivo. By implementing the previously published CROP-seq approach (Datlinger et. al. Nat Methods 2017) in an in vivo setting the authors are able to infer the transcriptional signature of individual genetic perturbations at the whole-tissue level, which is rather impressive. They further adapt the CROP-seq technology by incorporating a qualitative lineage tracing approach (lentiviral mCherry expression) to be able to sort the cells bearing genetic perturbations, and potentially track their fate.

Using this innovative approach, the authors reproduce previously published data in human skin (Martincorena et.al. Science 2015) showing that genetic perturbations in genes including Fat1, Notch1, Trp53 and Notch2 become enriched in adult mouse skin. This reflects their characteristic clonal expansion and validates the robustness of the approach used. Additionally, this study provides evidence of perturbations in other less characterised genes that also undergo clonal expansion/depletion, adding value to what it is already known.

The key aspect of this paper lays in the ability to explore genetic programs shared by multiple cancer gene perturbations. The authors conclude that the top expansion-driving perturbations exhibited gene expression changes associated with the activation of a non-cell autonomous TNF- α signalling, potentially through macrophage-epithelial communication. Interestingly, during tumour initiation they define a signalling switch, with TNF- α gene programme apparently becoming more cell-

autonomous and rather specific to a tumour cell subpopulation (TSKs). Ultimately, it is proposed that, in line with previous reports, this population is marking invasive tumour fronts and that the rewiring of the TNF- α signalling during carcinogenesis is a key step in tumour invasion.

Although the work presented here is interesting, and the question of high relevance, the presented conclusions are rather descriptive, overstated, and lack evidence to support some of the strong and definite claims made. Additional data and analysis will be needed to merit further consideration. See detailed comments below.

Major points

1-One of most relevant aspects of this paper lays in the ability to infer gene programs associated with particular genetic perturbations by combining sgRNA screening with scRNA-seq in vivo. This represents an innovative approach to identify the molecular mechanisms regulating mutant clonal expansion in a living tissue. Using this method, the authors conclude that “TNF- α signaling plays a universal role in mediating clonal expansions in normal epithelia”. However additional functional validation would be needed to substantiate this rather overemphasised statement. I have a few questions and concerns that would need to be resolved.

-As it stands, the authors mainly support the role of TNF- α clonal expansion in the form of a Tnfr1 KO experiment for which very limited data is presented. They choose to base their validation on TNFR1 dependency of the top 20 enriched genes at P4 (Figure 3E). Indeed, one would not expect that clonal dynamics is modulated in the same manner in adult and postnatal stages. Hence, validating the TNF- α module, identified at P60, using Tnfr1 KO mice at P4 seems rather unconventional. The authors should use the same time point for transcriptional characterization and validation experiments.

-Despite the power of their experimental set up, from which they can infer perturbation enrichment and clonal data through mCherry, the authors do not present any images and/or quantitative data showing changes in clonal expansion in Tnfr1 KO skin wholemounts. Although, I am aware that not all the enriched genetic perturbations showed TNF- α dependency, if “TNF- α signalling plays a universal role in mediating clonal expansions in normal epithelia”, then one would expect to see a significant shift in the size of the clones present in the KO tissue at P60 (rather than P4).

-Additionally, it would be interesting to obtain information on how perturbation enrichment is affected within a Tnfr1 KO background, rather than just focusing on the top 20 enriched genes at P4 (Figure 3e). Do different genetic perturbations become enriched in Tnfr1 KO, is this tilting expansion towards mutations that may be “less aggressive”? That would be interesting in its own right.

-The data suggests that the TNF- α stimulation is via a non-cell autonomous (potentially immune cells, in particular macrophages). However, the authors do not present any quantification of immune cells or macrophages within the mCherry areas of control versus Tnfr1 KO mice, to show an increase of immune recruitment. Nor they perform immune depletion experiments to prove the non-cell autonomous notion or validate the contribution of a particular immune population in the process. This is an important mechanistic aspect given that later on the authors suggest that TNF- α switches from paracrine to autocrine during tumorigenesis.

2-Next the authors explore “the contribution of TNF- α signaling to the cellular processes underlying tumor initiation” using the well-established DMBA/TPA carcinogenic protocol. Although this is interesting, one would have expected the data to follow an unbiased strategy towards identifying what makes genetic perturbations contribute to tumorigenesis. That is, i) determine the different

selection of genetic perturbations that occurs during tumorigenesis (as per Figure 5C and Extended 8A), ii) determine which tumour cell populations bear these perturbations (this is important and is missing), and iii) identify the transcriptional signature and regulatory networks defining the shift from “normal” clonal expansions (non-tumours) to selected perturbations in tumours (in the relevant cell population (ii)). This may still end up being TNF- α signalling and TSK cells, but this would be a more neutral approach than preselecting TNF- α signalling. This strategy would help render potential targets to tilt mutant clonal expansions towards “beneficial” genetic perturbations that may potentially prevent rather than promote tumorigenesis.

3- In relation to this point, there is an important aspect missing. Despite the richness of the data presented, the authors make absolutely no connection between the tumour cell populations, and the genetic alterations introduced using the sgRNA screen. It would be relevant to link the different genetic alterations with the expansion of specific tumour populations (similar to what is done in Figure 2). Is the TSK population the one bearing the genetic perturbations that are selected in tumours? Mechanistically it is rather unclear whether the TSK population, emphasized by the authors as a “subpopulation in clonal expansions that influences the predisposition for tumor initiation” is actually bearing the selected genetic perturbations. No data is presented validating the existence of TSKs in clonal expansion pre-tumour stage.

4- Following from the previous point, the authors study how TNF- α signalling contributes to the transition from clonal expansions to tumour initiation. The data shows that, although the TNF- α signature does not seem to be as apparent in mCherry cells in tumours, there is a subpopulation (termed TSK) showing an autocrine TNF- α signature. The relevance of TNF- α signalling for tumour formation is then validated by inducing TNF- α ectopic expression in epidermal cells (lentiviral approach). These results show that TNF- α expressing skin presents abnormalities including hyperproliferation (Figure 5L, which needs to be validated by EdU incorporation or other proliferative assays) and invasive behaviour (Figure 5N, which needs to be validated with EMT markers). However, an important experiment missing to fully validate the relevance of the TSK TNF- α signature in driving tumorigenesis would require blocking TNF- α signalling (maybe by using Tnfr1 KO) and define whether less tumours emerge, the TSK signature is diminished (at least by immunofluorescence), and the selection of genetic perturbations in the remaining tumours has shifted.

5-Strong statements are made throughout the text in relation to tumour/cancer progression and invasion. As far as I can see this study does not formally assess tumour progression or invasion. Hence, care should be taken when using those terms. Also, the data showing that the TSK population resides at the invasive tumour fronts, is rather unclear. Sections of side view tumours including a significant proportion of tumour stroma would be better suited to make this claim. The images presented, spatial transcriptomics and correlations shown make it very difficult to fully assess this point.

Minor points

6-Page 3, First line of the Introduction: “Tumor evolution begins with the transformation of a single cell that expands to form a tumor mass”. There is evidence suggesting that tumours may also have a polyclonal origin (Winton et.al Br J Cancer 1989; Revees et.al. Nat Cell Biol 2018).

7- The authors propose that the low MOI used for infection “minimized the possibility of perturbed cells being influenced by neighboring cells with different perturbations”. However, mCherry positive clusters (Figure 1c) appear rather close to each other (less than 15 to 20 cells distance). Suggesting that an interactive effect between mutations cannot be discarded. In order to claim that observations are not due to “perturbation” synergy, it would be necessary to validate mechanistic observations, such as Tnfr1 KO experiment, using low MOI infection for a limited (but relevant) selection of genetic CRISPR perturbations.

8-Despite the fact that authors use low MOI, it would be important to provide an immunofluorescence image showing mCherry labelling and CD45 staining for a large area of the skin (confocal tiled skin wholemount). It is important to visualize the extent to which skin cells show genetic cancer perturbations, which may lead to a systemic response (CD45, immune) beyond individual mutations.

9-There is also the concern as to whether the developmental aspect of the CRISPR screening used (gene perturbations are induced via in utero transduction) is influencing the clonal dynamics studied.

10-In relation to the authors statement “suggesting that Notch1 clonal expansion is primarily driven by the differentiation block and not by increased proliferation”, the fact that Notch inhibiting mutations lead to clonal expansion in squamous tissues through a differentiation imbalance has already been reported (Alcolea et.al. Nat Cell Biol 2014). Please revise.

11-How can the authors explain finding control sgRNAs among the most enriched perturbations (Fig 2A, ctrl1)?

12-It would be very helpful to highlight in Figure 2F which perturbations are enriched (undergo clonal expansion) as per Figure 2A.

13-When considering data on Figure 2H-K and Extended Figure 4D-E, one cannot help but wonder how specific this module is for TNF- α rather than a tissue stress response that would ultimately also involve TNF- α . The authors should include Pearson correlations for the other two main processes associated to the major EpSCs module to see how these compare to Figure 2K (UV response and Hypoxia). Along the same vein, one would need to see the equivalent of Figure 3A for those two additional processes.

14-In Figure 3e, the authors reveal how two of the major genetic perturbations contributing to clonal skin expansion Notch1 and Fat1 do not seemingly depend on TNF- α signalling. Given the relevance of these mutations and interest of the field, it would be important to define what other molecular traits are shown by these genetic perturbations that could explain their expansion phenotype. Indicating that these may be linked to reduced differentiation without formal validation represents a rather vague description.

15-Figure 4A and downstream analysis of tumour data. The clonal selection of sgRNA perturbations and cell populations involved are compared to a baseline of P60, which is the moment when the

DMBA/TPA treatment start. There is a 12 week difference between the baseline reference time point and the tumour samples. One cannot discard that some of the changes in gene selection are associated to this discrepancy. This should be acknowledged and referred to in the main text.

16-I am rather unconvinced by the way that perturbation selection is normalised in Figure 5C, and extended 8A. The fact that a genetic perturbation is selected in “normal tissue” and continues to be selected under the competitive clonal dynamics of a tumour indicates its potential participation in the process. Hence, this approach should be used to further mine the data for perturbations that are positively selected upon tumorigenesis (when compared to normal adult counterparts, P60). However, those perturbations that are expanded in both conditions (P60 and tumours) should not be disregarded. We do not know enough about mutant clonal dynamics to be able to make this assumption.

17-Please revise the order of the figures so that they are in line with the text. Also make sure that all the figures are mentioned in the text.

Referee #4 (Remarks to the Author):

In this manuscript Renz et al, the authors seek to evaluate the effect of squamous cell carcinoma driver mutations on clonal expansion of normal epidermis cells and oncogenic transformation. To do this the authors developed an in vivo CRISPR screening platform coupled with a single cell transcriptomic read-out. They generated a CROP-seq library against 150 frequently mutated genes in skin squamous cell carcinoma and head and neck squamous cell carcinoma and infected E9.5 embryos in utero to infect the ectoderm with the CROP-seq library and monitored the sgRNA abundance and transcriptional changes underlying clonal expansions at postnatal 4 days and 60 days. The authors identified Notch1, Fat1, Notch2 as the most enriched sgRNAs at P60 relative to control, and identified that indeed the loss of function of Notch1 restricted differentiation rather than increasing proliferation. The authors identified TNF-alpha as a convergent pathway that underlies clonal expansions for many of the perturbations and noted a perturbation-dependent cell type specific enrichment of TNF-alpha. The authors provide further evidence that clonal expansion is dependent on TNF-alpha signaling by repeating the experiment in a Tnfr1 heterozygote and state that for the 20 most perturbed genes, 17 have a Tnfr1 dependency. Furthermore, the authors induced tumorigenesis in their model using DMBA/TPA treatment and profiled 112 tumors. Using single-cell and spatial transcriptomics they map to the invasive front one of the cancer cell populations they annotate as the previously described tumor-specific keratinocyte (TSK, Ji et al., ref 31). The authors make a claim that the TSK state has an autocrine TNFa regulatory loop. When comparing the perturbations that led to clonal expansion in non-cancer to cancer settings, the authors report that an EMT state is enriched in the cancer expansions. Finally, they propose a model whereby TNFa is a state leading to cancer, which is stabilized in an invasive subpopulation with an autocrine loop.

The manuscript makes important and interesting claims. The authors have an elegant and powerful in vivo system for identifying fitness (clonal expansion) effects for specific perturbations. The authors use the results to address an open and important question regarding the relationship between clonal expansion and tumor initiation. The manuscript also has an extensive number of models and draws evidence from numerous experiments.

There however a number of concerns regarding the evidence provided for the claims detailed below.

1. The authors claim that a convergent phenotype of oncogenic mutations is the TNF-alpha signature. The authors provide evidence that the TNF-alpha gene expression program is enriched across different cell types of the epidermis with perturbations relative to control. However, the authors do not show how each perturbation and consequently the degree of clonal expansion relates with the activation of the TNF-alpha gene expression program. The authors may achieve this by ordering Figure 2g-h by their sgRNA representation.

Additionally, the authors claim that they corroborate their WGCNA TNF-alpha gene modules by two alternate approaches - while the first one included GSEA of differentially expressed genes that indeed supports their findings, the second approach simply shows that groups of perturbations have similar transcriptional responses but does not show how these transcriptionally similar perturbations converge on TNF-alpha signaling. (Extended Figure 5).

The observation that a TNFa state is present in many of the perturbations is very interesting. The authors do not make a case however regarding what is its functional role for the cells.

2. In support of the claim that TNF-alpha, they repeat the in vivo single cell CRISPR in Tnfr1 heterozygous mice. The Tnfr1 experiment is important however it is not possible to evaluate its results given the way the authors have reported on it. Extended data figure 3 shows the results for the first experiment however the authors fail to show the analogous results for the Tnfr1 experiment. Instead the authors restrict analysis to the top 20 most enriched genes from the control. Even for these they do not show the actual guide abundances but only a ratio of two ratios (Fig. 3d). Finally in Fig. 3e they compare the ratio of ratios (TNF1-dependency) with a ratio only (P4 enrichment). Thus for this crucial experiment, there is no way to evaluate the results.

Also for the tumorigenesis experiment described in Figure 4 and 5, the results are not shown for all of the guides (as shown in extended data figure 3 for the first experiment).

3. One of the major claims of the manuscript is that the TNF-alpha switches from an immune cell driven to an autocrine TNF-alpha signaling loop. This is supported by applying CellChat, but the authors do not provide any additional analysis or experimental data to support it.

4. The authors claim to identify two distinct subpopulations of TSK, that are differentiated by the expression of MMP10. As the authors cite, this was first identified as a single population in Ji et al., (ref.31). It seems intriguing that the two subpopulations of TSKs appear to be in two distinct clusters. The authors do not show enrichment of the TSK gene signature or module score, representing their annotation of cell types in their dataset to claim that both these clusters of cells are TSKs. This would be critical in understanding the difference between the two TSK sub populations.

Furthermore, the authors project the TSKs to their spatial transcriptomic data, that reveals the MMP10-negative and -positive cell state do not colocalize, while the MM10-negative appears to be at the edges of the tumor and the positive at the core of the tumor, suggesting an inherent difference between the two cell states. It is critical to understand the differences between these cell states, as the authors then use these annotations to claim the presence of autocrine TNF-alpha signaling in TSKs.

Minor Comments:

Please correct this reference as the wrong journal is listed: Ji, A. L. et al. Multimodal Analysis of Composition and Spatial Architecture in Human Squamous Cell Carcinoma. *J. Clean. Prod.* (2020). doi:10.1016/J.CELL.2020.05.039

The authors should clarify in the main text what it means to target a CNV. Are they targeting frequently amplified/deleted regions?

The legend for Extended Figure 4c is hard to understand

The equation to determine TNFR1 dependency (shown in Figure 3) needs to be presented better.

Clarification is required for the difference between Figure 3a and Figure 2g

Clarification is required for why the basal tumor cells are compared to P60 EpSc cells in Figure 5.

Author Rebuttals to Initial Comments:

We thank all four reviewers for their supportive comments and their very constructive suggestions that significantly improved the revised manuscript.

Referees' comments:

Referee #1 (Remarks to the Author):

This is a beautifully written manuscript that represents an enormous amount of work. As far as I am aware this is the first experiment to perform *in vivo* CROP-seq/perturb-seq to interrogate the combined effect of gene disruption whilst also examining cell-type/state at the interface of clonal dynamics.

I have a number of specific comments/questions:

1. A major technical bottleneck in experiments of this kind is getting good representation of gRNAs within the population of infected cells so as to explore their biological effect. The data in Supplementary Figure 1f is opaque to me. Clearly the strong FAT1 gRNAs are enriched but the representation of the control gRNAs does not change over time and is close to zero throughout the experiment. This is a surprise to me as I would have expected gRNAs that have no biological effect to be diluted over time but supp figure 1f is not informative (or is drawn in such a way that the message is unclear – maybe plot absolute normalised read counts instead, or at least provide them). I also find the exact experimental design confusing. i.e. up to 8 mice per litter were injected but how many litters were analysed and how many embryos in total? I can glean from Supp Figure 1i that 8 “samples” were analysed but was this 8 embryos? What was the variation in these experiments? The data in Figure 2a is not presented with any statistical analysis (or error bars) which makes me wonder how to interpret this result. The top 10 enriched and depleted genes are listed but I am unclear what to make of this beyond that several known skin drivers are shown. *Ryr3* (for example) is mainly expressed in muscle (according to GTEX) and has a TPM <1 in skin but is “enriched” in figure 2a. As I read further I see that Supp Fig 1i shows some of the raw data but also suggests there are significant sample/batch effects. i.e. WTA12 and 13 are (almost) always depleted while WTA18 and 17 show little variation.

We thank the reviewer for raising these important points and fully agree that a robust sgRNA representation is a critical aspect of the experimental strategy.

1. In Extended Data Figure 1f, we aimed to demonstrate how stable the control sgRNA library behaves *in vivo* and apologize that the initial presentation was confusing. We have now revised the graph to display normalized read counts for improved clarity, as suggested by the reviewer. The stability of the control library across P4 and P60 indicates the absence of significant fluctuations, which is crucial in an *in vivo* context with different cell types.

2. Regarding the expected dilution of non-effective gRNAs over time, we agree with the reviewer that control sgRNAs would be expected to be diluted over time if the rest of the library was enriched. However, our re-analysis using the MAGeCK pipeline and normalization against the control library revealed a more nuanced picture.

In general, the total percentage of infected cells with control sgRNAs and no biological effect should remain stable and become neither enriched nor depleted in the tissue. However, depending on the library, control sgRNAs could be indeed “diluted over time” if the rest of the library is strongly enriched (relative but not absolute quantification) and if the control library is presented as percentage of the total library.

Using the MAGeCK pipeline and normalizing the P60 library to the control library, we find that about two-thirds of the perturbations showed a positive log2 fold change, while the rest were negative. In addition, some of the depleted perturbations such as for example p63 are very strongly depleted in P60. This overall rather balanced distribution explains why there was no strong dilution of control sgRNAs. We now present these analyses in Supplementary Figure 5a.

3. Regarding the exact experimental design, we now highlight the experimental design also in the M&Ms to allow the reader a better assessment of the experiment.

P4: a total of 58 injected P4 animals were collected and assessed in 7 independent single-cell RNA-seq runs. Because the P4 skin contains less cells, more animals had to be pooled to FAC-sort a sufficient number of mCherry-positive cells.

P60: a total of 10 injected P60 animals were collected and assessed in 8 independent single-cell RNA-seq runs. Again, depending on the percentage of mCherry-positive cells, sometimes 2 animals had to be pooled to FAC-sort enough mCherry-positive cells. This led to the "8 samples" mentioned by the reviewer, corresponding to independent libraries and scRNA-seq runs rather than individual embryos.

This information is now included in the M&Ms under "in vivo experiments".

4. Regarding the variation in these experiments and the statistical analysis and error bars: we have now calculated the weighted standard deviation (where the weight corresponds to the number of cells) in each replicate of the total sgRNAs across the 8 replicates in the mouse P60 skin (related to Figure 2a). We have now added this analysis in Extended Data Figure 1i. We believe this addition provides a clearer interpretation of the variability and statistical significance (and moved the previous plot to the Supplementary Figure 2a due to space restrictions). Beyond the top 10 enriched and depleted in Figure 2a, we also have the full list of perturbations included in Extended Data Figure 3.

5. The reviewer raises an interesting point about Ryr3's enrichment despite its low TPM in the skin. It is correct that the TPM of Ryr3 in the skin is 1.06, although this does not take into account the various cell types of the skin and the potentially higher expression levels in a particular cell type of the skin.

Nevertheless, there is evidence in the literature that Ryr3 indeed could play a role in skin homeostasis. Denda et al. investigated Ryr3 expression in basal keratinocytes and found robust Ryr3 expression in epidermal stem cells and differentiated cells (e.g. by immunofluorescence or rt-PCR in Figure 1 and 2). Moreover, they provided evidence that Ryr gene modulators functionally influence differentiation and epidermal permeability barrier homeostasis, indicating the possibility that Ryr3 could - despite its low TPM - play an important role in skin homeostasis (Denda et al, Journal of Investigative Dermatology, 2012).

6. Concerning the potential batch effect, it is important to outline and discuss the overall context of the experiment.

As the reviewer correctly pointed out and recognizing the critical role of the coverage, we conducted extensive scRNA-seq experiments to ensure adequate sgRNA representation at P4 and P60 stages. However, subdividing these data into 8 separate scRNA-seq runs results in a noticeable reduction in coverage per run. We wanted to be completely transparent with our data and presented the 8 scRNA-seq runs, but this should be cautiously interpreted, as different mCherry percentages could also influence these differences, as outlined below.

The initial infection rate of the sgRNA library plays an important role. With an estimated 120,000 epidermal progenitor cells available for infection at E9.5 (Beronja et al, Nature, 2012), the theoretical coverage per sgRNA could vary significantly based on the infection percentage. For instance, at a 10% infection rate, each sgRNA would infect an average of 24 cells ($120,000 \times 0.1 / 500$), while at a 2% infection rate, this drops to just 4.8 cells per sgRNA. This variation can lead to stochastic differences in the initial representation of sgRNAs in each animal as in low infection rate animals not all sgRNAs may be present, contributing to a potential "batch effect". Given these factors, we would like to stress the importance of considering the entire dataset collectively, which together determine the calculated coverage required for the experimental strategy.

We now include the weighted standard deviation graph in Extended Data Figure 1i. This approach provides a better overview of the variance as it takes the number of cells per scRNA-seq run into account. We moved the previous graph to Supplementary Figure 2a.

7. Finally, we re-analyzed our P60 data using the MAGeCK pipeline, a standard in the CRISPR field. This analysis revealed that 17 perturbations (at the gene level, with $n=3$ sgRNAs each) are indeed significantly enriched, while 9 perturbations showed a significant depletion, as determined by MAGeCK's specific p-value thresholds for enrichment (p high) and depletion (p low). We included the MAGeCK analysis for the 150 perturbations in Extended Data Table 2.

2. The authors make a strong statement at the bottom of the second paragraph of page 5 that the "rates of clonal expansion in their model faithfully recapitulate...that of normal human skin". I don't feel that they provide evidence for this. In reality the "rate" of clonal expansion is a feature of time and clone size and studies in human the lack temporal information required to make such conclusions.

The reviewer is correct.

Initially, our intention was to emphasize that the genes commonly mutated and clonally expanded in human skin are also those we identified as top enriched perturbations in our P60 experimental setup. However, we completely agree that the studies in human skin lack any temporal information and we do not know which mutations were acquired at which time point (and whether different genes show different onsets of mutations).

We therefore removed this sentence.

3. I was surprised to see that Sox10 was pulled out as being a gene involved in a subcluster in supp Figure 4C. Sox10 (as far as I can see) is not displayed in this figure. Sox10 is required for melanocyte survival so I would expect its disruption in these cells to be lethal.

We apologize for the confusion. In Extended Data Figure 4c, we present the 44 modules identified by the WGCNA strategy and the 150 perturbations. Sox10 is a module gene present in several modules, defining the melanocyte population within these cell types, along with other melanocyte-specific genes such as Dct, Tyrp1 or Kit. As mentioned in the text and also as seen in other studies using WGCNA, the analysis can also reveal subclusters within clusters, such as melanocytes that do not have their own separate cluster in P60.

(Page 7: Besides this major gene cluster, we found many modules that reflected subclusters present in our major cell clusters, such as a Dct, Tyrp1, Sox10 module in multiple clusters reflecting melanocytes within these cell populations (Extended Data Figure 4c).

We have now added this information to the Extended Data Figure 4c legend to clarify this point further and revised the figure legend.

4. I would suggest the authors do a multi-variate analysis in Figure 5o and replicate the signature across other datasets. While I think the biological hypothesis is sound an analysis using a heterogeneous cohort such as TCGA can be problematic. For example, what happens if the HPV-driven cases and those that are not HPV-associated are analysed separately? These subtypes are markedly different in terms of mutational load and driver landscape, the former lacking FAT1 and TP53 mutations, for example. Similarly, HNSCC can be classified as atypical, classical, mesenchymal and basal with notable differences in mutation profile and expression and by definition histopathological appearance. How does this influence the association between the TSK signature and outcome? HPV-associated disease is more common in men. How does sex influence the TSK signature?

This is an excellent suggestion.

We now conduct a multivariate Cox regression analysis for HNSC, incorporating the mean expression of the 11 TSK genes, along with sex, age, and HNSC mRNA subtypes (atypical, classical, mesenchymal, and basal) as described in the referenced study (doi:10.1038/nature14129). This approach now allows us to control for these variables. The results from this adjusted analysis still indicate a significant association, with a Hazard Ratio of 1.23 (P: 0.013). This suggests that the mean expression of the TSK genes remains a significant factor in HNSC prognosis, even after accounting for these covariates.

Furthermore, including an additional covariate for HPV status (HPV+ or HPV-) in the analysis yielded a similar Hazard Ratio of 1.20 (P: 0.043), reinforcing the robustness of our findings against these potential confounders.

We now mention the multivariate analysis in the main text (Page 14: Multivariate analysis confirmed that the TSK signature is an independent prognostic factor for overall survival (Extended Data Figure 10d-e) and included these interesting analyses in Extended Data Figure 10d-e.

5. The authors note by citing reference 29 that it has long been established that the TNF-alpha pathway is implicated in SCC tumour initiation and loss of TNFR1/2 severely decreases pre-malignant (papilloma) tumour initiation and growth in classical mouse carcinogen models such as DMBA/TPA. Indeed, the experiments using *Tnfr1* knockouts in this paper are an extension of this original work. I think the authors should more actively acknowledge this prior work.

We agree with the reviewer.

To acknowledge the seminal work by the Khavari and Balkwill labs more actively, we extended this paragraph and added additional references.

Revised paragraph:

TNF- α signaling directly confers clonal expansion

Intracellular TNF- α signaling is mainly activated by binding of the TNF- α ligand to the TNF receptor 1 (TNFR1), which induces several signal transduction arms, including c-Jun and nuclear factor- κ B (NF- κ B) activation^{27,28}. These transcription factors are responsible for diverse biological processes, including cell growth, immune and stress responses. Previous studies have demonstrated the involvement of *Tnfr1*, *Jnk* and *NF- κ B* as proliferative signal for controlling epidermal growth²⁹. Furthermore, *Tnfr1* and *Tnf- α* knockout mice are resistant to skin carcinogenesis³⁰. Moreover, TNF- α

signaling has been implicated in clonal hematopoiesis in the context of aging and myeloproliferative neoplasms^{31,32}, indicating that the TNF- α signaling module could be directly mediating clonal expansions.

Minor point

I think the supplementary table "500 sgRNAs" is truncated from line 452 onwards. From this file it's not immediately clear to me which of the gRNAs are controls. Further the gRNA names are not present in the additional tabs with the counts. This makes it very hard for me to follow (or replicate what was done without re-writing some code myself). What does library_A, _B, _C represent? I can't follow from the methods. How do these counts relate back to the data in (for example) Figure 2A. Are they normalised counts or raw counts? The supp tables don't have titles or explainers to help and I can't see any table legends. I would strongly suggest that all of the analyses are presented as Jupyter notebooks (or similar) so that readers can follow what was done.

We apologize for the corrupted file. We now fixed the table and included the full list of 500 sgRNAs. Additionally, an info tab has been added to clarify the terminology used, such as 'library_A'. This updated table can be found in Extended Data Table 9.

Furthermore, we now include an R Markdown notebook to replicate the main Figures 1-5. We also include accompanying codes and information on the processing of the single-cell CRISPR and single-cell RNA sequencing data processing on our github page (link provided in the Material & Methods section), which allows the reader to reproduce our results.

This information and the R Markdown notebook are included under the following link:
https://github.com/sendoellab/single-cell_CRISPR.

Referee #2 (Remarks to the Author):

In their manuscript, Lenz and colleagues study how clonal expansion influences malignant transformation in the epidermis. Using a targeted CRISPR/Cas-based approach, the authors delete 150 commonly mutated cancer genes (mainly tumor suppressors) in the developing epidermis. They then follow the clonal behavior of the perturbed cells in neonatal and adult skin. As expected, they find that cancer-related gene losses, such as p53, Notch1/2, and Fat1, correlate with clonal expansion. A commonly up-regulated pathway in cancer gene perturbed epidermal cells is TNF-alpha signaling that promotes clonal expansion, yet is down-regulated in early stages of tumorigenesis, and specifically up-regulated at the tumor invasive front.

The study addresses the question how clonal expansion in normal epithelia shape or drive malignant transformation. The novelty of this study is that potential driving genetic events are induced in all cell types. Targeting the commonly perturbed genes in squamous cell carcinoma in development and following the long-term effects in an interesting approach that revealed novel insights into the role of clonal expansion to malignant transformations. My major concern with this study is that the authors draw firm conclusions from largely correlative data. The authors claim that TNF-alpha signaling plays different roles during clonal expansion, early stages of tumorigenesis and at the invasive front. As a major regulator of inflammatory responses, TNF-alpha is known to regulate cell death and survival and increasing expression of angiogenic factors. Thus, it remains unclear whether changes in TNF-alpha signaling are cause or consequence of perturbed cancer gene expression.

Major Comments:

1. Why is it required to sort the mCherry positive cells as opposed to just perform single cell sequencing of the whole skin, which would have the advantage to analyse cell-cell interaction

between perturbed and non-perturbed cells and quantify clone size? Presumably this is done because the number of mCherry-positive cells is too low. In that case, the authors should comment on the efficiency of the *in vivo* injection method.

Thank you for raising these important points. There are a number of points we would like to clarify here:

1. The low number of mCherry-positive cells is by design, not due to any inefficiency in our *in vivo* injection method. Our high-titer lentivirus pipeline and injection method is highly efficient and we could also target >95% of epidermal cells if needed. However, we deliberately chose low infection rates to minimize the likelihood of double infections. With 15% infection rates, we know that the double infection rates are < 1% (Beronja et al, Nature, 2012). Therefore, to ensure that we have a very low double infection rate, our experimental strategy requires the infection of a low percentage of mCherry-positive cells and FAC-sorting of mCherry cells to compile scRNA-sequencing libraries.

2. The reason why we sort mCherry-positive cells is also driven by the cost associated with single-cell RNA sequencing. The costs for achieving adequate coverage in scRNA sequencing are significant, and it was imperative to manage these expenses. In addition, it also serves as a quality control to sort viable cells and ensures that we can exclude by FACS any potentially dying cells.

3. Although an intriguing question, identifying specific interactions between mCherry-positive cells and others, especially regarding their perturbations, is beyond the scope of our current methodology. Even if we performed single-cell RNA sequencing of the whole skin, it would not be possible to analyze cell-cell interactions, as there is no possibility with our experimental strategy to detect which cells were in contact with mCherry-positive cells (and which sgRNA they expressed). This interesting area of research would require a different strategy, as suggested for example in the study by Ombrato et al. (Nature, 2019).

4. Regarding the comment that “the authors draw firm conclusions from largely correlative data”, it is important to emphasize the breadth of our study. Our manuscript encompasses an extensive *in vivo* data set, comprising 150 gene knockout conditions in 9 different cell types over 2 time points. These data allow us to scrutinize the direct consequences of each of these 150 knockout conditions at single-cell transcriptomic resolution. We use these extensive data sets to distill the pathways shared by multiple cancer gene mutations involved in clonal expansions in phenotypically normal tissues. However, these changes represent direct evidence for transcriptional changes induced by groups of perturbations, rather than mere correlation. Moreover, we now bolster the TNFR1-dependency of clonal expansions by additional experiments using mouse knockout models (Figure 3d), determine the role of immune cells in clonal expansion by immune depletion experiments (Figure 3f-h) and conducted additional characterization of the *in vivo* TNF- α expression experiments to confirm the direct effect on tissue invasion.

Regarding the “firm conclusions”, we agree with the reviewer and toned down several sentences throughout the manuscript.

2. The enrichment of sgRNA targeting only one gene (Fat1) is shown (EDF1f). However, the design of the experiment would predict that nearly all sgRNAs should lead be enriched at P60. The authors should show the enrichment of all sgRNAs to see how many of the targeted genes caused indeed a clonal expansion, and the authors should show which sgRNAs are significantly enriched or depleted and include the information whether the average enrichment was shown (n=3 sgRNAs) or single guide RNAs.

We appreciate the opportunity to clarify several key aspects of our study:

1. The selection of sgRNAs was based on the 150 most mutated genes in human skin and head and neck SCCs. It is important to note that these genes are frequently mutated in cancer, but we have no evidence to expect that the cohort of genes also necessarily results in enrichment in normal, pre-cancerous epithelia at P60. In addition to cancer gene mutations, the study also encompasses 16 copy number aberrations, 15 of which are amplifications. These aberrations, including genes like p63, are expected to be strongly depleted rather than enriched. Together, we would not necessarily predict that all sgRNAs should be enriched at P60, but rather the study aims at determining the role of these cancer mutations in pre-cancerous normal tissues.

2. The reviewer raises a good question regarding the enrichment or depletion of perturbations. We now have undertaken a systematic comparison against a baseline control library comprising 50 sgRNAs (as opposed to normalizing over the whole library). By implementing the MAGeCK pipeline, a standard in CRISPR screen analysis, we normalized our perturbations against the 50 control sgRNAs. This analysis has allowed us to assess directly whether a perturbation is enriched/depleted compared to the 50 nontargeting control sgRNAs. We found that about two-thirds of the perturbations showed a positive log₂ fold change, while the remaining one-third were negative. We now included these analyses in Supplementary Figure 5a. Given the notion in point 1, these analyses enable a more nuanced view of the cancer gene perturbations in normal epithelia and indicate that only about two thirds of our library are indeed positively selected in P60 skin.

3. In Extended Data Figure 1f, our goal was to demonstrate how stable the control sgRNA library behaves *in vivo*. The enriched example of *Fat1* originally included was intended as a comparative reference. In response to the need for clearer demonstration, we have revised and simplified this figure and now use normalized read counts. Additionally, we have included three perturbations (combined 3 sgRNA for each perturbation) as "positive" controls to show examples of enrichment of sgRNAs.

4. We appreciate the reviewer's suggestion to delineate the perturbations exhibiting significant enrichment and depletion. Accordingly, we re-analyzed our data using the MAGeCK pipeline, a standard in the CRISPR field. This analysis revealed that 17 perturbations (at the gene level, with n=3 sgRNAs each) are significantly enriched, while 9 show a significant depletion, as determined by MAGeCK's specific p-value thresholds for enrichment (p high) and depletion (p low).

As a side note, to assess the robustness of our findings, we also conducted MAGeCK analysis on individual sgRNAs. This revealed consistent results across sgRNAs for key perturbations. For example, all 3 sgRNAs were significantly altered for *Notch1*, *p53*, and *p63* and 2 sgRNAs were significantly altered for *Fat1*, underscoring the high reproducibility of our results.

In line with the reviewer's recommendation, we have now incorporated these detailed MAGeCK analyses in Extended Data Table 2 for a comprehensive overview of our findings for the reader.

3. The finding that TNF-alpha signaling pathways are enriched in expansion-driven perturbations is interesting but not entirely unexpected given that pro-inflammatory cytokines are commonly induced in damaged tissues and neoplasms. TNF-alpha has also been linked to clonal expansion in myelodysplastic syndromes (Abegunde et al, 2018; Fleischman et al, 2011) and the authors should acknowledge these findings.

We thank the reviewer for bringing these interesting publications to our attention. We now acknowledge these findings and cite these two publications.

Revised paragraph:

TNF- α signaling directly confers clonal expansions

Intracellular TNF- α signaling is mainly activated by binding of the TNF- α ligand to the TNF receptor 1 (TNFR1), which induces several signal transduction arms, including c-Jun and nuclear factor- κ B (NF- κ B) activation. These transcription factors are responsible for diverse biological processes, including cell growth, immune and stress responses. Previous studies have demonstrated the involvement of *Tnfr1*, *Jnk* and *NF- κ B* as proliferative signal for controlling epidermal growth. Furthermore, *Tnfr1* and *Tnf- α* knockout mice are resistant to skin carcinogenesis. Moreover, TNF- α signaling has been implicated in clonal hematopoiesis in the context of aging and myeloproliferative neoplasms (Abegunde et al, 2018; Fleischman et al, 2011), indicating that the TNF- α signaling module could be directly mediating clonal expansions.

4. It remains unclear whether the up-regulation of the TNF-alpha signaling pathway is directly caused by the loss of the respective genes or indirectly by for instance enhanced cell death of the affected cell populations. This could be addressed by testing whether the TNF-alpha signaling is commonly up-regulated in squamous cell carcinoma patients (with the respective mutations) and whether it correlates with disease stage.

This is an excellent suggestion. As indicated by the reviewer, we now systematically tested whether TNF-alpha signaling is commonly upregulated in human squamous cell carcinoma patients.

1. Our working model suggests that conventional TNF- α signaling confers clonal expansion in normal epithelia. While conventional TNF- α signaling is downregulated in tumorigenesis, an autocrine TNF- α gene program is upregulated in specific cellular populations in squamous cell carcinomas. To address the reviewer's suggestion, we therefore subdivided the analysis into TNF- α module genes (as defined in Figure 2h, either assessing the 16 TNF- α pathway genes or the 24 complete TNF- α module genes) and autocrine TNF- α /TSK signaling genes (as defined by the TSK genes used in Figure 5o and in the human SCC paper defining TSKs by the Khavari lab).

Using these two lists, we documented the expression of the two distinct TNF- α signaling pathways in healthy skin, actinic keratosis (pre-cancerous) and skin squamous cell carcinomas. As a second cancer type, we also included healthy head and neck tissues and head and neck squamous cell carcinomas (HNSCC), subdivided into 4 stages (to address the reviewer's suggestion whether the expression correlates with disease stage).

2. TNF- α module genes are also downregulated in skin SCCs compared to actinic keratosis or healthy skin. In contrast, TSK genes are upregulated in skin SCCs compared to actinic keratosis or healthy skin, with a trend that actinic pre-cancerous keratosis exhibits already higher TSK levels (albeit without reaching significance). As a side note, "conventional" TNF- α signaling was the top downregulated pathway in skin SCC patients compared to healthy tissue, an analysis that was already included in Extended Data Table 4 (now Extended Data Table 7).

3. In line with these observations, TNF- α module genes are downregulated in head and neck SCCs compared to healthy head and neck tissue. Interestingly, this TNF- α module downregulation indeed significantly correlates with disease stage, with a stepwise downregulation from HNSCC stage 1-3. In contrast, TSK gene levels are strongly increased in HNSCC patients compared to normal tissue ($p < 2.2e-16$). There is also a clear trend toward an increase in TSK gene expression with advanced disease stage, albeit without reaching the significance threshold (e.g. stage 3 compared to stage 2, $p=0.059$). Given that TSK genes are likely only expressed in a subset of cancer cells, future studies may elucidate the cell-type-specific expression levels of TSK genes in more detail.

In summary, our analyses in human cancer samples corroborate our working model, suggesting that there is a downregulation of the TNF- α signaling module in squamous cell carcinomas. This downregulation is coupled with an emerging distinct TNF- α -TSK gene program as cancer progresses.

These interesting observations have now been integrated into our study in Extended Data Figure 6f and Supplementary Figure 8.

5. The authors show that the TNF-alpha signaling pathway is up-regulated in about 30% of sgRNAs but they do not show that the respective perturbed genes indeed consistently cause clonal expansion.

The reviewer is correct. To address the reviewer's comment, we included two additional sets of experiments and additional analyses.

1. We believe the reviewer is referring to Figure 2h, which is the heatmap of TNF- α module genes and their differential expression following the 150 cancer gene perturbations. From this heatmap, the reviewer is correct and one could state that approx. a third of these perturbations result in upregulation of the TNF- α module genes. To link this observation more closely with clonal expansion, we took advantage of the hierarchical clustering of the TNF- α module genes and perturbations and analyzed the 4 major clusters further. We found that the two clusters with high TNF- α module gene expression (cluster 1 and 2) exhibit significantly higher rates of clonal expansion than the other two clusters (new Figure 2h-i). This additional analysis will help the reader to link early on TNF- α module gene expression and clonal expansion.

2. To assess the role of ectopic TNF- α and epithelial TNF- α expression on keratinocyte proliferation, we either treated keratinocytes with TNF- α or expressed a lentiviral TNF- α construct in keratinocytes. We then analyzed proliferation rates and observed that both conditions increased proliferation of keratinocytes. We included these data in Extended Data Figure 9c.

3. Finally, to directly assess whether any TNF- α module genes cause clonal expansion, we now experimentally tested the role of select TNF- α module genes in keratinocytes. Previous work from the Khavari lab had already implicated TNF- α downstream signaling in skin hyperproliferation, where they nicely demonstrated that TNFR1-dependent JNK activation conferred skin hyperproliferation (Zhang et al, Genes Dev, 2004).

We have now experimentally tested the role of three TNF- α module genes on keratinocyte proliferation. We overexpressed the TNF- α module genes *Jun*, *Fos* and *Ccn1* in keratinocytes using lentiviral constructs and carefully assessed proliferation rates in control infected and TNF- α module gene-expressing cells. We observed *Jun* and *Fos* expression indeed enhanced keratinocyte proliferation. *Ccn1* expression increased proliferative rates, albeit without reaching the significance threshold. These experiments confirm that increased expression of TNF- α module genes can indeed directly confer keratinocyte proliferation. We now include these data in Extended Data Figure 4f.

6. Page 8, first paragraph: The data and analyses shown do not warrant the conclusion that 'clonal expansions may need to be coupled to EMT downregulation to ensure that the expanding clones confine to the physiological tissue architecture of the epidermis.' The authors' statement indicates that the down-regulation of EMT genes is an active process to maintain tissue architecture, yet they provide no evidence. The analyse was biased to define expanding epidermal cell populations and dividing cells cannot migrate. Thus, the down-regulation of some ECM genes might simply be the consequence of enhanced division.

We agree with the reviewer and have now removed this sentence.

7. I am not convinced that the authors unequivocally showed that TNF-alpha signaling directly confers clonal expansion. TNF-alpha could be up-regulated due to cell damage or death caused by another sgRNA targeting for example p63. The loss of cells will be compensated by neighboring cells, always expand clones and give proliferation enhancing sgRNAs a competitive advantage. In that case, the clonal expansion is not directly caused by TNF-alpha but sgRNAs causing cell death close by. This would also explain the down-regulation of TNF-alpha modulated genes in early stages of tumorigenesis.

To address this interesting point by the reviewer, we would like to highlight a number of points:

1. There is indeed evidence for density-dependent proliferation in the mouse skin, indicating that in lower density regions, neighboring cells could compensate for this drop in density and proliferate into this region (e.g. reviewed by McClatchey & Yap, 2012). However, this is a local mechanism and mediated by neighboring cells and likely includes cell-cell contacts. The likelihood of such a potential mechanism occurring is the same for a control sgRNA as for Notch1 sgRNA. In addition, given that our library includes 50 controls sgRNA, the likelihood that p63 sgRNA has a neighboring control sgRNA is 16.6x higher than a Notch1 sgRNA ($50/3=16.66$). Therefore, in principle, it is rather unlikely that such a mechanism would favor for example Notch1 sgRNAs, especially given that Notch1 knockout does not necessarily lead to higher proliferation rates but rather alters differentiation dynamics (Figure 2b).

2. We also have the advantage to assess two time points in our data set, P60 and P4. p63 sgRNAs are already highly depleted at P4 stage and there are very few p63 sgRNA cells detectable at P4. However, between P4 and P60, *Notch1* and other enriched sgRNAs are still readily expanding in the absence of such a potential cell death-dependent mechanism.

3. The TNF- α signaling module has been identified by the WGCNA strategy in P60 epidermal stem cells. In P60 skin, sgRNAs like p63 are already strongly depleted, many of which even before P4 (Extended Data Figure 3b). Therefore, dying cells could not contribute anymore to the identification of a TNF- α signaling module in P60 epidermal stem cells. This suggests that the induction of TNF- α signaling is independent of any depleted sgRNAs, underscoring a separate regulatory mechanism at play in these cells.

Together, based on the experimental design and the advantage of exploiting two time points in clonal expansion, these observations speak against the interesting notion that cell death is the primary driver of TNF- α signaling upregulation.

8. Without any information about initial clone sizes, little conclusion can be drawn by the data shown in Figure 5a-e. A larger clone is per se more likely to be hit by another mutation.

We apologize for the confusion with Figure 5 and we appreciate the opportunity to revise Figure 5 to make it clearer for the reader.

We fully agree with the reviewer, and we indeed had normalized our tumor data in Figure 5c to the initial clonal size at P60. As the reviewer indicates, this is a very important point because the P60 clone size is the baseline when starting the DMBA/TPA treatment. A larger clone is indeed more likely to be hit by a DMBA/TPA-induced mutation.

To ensure the clarity of this key point, we have revised Figures 5a-d, updated the schematic and labelling and clarified Figure 5c, with the aim to present these aspects more transparently.

Minor Comments:

1. Three sgRNAs per gene is quite low in particular for in vivo screens. The authors need to comment on how they reached the decision that three sgRNAs are sufficient in that case.

This is an important question regarding our choice of using three sgRNAs per gene in our screen. This decision was carefully considered, balancing costs, methodological rigor and practical constraints. Our experience with the Doench et al. library, which contains four sgRNAs per gene, has shown high editing efficiency in both keratinocytes and *in vivo* mouse skin, usually across all four sgRNAs. This was further validated by our efficiency tests showing reliability in editing, with 67-87% editing efficiency (Extended Data Figure 1e). Importantly, we observed that the effects of different sgRNAs are consistent, as evidenced by significant changes in sgRNA representation for all three sgRNAs in key genes such as *Notch1*, *p53*, and *p63*.

Cost considerations also played a role in our decision. While in vitro pooled CRISPR screens can accommodate larger numbers of sgRNAs without significantly impacting costs, in vivo single-cell CRISPR approaches entail markedly higher expenses. Achieving adequate coverage in CRISPR screens, which is essential for robust analysis, necessitates extensive single-cell RNA sequencing, thereby escalating costs. Moreover, we prioritized including a large control sgRNA library for a solid baseline and maintaining the P4 time point as a critical postnatal reference time point.

Given the high consistency observed across the four sgRNAs and the stable, replicable outcomes of the perturbations, we therefore chose to include 3 sgRNAs per gene and prioritize high sgRNA coverage, a robust control library and the additional time point.

Referee #3 (Remarks to the Author):

Nature manuscript 2023-06-10719

Title: "In vivo single-cell CRISPR uncovers distinct TNF- α programs in clonal expansion and tumorigenesis" by Renz et. al.

In the last 8 years, since the seminal work by Martincorena et.al. (Science 2015), mounting evidence has shown how our seemingly normal tissues accumulate somatic mutations throughout life, becoming a patchwork of mutant clones that compete with each other. Interestingly, despite the emergence of numerous mutations, including in cancer-associated genes, tissues remain healthy at large, suggesting that the nature of the mutations present in a particular tissue does not represent the sole cause of cancer. This raises a critical question "what are the processes that determine the predisposition of particular mutant clones to transform into cancer?".

The authors of the current manuscript are framing their work in this context, studying one of the most important and timely questions in the early cancer field. In particular, the authors aim at identifying common mechanisms dictating mutant clonal expansion under normal conditions (developmental and homeostatic) in the mouse skin, and go one step further by investigating how these change during carcinogenesis. The research questions covered in this manuscript are of significant relevance in the cancer field, offering the opportunity to understand the largely unclear factors defining mutant clonal predisposition to develop cancer; an aspect of paramount importance to develop more effective strategies to prevent and treat cancer.

What makes this study stand out is the elegant in vivo model used to address the ambitious question at task. The authors adapt the in utero lentiviral transduction method to develop an in vivo single-cell CRISPR screen that enables them to investigate clonal dynamics of the 150 most frequently mutated squamous cell carcinoma genes at the same time and in vivo. By implementing the previously

published CROP-seq approach (Datlinger et. al. Nat Methods 2017) in an in vivo setting the authors are able to infer the transcriptional signature of individual genetic perturbations at the whole-tissue level, which is rather impressive. They further adapt the CROP-seq technology by incorporating a qualitative lineage tracing approach (lentiviral mCherry expression) to be able to sort the cells bearing genetic perturbations, and potentially track their fate.

Using this innovative approach, the authors reproduce previously published data in human skin (Martincorena et.al. Science 2015) showing that genetic perturbations in genes including *Fat1*, *Notch1*, *Trp53* and *Notch2* become enriched in adult mouse skin. This reflects their characteristic clonal expansion and validates the robustness of the approach used. Additionally, this study provides evidence of perturbations in other less characterised genes that also undergo clonal expansion/depletion, adding value to what it is already known.

The key aspect of this paper lays in the ability to explore genetic programs shared by multiple cancer gene perturbations. The authors conclude that the top expansion-driving perturbations exhibited gene expression changes associated with the activation of a non-cell autonomous TNF- α signalling, potentially through macrophage-epithelial communication. Interestingly, during tumour initiation they define a signalling switch, with TNF- α gene programme apparently becoming more cell-autonomous and rather specific to a tumour cell subpopulation (TSKs). Ultimately, it is proposed that, in line with previous reports, this population is marking invasive tumour fronts and that the rewiring of the TNF- α signalling during carcinogenesis is a key step in tumour invasion.

Although the work presented here is interesting, and the question of high relevance, the presented conclusions are rather descriptive, overstated, and lack evidence to support some of the strong and definite claims made. Additional data and analysis will be needed to merit further consideration. See detailed comments below.

Major points

1-One of most relevant aspects of this paper lays in the ability to infer gene programs associated with particular genetic perturbations by combining sgRNA screening with scRNA-seq in vivo. This represents an innovative approach to identify the molecular mechanisms regulating mutant clonal expansion in a living tissue. Using this method, the authors conclude that “TNF- α signaling plays a universal role in mediating clonal expansions in normal epithelia”. However additional functional validation would be needed to substantiate this rather overemphasised statement. I have a few questions and concerns that would need to be resolved.

-As it stands, the authors mainly support the role of TNF- α clonal expansion in the form of a *Tnfr1* KO experiment for which very limited data is presented. They choose to base their validation on TNFR1 dependency of the top 20 enriched genes at P4 (Figure 3E). Indeed, one would not expect that clonal dynamics is modulated in the same manner in adult and postnatal stages. Hence, validating the TNF- α module, identified at P60, using *Tnfr1* KO mice at P4 seems rather unconventional. The authors should use the same time point for transcriptional characterization and validation experiments.

The reviewer is absolutely correct.

Therefore, to directly address this important point, we carried out a new set of experiments to address the role of TNFR1 in clonal expansions at P60 skin. We re-injected our library of 500 sgRNAs into *Cas9; Tnfr1 +/-* knockout animals and collected the skin of P60 animals. We then created single-cell suspensions and performed amplicon sequencing of sgRNAs.

Taking the same approach as for the previous P4 analysis, we calculated TNFR1-dependency for each perturbation and now include the P60 TNFR1-dependency data in the main Figure 3d. We now

also show the complete set of 150 perturbations and their TNFR-1-dependency as suggested by the reviewer, and not just the top 20.

There are several interesting aspects of the new P60 data set analysis.

1. Our complete P60 data set clearly indicates a pronounced TNFR1-dependency primarily in the enriched perturbations. This is evident from the locally weighted regression curve (red line, Figure 3d), which distinctly shows the TNFR1-dependency for these enriched perturbations on the left side of the graph. Conversely, for non-enriched perturbations (middle and right side of the graph), the TNFR1-dependency diminishes, returning basically to zero. This stark contrast underscores the specificity of the TNFR1-dependency in relation to perturbation enrichment. We have also noted that for highly depleted perturbations like p63, where cell counts are very low or almost lost, as expected the data becomes less reliable due to increased noise in ratios.

2. As pointed out by the reviewer, clonal dynamics are indeed not modulated in the exact same manner in adult and postnatal stages. While the overall pattern of TNFR1-dependency remains broadly consistent for the enriched perturbations between P4 and P60, there are two notable and important exceptions: *Fat1* and *Notch1* now also exhibit a clear TNFR-1 dependency at P60. This finding at P60, differing from the postnatal stage, underscores the importance of developmental context in the modulation of clonal dynamics. This reinforces the value of including data from the more accurate P60 time point for a comprehensive understanding.

3. To include statistical analyses, we utilized the MAGeCK pipeline, a standard and robust method for CRISPR screen analysis, to scrutinize our control and *Tnfr1* +/- knockout data set. From the top 20 enriched perturbations, 17 perturbations show significant False Discovery Rates (FDR), underscoring that the majority of the top enriched perturbations at P60 indeed show a strong TNFR1-dependency.

This new set of experiments is now included in Figure 3d.

4. To avoid overemphasizing TNF- α , we now also included the information about other enriched pathways detected in clonal expansion.

For example, page 8: *"In each of these comparisons, TNF- α signaling was the most enriched pathway among the upregulated genes, followed by a notable enrichment in hypoxia and p53 signaling (Figure 2m)."*

In general, we revised a number of sentences to avoid overstating the role of TNF- α signaling. We also toned down the sentence mentioned by the reviewer on page 9: *"TNF- α signaling plays a universal role in mediating clonal expansions in normal epithelia"* to *"TNF- α signaling plays a **widespread** role in mediating clonal expansions in normal epithelia"*.

-Despite the power of their experimental set up, from which they can infer perturbation enrichment and clonal data through mCherry, the authors do not present any images and/or quantitative data showing changes in clonal expansion in *Tnfr1* KO skin wholemounts. Although, I am aware that not all the enriched genetic perturbations showed TNF- α dependency, if "TNF- α signalling plays a universal role in mediating clonal expansions in normal epithelia", then one would expect to see a significant shift in the size of the clones present in the KO tissue at P60 (rather than P4).

This is a completely valid point. We have now repeated the lentiviral injections of our sgRNA library into *Tnfr1* knockout mice and carried out wholemount imaging to monitor the mCherry clones.

1. We now present additional wholemout staining in *Tnfr1* +/- knockout P60 skin. Although there is a clear trend toward smaller clones in *Tnfr1* +/- knockout compared to control P60 skin, we must be cautious with the interpretation. The issue is that without a multicolor analysis to confirm the monoclonality of mCherry areas, we can technically not call them “clones” and cannot be sure whether these areas are indeed monoclonal areas, which complicates the quantifications. We also do not know which mCherry clones are controls, representing 10% of the library. These issues complicate the comparison of clones in control vs. *Tnfr1* +/- knockout skin at the level of immunofluorescence/wholemouts and can be quantified in a much more exact manner in the analyses outlined under 3. Nevertheless, our quantifications assuming the clones are indeed monoclonal origin suggest that mCherry clones in control P60 are indeed larger compared to *Tnfr1* +/- knockout skin, however, we want to be very cautious with drawing any conclusions from these quantifications (Reviewer Figure).

Reviewer Figure. mCherry area quantification in *Tnfr1* +/- animals.

Reviewer Figure.
a, mCherry area was quantified in *Tnfr1*+/+ and *Tnfr1*+/- animals injected with the library comprising 500 sgRNAs. *Please note that in the absence of multicolor analysis, we cannot definitively ascertain that the observed areas are indeed monoclonal. P value indicates a Wilcoxon test. N = 3 *Tnfr1*+/- animals, 2*Tnfr1*+/+ animals.

The representative wholemout staining is included in Supplementary Figure 5c.

2. Additionally, we administered injections of our 500 sgRNA library into both wild-type and *Tnfr1* +/- knockout mice at E9.5. However, in this experiment, we allowed the clones to expand over a substantially longer period, nearly 6 months, reaching P115. This was done with the idea that any potential role of TNFR1 in clonal expansion might become more visually apparent over an extended timeframe. As a side note, these animals were injected on the same day with the same virus. At the P115 stage, we harvested the entire back skin for analysis, capturing images of the approximately 7x4.5 cm area using the Nightsea DFP-1 laser and a camera. Again, we want to be cautious with any quantification of the area, but it is evident that the maximum mCherry-positive areas were clearly much smaller in the *Tnfr1* +/- knockout mice as compared to their wild-type littermates.

The results of this experiment are included in Supplementary Figure 5d.

3. Regarding quantitative data, we think that the best possible quantification is indeed the sgRNA representation in control vs. *Tnfr1* +/- knockout skin, normalized to the control sgRNA library (Figure 3d, Supplementary Figure 5a-b). We now also include additional analyses of wild-type and *Tnfr1* +/- knockout skin and assess P60 representation of the 150 perturbations using MAGeCK. In wild-type P60 skin, 99 perturbations showed a positive log₂ fold change compared to the 50 control sgRNA library, whereas in *Tnfr1* +/- knockout animals, only 46 perturbations exhibited a positive log₂ fold change. We also find that the distribution of enriched perturbations is tilted, with only 3 perturbations with log₂ fold change > 1 (in stark contrast to the wild-type skin). Together, these observations

suggest that clone sizes of perturbations enriched in the wild-type skin are indeed clearly shifted and reduced in *Tnfr1* +/- knockout skin.

These analyses are included in Supplementary Figure 5a-b.

-Additionally, it would be interesting to obtain information on how perturbation enrichment is affected within a *Tnfr1* KO background, rather than just focusing on the top 20 enriched genes at P4 (Figure 3e). Do different genetic perturbations become enriched in *Tnfr1* KO, is this tilting expansion towards mutations that may be “less aggressive”? That would be interesting in its own right.

This is an intriguing idea.

To address the reviewer’s question whether different perturbations become enriched in *Tnfr1* +/- knockout skin and whether these perturbations are “less aggressive”, we assessed again the complete set of perturbations in wild-type and *Tnfr1* +/- knockout P60 skin.

Although there is no perfect readout for “aggressiveness” of a mutation in skin SCCs, we used mutation rates in skin SCCs (new project GENIE database) as a proxy for aggressiveness. We first calculated the mutation rates of these genes from project Genie in wild-type and *Tnfr1* +/- knockout skin (first and second graphs). We then compared the distribution of “aggressive” perturbations in control vs. *Tnfr1* +/- knockout skin in a GSEA, with strongly TNFR1-dependent perturbations on the left side and TNFR1-independent perturbations on the right side. We found that wild-type P60 has a significant enrichment of aggressive perturbation (normalized enrichment score NES of 1.45, FDR = 0.026). These data indeed indicate that the expansion of perturbations in *Tnfr1* +/- knockout skin is significantly tilted toward less aggressive perturbations.

We included these interesting analyses in Supplementary Figure 6.

-The data suggests that the TNF- α stimulation is via a non-cell autonomous (potentially immune cells, in particular macrophages). However, the authors do not present any quantification of immune cells or macrophages within the mCherry areas of control versus *Tnfr1* KO mice, to show an increase of immune recruitment. Nor they perform immune depletion experiments to prove the non-cell autonomous notion or validate the contribution of a particular immune population in the process. This is an important mechanistic aspect given that later on the authors suggest that TNF- α switches from paracrine to autocrine during tumorigenesis.

This is an excellent suggestion by the reviewer to include immune depletion experiments.

To explore the role of immune cells in clonal expansion, we assessed the rate of clonal expansions by the 150 perturbations upon immune depletion experiments. We re-injected animals with the library of 500 sgRNAs into E9.5 embryos and subsequently divided the animals into three experimental groups from the P4 stage on:

Experimental group 1: Macrophage depletion by i.p. injecting CSF1R/CD115 antibodies twice a week.

Experimental group 2: T cell depletion by i.p. injecting CD4/CD8 antibodies.

Experimental group 3: Control arm by i.p. injecting IgG2a isotype control antibodies.

At P60, we harvested back skin from the animals, prepared single-cell suspensions, and conducted amplicon sequencing of the sgRNAs to evaluate their representation in the different experimental groups. This strategy parallels the new set of *Tnfr1* knockout experiments at the P60 stage.

We carried out these experiments in two independent runs, including 14 amplicon sequencing runs from 7 animals in control and macrophage depleted samples and 12 amplicon sequencing runs/6 animals in T cell depleted samples (to ensure robust amplification of sgRNAs, we collected and prepared the anterior and posterior back skin separately from each animal).

As a side note, since we can only deplete immune cells between P4-P60, we cannot expect the same effect as *Tnfr1* KO animals as P4 animals already have substantial level of clonal expansion (Extended Data Figure 3b). Rather, we would expect a reduction of the expansion seen in the range of the P60 to P4 comparison as outlined in Extended Data Figure 3c. Moreover, the immune depletion in the skin is not 100% efficient but rather reduces macrophages and T cells to 20-30% (Supplementary Figure 7c).

Amplicon sequencing data from 12-14 replicates were analyzed using DESeq2. First, we normalized the sgRNAs in each sample to the 50 sgRNA control library using DESeq2 and then computed differential representation of the 150 perturbations.

T cell depletion did clearly not reduce clone sizes. However, macrophage depletion led to a small but reproducible shift in the overall distribution of the enriched perturbations (Figure 3g), reminiscent of the *Tnfr1* KO data (Figure 3d).

Focusing on the top 20 enriched perturbations, we found that while T cell depletion did clearly not reduce clone sizes, macrophage depletion had a significant effect on the top 20 enriched perturbations (Wilcoxon rank-sum test, $p=0.00071$). Again, we want to be careful with the interpretation, but given the high number of replicates, these data suggest that macrophages at least contribute to the TNF- α source in clonal expansion, but are likely complemented by other TNF- α sources such as dermal cells or systemic sources.

We also quantified immune cells in mCherry-positive regions and did not find any increased in CD45+ cells in perturbed areas, suggesting that there is no overall recruitment of immune cells to mCherry areas (Supplementary Figure 7d).

These data are included in Figure 3f-h, Supplementary Figure 7a-d and Extended Data Table 5.

2-Next the authors explore “the contribution of TNF- α signaling to the cellular processes underlying tumor initiation” using the well-established DMBA/TPA carcinogenic protocol. Although this is interesting, one would have expected the data to follow an unbiased strategy towards identifying what makes genetic perturbations contribute to tumorigenesis. That is,

i) determine the different selection of genetic perturbations that occurs during tumorigenesis (as per Figure 5C and Extended 8A),

This is an important point and we apologize if there was any confusion related to how we presented these data.

We believe we followed an unbiased strategy towards identifying what makes genetic perturbations contribute to tumorigenesis, but we take it as an opportunity to adjust how we present the data and clarify. Regarding i), we determine the different selection of genetic perturbations that occurs during tumorigenesis. In Figure 5b, we show the total sgRNA representation in 168 tumors (not normalized to P60). In Figure 5c, we show the different selection of genetic perturbations, as defined by tumor representation normalized to P60 skin representation (dial-out data). We also show tumor and P60 representation as a reference in the same heatmap.

ii) determine which tumour cell populations bear these perturbations (this is important and is missing), and

Thank you for raising this important point. Our decision to employ sgRNA amplicon sequencing in tumors, as opposed to combining single-cell RNA sequencing (scRNA-seq) with sgRNA capture, was informed by several critical factors in our previous manuscript.

1. Our methodological choice to focus on sgRNA amplicon sequencing in tumors over single-cell RNA-seq was driven by coverage limitations and the complexity of the clonal/multiclonal nature during tumorigenesis.

Even if we take the potential multiclonal origin of skin tumors into account as mentioned by the reviewer (Revees et.al. Nat Cell Biol 2018), the coverage is limiting. Therefore, it is critical to assess a high number of tumors to make any conclusions. In our previous scRNA-seq experiment of tumors (Figure 4c-d), we included 30 tumors, which is not sufficient to obtain reliable sgRNA representations, hence we did not assess sgRNAs in that experiment. This is the reason why we carried out sgRNA sequencing of individual tumors. In addition, sequencing reads (which are in the millions) by amplicon sequencing allow more granularity than cell numbers determined by scRNA-seq and sgRNA capture.

For our P4 and P60, our coverage (cells/sgRNA) was 240x and 366x, respectively, which is a good coverage that allows us to infer gene function for each perturbation. However, with the tumor experiments, the clonal/multiclonal nature of tumorigenesis changes the coverage calculations and currently not all factors are known to calculate the coverage reliably. For example, if we assume that a tumor is of polyclonal origin (e.g. 10 clones), we technically assess with each tumor 10 initial cells and thus 10 sgRNAs. In addition, sgRNAs are already enriched and depleted in P60 skin, further complicating the coverage aspect. For 112 tumors, to be on the cautious side, we focused on prioritizing the top 20 tumor and the top 20 enriched perturbations at the P60 stage in Figure 5c. In conventional CRISPR screens, enrichments are considered more robust than depletion due to several factors. Similarly, given the various factors outlined above, we want to be cautious and focus on the perturbations with high representation in either P60 skin or tumors.

To enhance our data set further, we have now incorporated additional 56 tumors (total of 168) into our analysis for Figure 5b-c, thereby improving the coverage and strength of our conclusions. Thus, we now focus on the top 30 perturbations with high coverage in tumor and in P60 skin.

2. Despite these limitations, we agree with the reviewer on the significance of identifying which tumor cell populations harbor specific perturbations, particularly in the context of the TSK population. To this end, we re-injected animals with our library of 500 sgRNAs. At P60, we then induced chemical carcinogenesis by DMBA/TPA for 12 weeks and harvested an additional 68 tumors. We also carried out sgRNA capture of the previous 30 tumors, totaling to 98 tumors.

Several technical challenges emerge in this context. The pronounced variation in guide RNA abundance, attributable to differential impacts on expansion and transformation, necessitates the profiling of a substantial number of individual tumors to accurately characterize single-cell perturbations with sufficient coverage (as outlined under 1). Although multiplexing identifiable samples for scRNA-seq is feasible using cell hashing methods, our revision involved conducting a comprehensive tumor hashing experiment utilizing lipid-anchor based hashing (MULTI-Seq, (McGinnis et al, Nature Methods, 2019). However, this technique proved incompatible with SCCs and the library preparation protocol we employed (BD Rhapsody). Alternative hashing strategies, such as antibody-based hashing, are effective for multiplexing a smaller number of samples (up to 12) but become excessively costly at the scale required for the proposed experiment (requiring a large number of scRNA-seq runs). Furthermore, computational demultiplexing via single nucleotide

polymorphisms, another widely-used deconvolution method, depends on a greater degree of genotype variability than what is found in our B6 mouse tumors and was also not successful.

As an alternative approach, we then wanted to individually prepare single-cell suspensions from different tumors and sort the preparations individually to ensure similar numbers of cells per tumor. However, this approach prolonged the single-cell preparation and FAC-sorting time significantly and led to lower library quality. Therefore, at the end, we had to prepare the tumors collectively for single-cell suspensions and FAC-sorting. However, even by adding similar amounts of tumor tissue, we cannot control whether few tumors disproportionately contribute to the single-cell suspension and we do not know how many of the tumors indeed have a specific cluster or TSK population (which is anyway a small cluster).

The results and the approaches outlined below are shown in the Reviewer Figure below. When assessing the TSK population within the scRNA-seq data of the 98 tumors, we found a high representation of *Notch1* and *Dnah11* perturbations. However, this was not specific to the TSK population as the same representation could also be observed in other clusters. Comparing to the 168 individually sequenced tumors and their sgRNA representation, we only find a robust *Dnah11* representation in 5 tumors (>10,000 reads). Generally, only 44 out of the 150 perturbations could be detected. These data raise the possibility that a small subset of tumors may disproportionately contribute to the single-cell RNA-seq dataset and we must be very cautious in drawing conclusions about TSK-specific signatures from these results.

Together, we believe that the individual tumor sgRNA sequencing (168 tumors) is the more robust data set. Our data do not show a trend toward specific genetic perturbations within TSK populations compared to other clusters. Thus, at the collected tumor stages, we do not have any evidence for a clonal drift within tumors (percentages of TSKs to other clusters), where monitoring by scRNA-seq and sgRNA capture would add additional insights. We are required to collect animals at 12 weeks post-DMBA treatment (due to very strict animal welfare regulations in Switzerland) and believe that assessing the TSK signature will require a different experimental approach with late-stage tumors and ideally an enrichment of TSKs (for which currently surface markers for FACS are missing) for sgRNA capture. Given the high number of tumors and extensive scRNA-seq libraries required for such an experiment, these experiments will also come with very high costs.

Together, deciphering a guide signature of different clusters or TSKs would be indeed highly interesting, but in our opinion will require an extensive and new experimental strategy that is outside the scope of this project.

The summary of the approaches and the data are presented in the Reviewer Figure.

iii) identify the transcriptional signature and regulatory networks defining the shift from “normal” clonal expansions (non-tumours) to selected perturbations in tumours (in the relevant cell population (ii)).

This may still end up being TNF- α signalling and TSK cells, but this would be a more neutral approach than preselecting TNF- α signalling. This strategy would help render potential targets to tilt mutant clonal expansions towards “beneficial” genetic perturbations that may potentially prevent rather than promote tumorigenesis.

This is an important point and we have now revised the manuscript to highlight this point further.

1. The main transcriptional signature we had assessed in the original manuscript was comparing basal tumor cells compared to P60 epidermal cells, which was in our opinion one of the most relevant comparisons.

Reviewer Figure. sgRNA capture and representation in scRNA-seq data from 98 tumors.

Reviewer Figure. sgRNA representation in 98 tumors.

a, Schematic outline of the experimental strategies to multiplex scRNA-seq and sgRNA capture of tumors.

b-f, sgRNA representation in different clusters.

g, Percentage of sgRNAs in main clusters and TSKs.

h, sgRNA representation in UMAP of combined 98 tumor scRNA-seq.

Page 10 in original manuscript: Gene set enrichment analysis (GSEA) comparing basal tumor cells versus P60 epidermal stem cells (control, Notch1 or Fat1 sgRNA) further supported the significant downregulation of TNF- α signaling and the upregulation of E2F targets and G2/M checkpoint genes, indicative of increased tumor cell proliferation (Extended Data Figure 6d). Consistent with this observation, TNF- α signaling is also strongly downregulated in human skin SCCs³⁴ compared to normal skin, including an overlap with 14 downregulated TNF- α signaling module genes (Extended Data Table 4).

Therefore, our focus on TNF- α signaling was driven by its strong downregulation (top downregulated pathway, NES -1.91).

2. To broaden our investigation, we have now carried out several additional analyses to systematically delineate the transcriptional signatures between relevant cell populations. We compared the transcriptional profiles of basal, differentiated, cycling and TSK tumor cell populations with the respective P60 skin populations. Our findings provide an overview of the different transcriptional signatures between these cell populations. Comparing basal tumor cells to P60 epidermal stem cells (control sgRNA), TNF- α signaling is the top downregulated pathway. This is also true when comparing basal tumor cells to Notch1 sgRNA P60 epidermal stem cells.

However, the TSK population compared to P60 epidermal stem cells suggest an upregulation of a distinct TNF- α signaling pathway (NES of 2.11) and EMT, including several of the TSK genes but none of the TNF- α module genes. These comparisons of relevant cell populations strengthen the evidence for our proposed working model of two distinct TNF- α gene programs during clonal expansion and tumorigenesis, with a switch from the TNF- α module genes in clonal expansion to epithelial autocrine TNF- α signaling in tumorigenesis.

We now incorporate these systematic analyses between the relevant cell populations in Extended Data Table 6.

3- In relation to this point, there is an important aspect missing. Despite the richness of the data presented, the authors make absolutely no connection between the tumour cell populations, and the genetic alterations introduced using the sgRNA screen. It would be relevant to link the different genetic alterations with the expansion of specific tumour populations (similar to what is done in Figure 2). Is the TSK population the one bearing the genetic perturbations that are selected in tumours? Mechanistically it is rather unclear whether the TSK population, emphasized by the authors as a “subpopulation in clonal expansions that influences the predisposition for tumor initiation” is actually bearing the selected genetic perturbations. No data is presented validating the existence of TSKs in clonal expansion pre-tumour stage.

This is an interesting suggestion to assess more closely the TSK population.

1. Our methodological choice to focus on sgRNA amplicon sequencing in tumors over single-cell RNA-seq was driven by coverage limitations and the complexity of the clonal/multiclonal nature during tumorigenesis, as outlined above (please see answer in question 2).

2. Furthermore, recognizing the importance of exploring the TSK population as the reviewer suggested, we carried out an additional extensive scRNA-seq experiments and coupled it with sgRNA identification. Please see our answer outlined in response to your question 2 ii) and the TSK data in the accompanying Reviewer Figure.

3. Finally, we assessed the existence of TSK-like cells (epidermal stem cells expressing several TSK markers) in clonal expansion at the pre-tumor stage. We analyzed all P60 epidermal stem cells and selected cells that show expression > 2 TSK genes (to increase robustness and minimize the possibility of noise). We identified 289 cells (out of 54,780 epidermal stem cells at P60) with robust expression of 2 or more TSK markers. We now present these analyses in Supplementary Figure 10.

Second, we found a notable overlap between the genes upregulated in the positively selected perturbations and TSK markers, with a subset also induced by TNF- α (Extended Data Figure 8e). We then employed SEACells, which groups single cells into distinct cell states, so-called metacells, to enable the robust identification of rare cell states such as putative EMT-inducing epidermal stem cells (Persad et al, Nature Biotech, 2023). The SEACells analysis exposed a cluster of metacells that showed an induction of these overlapping TSK marker genes even in expanded P60 epidermal stem cells (Extended Data Figure 8f).

Third, we also included a SEACells analysis for all 150 perturbations in epidermal stem cells and the classic TSK marker genes. We found two small cluster of metacells that strongly express the classic TSK marker genes. Together, these observations suggest the presence of a small TSK-like cell population in expanded clones in the P60 skin. These data are included in Supplementary Figure 10b.

4- Following from the previous point, the authors study how TNF- α signalling contributes to the transition from clonal expansions to tumour initiation. The data shows that, although the TNF- α signature does not seem to be as apparent in mCherry cells in tumours, there is a subpopulation (termed TSK) showing an autocrine TNF- α signature. The relevance of TNF- α signalling for tumour formation is then validated by inducing TNF- α ectopic expression in epidermal cells (lentiviral approach). These results show that TNF- α expressing skin presents abnormalities including hyperproliferation (Figure 5L, which needs to be validated by EdU incorporation or other proliferative assays) and invasive behaviour (Figure 5N, which needs to be validated with EMT markers). However, an important experiment missing to fully validate the relevance of the TSK TNF- α signature in driving tumorigenesis would require blocking TNF- α signalling (maybe by using Tnfr1 KO) and define whether less tumours emerge, the TSK signature is diminished (at least by immunofluorescence), and the selection of genetic perturbations in the remaining tumours has shifted.

1. We agree with the reviewer and conducted additional experiments to validate our findings. Specifically, we have now included EdU incorporation assays to confirm the proliferative rates of TNF- α -expressing P4 epidermal cells compared to controls. The EdU incorporation data suggest a clear increase in proliferation rates in TNF- α expressing P4 areas. These data are included in the main Figure 5h and Extended Data Figure 9e.

2. Additionally, we assessed the invasive behavior of TNF- α expressing epidermal cells using the EMT marker *Twist1*, *Vegfa*, *Mmp9* and *Mmp10* in qRT-PCR experiments. To this end, we injected E9.5 animals with our library of 500 sgRNAs and FAC-sorted TNF- α expressing and control cells of P4 animals. These new results indicate that TNF- α -expressing epidermal cells indeed promote EMT genes and are presented in Figure 5i.

3. We also characterized TNF- α expressing cells and control keratinocytes *in vitro* and found that TNF- α expressing cells exhibit faster proliferation rates. These data are included in Extended Data Figure 9c.

3. Regarding the suggestion “to validate the relevance of the TSK TNF- α signature in driving tumorigenesis by blocking TNF- α signalling (maybe by using Tnfr1 KO) and assess whether the TSK

a

b

b

genotype	maximum average papillomas per mouse	AUC method p value
WT	6.2	-
TNFR1 ^{-/-}	0.8	<0.0001
TNFR2 ^{-/-}	3.6	0.0006

Fig. 1 Skin carcinogenesis in wild-type and TNF- $\alpha^{-/-}$ mice. **a-c**, Initiation promotion. Mice were treated with one dose of 25 μ g DMBA and then 4 μ g TPA twice weekly for 15 weeks. **a**, C57Bl/129 wild-type mice (\blacksquare ; $n = 14$) and TNF- $\alpha^{-/-}$ mice (\bullet ; $n = 15$). **b**, Balb/c wild-type mice (\blacksquare ; $n = 14$) and TNF- $\alpha^{-/-}$ mice (\bullet ; $n = 13$). **c**, C57Bl/ [AUTHOR: IS THIS STRAIN DESIGNATION CORRECT?] wild-type mice (\blacksquare ; $n = 15$) and MCP-1^{-/-} mice (\bullet ; $n = 15$) **d**, Complete carcinogenesis. C57Bl/129 wild-type mice (\blacksquare ; $n = 15$) and TNF- $\alpha^{-/-}$ mice (\bullet ; $n = 20$) were treated with 5.1 μ g DMBA twice weekly for 20 weeks.

Figure 1 Genetic deletion of either TNFR1 or TNFR2 reduces the susceptibility of mice to skin carcinogenesis. In all, 10 wt, 10 *TNFR1^{-/-}* and 10 *TNFR2^{-/-}* mice were subjected to skin carcinogenesis by DMBA initiation and TPA promotion. The average number of papillomas per mouse (tumour multiplicity) was recorded weekly for 26 weeks after the start of promotion (**a**). Comparison of maximum tumour multiplicity in wt, *TNFR1^{-/-}* and *TNFR2^{-/-}* mice and *P*-values comparing AUC data for wt versus *TNFR1^{-/-}* or *TNFR2^{-/-}* mice (**b**). Data are representative of two independent experiments

Reviewer Figure. TNFR1 and TNF-alpha knockout mice are resistant to chemical carcinogenesis.

a-b, Original tumor induction plots and figure legends illustrating that TNF-alpha and TNFR1 mutant mice are resistant to chemical carcinogenesis. Left panel: Moore, R., Owens, D., Stamp, G. et al. Mice deficient in tumor necrosis factor- α are resistant to skin carcinogenesis. Nat Med 5, 828-831 (1999). <https://doi.org/10.1038/10552> Right panel: Arnott, C., Scott, K., Moore, R. et al. Expression of both TNF- α receptor subtypes is essential for optimal skin tumour development. Oncogene 23, 1902-1910 (2004). <https://doi.org/10.1038/sj.onc.1207317>

signature is diminished and the selection of genetic perturbations has shifted”: we acknowledge that the idea is intriguing. However, we believe that this experiment is technically not feasible.

Based on seminal work from the Balkwill and Khavari labs and our preliminary findings, *Tnfr1* knockout and *TNF- α* knockout mice both show severely abrogated tumor induction upon DMBA/TPA treatment (Reviewer Figure). This means that it is not possible to collect a nearly sufficient number of tumors to assess TSK signatures (which typically requires an advanced tumor size) and the selection of sgRNAs. Especially the selection of genetic perturbations requires a large number of tumors to acquire a sufficient coverage (please see also the explanation about the coverage in tumorigenesis experiments) to draw any meaningful conclusions. Based on the published data and our own observations, it is not possible to reach a sufficient number of tumors in *Tnfr1* KO mice. However, we recognize the importance of understanding TNFR1's role in different tumorigenesis phases (such as clonal expansion vs. tumor initiation) as a potential future research direction. Given the extensive nature of such a study, this is unfortunately beyond the scope of our current study.

4. Finally, we further assessed the relevance of TNF- α signalling and contrasted conventional from autocrine TNF- α signaling. To this end, we co-cultured TNF- α -expressing and receiving keratinocytes. Both cell types were equally exposed to TNF- α , being mixed 1:1 on the same plate and subjected to similar TNF- α levels in the media (secreted by the TNF-expressing cells). In a parallel experiment, keratinocytes were treated with an anti-TNF- α antibody. TNF- α -expressing and control keratinocytes were then sorted and subsequently subjected to bulk RNA sequencing. Notably, despite this equal TNF- α exposure and similar levels of TNF receptor 1, the two cell types displayed distinct transcriptional profiles. We identified 498 differentially expressed genes (adjusted p-value < 0.05, log₂ FC >1 or < -1, DESeq2), highlighting the significant impact of the autocrine TNF gene program on keratinocyte behavior, independent of the interaction between TNF- α and TNF receptor 1 (Supplementary Figure 9).

Cells receiving TNF- α showed an upregulation of genes associated with canonical TNF signaling pathways. Conversely, TNF- α -expressing cells demonstrated an upregulation of genes linked to epithelial-mesenchymal transition (EMT), aligning with the invasive characteristics promoted by the autocrine TNF gene program. The comparisons also revealed that while TNF- α signaling can be inhibited by TNF antibody treatment in TNF-receiving cells, such treatment did practically not elicit any changes (only 2 differentially expressed genes with p adj < 0.05) in the gene expression of TNF- α -expressing cells. This suggests that TNF- α antibody treatment is ineffective in inhibiting the autocrine TNF- α gene program. Consequently, these observations suggest that TNF antibody therapies in cancer patients require a more nuanced understanding of their inhibitory effect on conventional versus autocrine TNF- α signaling.

Collectively, these data suggest that a potential TNF- α antibody treatment to block TNF- α signaling during the tumor initiation stage may also be ineffective and that the *in vivo* epithelial TNF- α expression experiments in Figure 5 are currently in our opinion the most compelling experiments to validate the role of the autocrine TNF- α gene program *in vivo*.

This set of experiments is included in Supplementary Figure 9.

5-Strong statements are made throughout the text in relation to tumour/cancer progression and invasion. As far as I can see this study does not formally assess tumour progression or invasion. Hence, care should be taken when using those terms. Also, the data showing that the TSK population resides at the invasive tumour fronts, is rather unclear. Sections of side view tumours including a significant proportion of tumour stroma would be better suited to make this claim. The images presented, spatial transcriptomics and correlations shown make it very difficult to fully assess this point.

1. The reviewer is correct that we do not formally assess tumor progression or invasion (where we rely on the comprehensive characterization of TSKs by the Khavari lab, Ji et al., Cell, 2020). We revised the manuscript to remove or tone down any strong statements regarding tumor/cancer progression or invasion when referring to our own data (e.g. page 15 “Taken together, these findings underscore the potential relevance of the invasive TSK signature in driving human cancer progression and position the TSK gene program as a target for cancer therapy”.)

2. Regarding the TSK population at the leading edge, we initially refer to the very elegant paper by the Khavari lab (Ji et al., Cell, 2020). They defined for the first time the TSK population in their comprehensive multimodal analyses of human SCC and analyzed the TSK population by spatial transcriptomics in humans. The authors show convincingly that the TSK population resides indeed at the leading edge (e.g. Figure 3A, E, J, K, Figure S3D, E, M). In our study, we found a strong alignment in marker gene expression between our mouse tumor TSK population and that identified by Ji et al., affirming that both populations indeed represent the same population of TSKs.

3. To take into account the reviewer’s comment that we did not convincingly show that the TSKs reside at the leading edge, we re-injected our library of 500 sgRNAs into E9.5 embryos and carried out new DMBA/TPA experiments to induce chemical carcinogenesis. We collected additional tumors and now included a set of new spatial transcriptomics data. As suggested by the reviewer, we searched for side view tumors that include a significant amount of tumor stroma.

We now included 3 additional tumors in the manuscript. For example, the new tumor spatial transcriptomics data presented in Figure 4m show a side view of the tumor with the exophytic part and the base. The spatial transcriptomics data clearly underscored that the *Mmp10*-positive TSKs reside in the leading edge of the tumor at the base (Figure 4m). A similar localization can also be observed in the new tumor presented in Extended Data Figure 7d. The TSK localization is therefore reminiscent to the human TSK localization in SCCs presented by the Khavari lab (Ji et al, Cell, 2020).

Minor points

6-Page 3, First line of the Introduction: “Tumor evolution begins with the transformation of a single cell that expands to form a tumor mass”. There is evidence suggesting that tumours may also have a polyclonal origin (Winton et.al Br J Cancer 1989; Revees et.al. Nat Cell Biol 2018).

Thank you for pointing out the evidence suggesting a polyclonal origin of tumors (skin papillomas and SCCs) and including the references to the interesting studies by Winton et al. (1989) and Reeves et al. (2018).

We agree with the reviewer and now revised the sentence in the introduction to more accurately represent the diversity in tumor origins.

Revised sentence: Tumor evolution begins with the transformation of single or multiple cells that expand to form a tumor mass.

7- The authors propose that the low MOI used for infection “minimized the possibility of perturbed cells being influenced by neighboring cells with different perturbations”. However, mCherry positive clusters (Figure 1c) appear rather close to each other (less than 15 to 20 cells distance). Suggesting that an interactive effect between mutations cannot be discarded. In order to claim that observations are not due to “perturbation” synergy, it would be necessary to validate mechanistic observations, such as Tnfr1 KO experiment, using low MOI infection for a limited (but relevant) selection of genetic CRISPR perturbations.

The reviewer is correct that we cannot completely exclude local effects between perturbations such as for instance secreted factors that influence neighboring clones at a certain distance.

1. To clarify, our aim with this sentence was to highlight that we can minimize the possibility that perturbed cells are being influenced **directly by cell-cell** contacts of neighboring perturbed clones. Despite the lack of multicolor analysis to confirm the monoclonality of mCherry areas, we believe the low infection rates (median 2.4%, range 1-12.5%) largely mitigate the risk of direct interaction among many clones, at least initially. Furthermore, considering our library contains 500 sgRNAs and 50 control sgRNAs, rare interactive effects between mutations should be in general normalized out, given the potentially 11,175 different pairs of perturbed clones (based on 150 genes).

To clarify, we now **revised this sentence to:** These low MOIs minimized the possibility of perturbed cells being influenced by *cell-cell contacts* of neighboring cells with different perturbations.

2. Taking into account the reviewer's input, we have incorporated wholemound imaging over larger skin areas to better illustrate the distribution of mCherry clones. These images reveal significant gaps between clones, often exceeding 15-20 cell distances, providing a clearer visualization of the clone distribution. Of course, we cannot rule out rare synergistic effects between perturbations or large range interactions, but generally, the imaging and the quantifications indicate that the perturbations are in general separated to avoid cell-cell interactions. These additional data should offer the reader a better assessment of the distribution of mCherry clones in our study.

These images are now included in Supplementary Figure 1.

8-Despite the fact that authors use low MOI, it would be important to provide an immunofluorescence image showing mCherry labelling and CD45 staining for a large area of the skin (confocal tiled skin wholemound). It is important to visualize the extent to which skin cells show genetic cancer perturbations, which may lead to a systemic response (CD45, immune) beyond individual mutations.

This is an interesting point. We now include additional CD45 quantifications.

1. We now include a larger area of the skin and show the distribution of the mCherry clones. This is included in Supplementary Figure 1.

2. We included the quantification of CD45-positive cells in the skin, comparing CD45-positive cells in mCherry-positive and negative areas. We observed no change in CD45-positive cells in mCherry-positive areas, suggesting that immune cells are not enriched in the infected sgRNA areas. These analyses are included in Supplementary Figure 7d.

9-There is also the concern as to whether the developmental aspect of the CRISPR screening used (gene perturbations are induced via in utero transduction) is influencing the clonal dynamics studied.

The reviewer raises a valid point.

Our experimental design was indeed structured to provide the possibility to distinguish these two processes, the developmental aspects (which may be equally interesting) from postnatal clonal dynamics. By incorporating the P4 time point, we established a potential second baseline for our analysis. This baseline allows us to compare gene expression and clonal dynamics at P60 relative to P4, effectively differentiating between developmental processes and postnatal clonal expansions. Notably, our data show that many clonal expansions observed at P4 continue to evolve and expand by the P60 stage. This ongoing expansion beyond the early developmental stage underscores the postnatal influence on clonal dynamics.

10-In relation to the authors statement “suggesting that Notch1 clonal expansion is primarily driven by the differentiation block and not by increased proliferation”, the fact that Notch inhibiting mutations lead to clonal expansion in squamous tissues through a differentiation imbalance has already been reported (Alcolea et.al. Nat Cell Biol 2014). Please revise.

We thank the reviewer for bringing this publication to our attention. We now revised this sentence and included the reference of the study by Alcolea et al.

Revised sentence: However, *Notch1* knockout epidermal stem cells exhibited a low proliferation index, suggesting that *Notch1* clonal expansion is primarily driven by the differentiation block and not by increased proliferation, an aspect reflecting the differentiation imbalance previously demonstrated in elegant studies in the esophagus (Alcolea et al, Nat Cell Biol, 2014).

11-How can the authors explain finding control sgRNAs among the most enriched perturbations (Fig 2A, ctrl1)?

We would like to underscore several aspects regarding the control library in our experiment:

1. We deliberately included a large sgRNA library comprising 50 guides, representing 10% of the total library. The large control sgRNA ensures that we have the possibility to gain a robust understanding of the behavior of control sgRNAs in different cell types and throughout tumorigenesis.

2. In the interest of full transparency, we have highlighted the presence of these 50 sgRNAs in Figure 2a and Extended Data Figure 3. The distribution of our control library appears to be uniform and is consistent with distributions seen in other similar CRISPR screening studies.

3. We acknowledge the possibility that the control library might have outliers and there could be instances where control sgRNAs exhibit off-target effects, leading to either enrichment or depletion. An example of this is Ctrl1, which indeed ranks in the top 10. We have included these details to be completely transparent and provide a complete picture of the control library's distribution for our readers.

12-It would be very helpful to highlight in Figure 2F which perturbations are enriched (undergo clonal expansion) as per Figure 2A.

This is a very helpful suggestion. We now reordered Figure 2e and 2g (Figure 2f was the WGCNA schematic) according to the enrichment rates, similar to Figure 2a.

13-When considering data on Figure 2H-K and Extended Figure 4D-E, one cannot help but wonder how specific this module is for TNF- α rather than a tissue stress response that would ultimately also involve TNF- α . The authors should include Pearson correlations for the other two main processes associated to the major EpSCs module to see how these compare to Figure 2K (UV response and Hypoxia). Along the same vein, one would need to see the equivalent of Figure 3A for those two additional processes.

We are grateful for the reviewer's suggestion and have accordingly expanded our analysis.

As recommended, we have now included Pearson correlations for the other two main processes associated with the major EpSCs module (UV response and hypoxia) to assess how these compare to Figure 2K.

Given that these processes encompass a subset of the 25 module genes (detailed in Extended Data Figure 4d), a reasonably high Pearson correlation was anticipated. However, our analysis revealed that the complete module genes or the specific 17 TNF- α genes exhibited a higher correlation (0.683 and 0.668, respectively) compared to those for UV response and hypoxia (0.599 and 0.583). To provide a balanced view, we also mention the enrichment of hypoxia and p53 signaling along TNF- α signaling on page 8.

Additionally, we examined gene cohorts differentially expressed between the top 5 and middle 5 perturbations (as shown in Figure 2m). While there are overlaps with genes in the TNF- α module, such as those related to p53, resulting in a decent Pearson correlation, the correlation significantly diminishes for unrelated gene cohorts, such as those involved in apoptosis or IL-2/STAT5 signaling (Supplementary Figure 3b, middle and right panel).

We now included these analyses in Supplementary Figure 3.

14-In Figure 3e, the authors reveal how two of the major genetic perturbations contributing to clonal skin expansion *Notch1* and *Fat1* do not seemingly depend on TNF- α signalling. Given the relevance of these mutations and interest of the field, it would be important to define what other molecular traits are shown by these genetic perturbations that could explain their expansion phenotype. Indicating that these may be linked to reduced differentiation without formal validation represents a rather vague description.

Our new set of P60 control and *Tnfr1* knockout experiments suggests that *Notch1* and *Fat1* also show a significant dependency on TNF- α (Figure 3d). As indicated by the reviewer, testing TNFR1-dependency at the P60 time point analysis is indeed the better experiment and we currently have no evidence that the *Notch1* and *Fat1* perturbations at the P60 time point behave differently than the other strongly enriched perturbations.

15-Figure 4A and downstream analysis of tumour data. The clonal selection of sgRNA perturbations and cell populations involved are compared to a baseline of P60, which is the moment when the DMBA/TPA treatment start. There is a 12 week difference between the baseline reference time point and the tumour samples. One cannot discard that some of the changes in gene selection are associated to this discrepancy. This should be acknowledged and referred to in the main text.

The reviewer is correct and raises an important point. We now acknowledge this important caveat in the main text and added the following sentence:

Revised sentence: This score reflects the potential of each perturbation to promote tumor initiation, although it is important to consider the potential for ongoing clonal evolution in normal epithelia from P60 until the conclusion of the DMBA/TPA experiment.

16-I am rather unconvinced by the way that perturbation selection is normalised in Figure 5C, and extended 8A. The fact that a genetic perturbation is selected in “normal tissue” and continues to be selected under the competitive clonal dynamics of a tumour indicates its potential participation in the process. Hence, this approach should be used to further mine the data for perturbations that are positively selected upon tumorigenesis (when compared to normal adult counterparts, P60). However, those perturbations that are expanded in both conditions (P60 and tumours) should not be disregarded. We do not know enough about mutant clonal dynamics to be able to make this assumption.

We thank the reviewer for bringing these issues up and take the opportunity to clarify and revise Figure 5c.

1. We agree with the reviewer and clarified these aspects. By comparing tumor perturbations to the P60 baseline, we indeed focus on genes that “continue to be selected under the competitive clonal dynamics of a tumour” (Figure 5c, left panel side labelled as “positively selected in tumors”). We therefore mine the data for perturbations that are positively selected upon tumorigenesis.

2. We also agree with the reviewer that the ones expanded in both conditions should not be discarded. The perturbations that are expanded in both conditions, tumors and P60, are also accessible to the reader in the middle row of the heatmap. The representation in tumors is shown in Figure 5b.

17-Please revise the order of the figures so that they are in line with the text. Also make sure that all the figures are mentioned in the text.

We now revised the order of the figures and made sure they are, whenever possible, in line with the text (there was one exception in Figure 5 where it was not possible to separate the immunofluorescence images).

We now also made sure all figures are mentioned in the text.

Referee #4 (Remarks to the Author):

In this manuscript Renz et al, the authors seek to evaluate the effect of squamous cell carcinoma driver mutations on clonal expansion of normal epidermis cells and oncogenic transformation. To do this the authors developed an in vivo CRISPR screening platform coupled with a single cell transcriptomic read-out. They generated a CROP-seq library against 150 frequently mutated genes in skin squamous cell carcinoma and head and neck squamous cell carcinoma and infected E9.5 embryos in utero to infect the ectoderm with the CROP-seq library and monitored the sgRNA abundance and transcriptional changes underlying clonal expansions at postnatal 4 days and 60 days. The authors identified Notch1, Fat1, Notch2 as the most enriched sgRNAs at P60 relative to control, and identified that indeed the loss of function of Notch1 restricted differentiation rather than increasing proliferation. The authors identified TNF-alpha as a convergent pathway that underlies clonal expansions for many of the perturbations and noted a perturbation-dependent cell type specific enrichment of TNF-alpha. The authors provide further evidence that clonal expansion is dependent on TNF-alpha signaling by repeating the experiment in a Tnfr1 heterozygote and state that for the 20 most perturbed genes, 17 have a Tnfr1 dependency. Furthermore, the authors induced tumorigenesis in their model using DMBA/TPA treatment and profiled 112 tumors. Using single-cell and spatial transcriptomics they map to the invasive front one of the cancer cell populations they annotate as the previously described tumor-specific keratinocyte (TSK, Ji et al., ref 31). The authors make a claim that the TSK state has an autocrine TNFa regulatory loop. When comparing the perturbations that led to clonal expansion in non-cancer to cancer settings, the authors report that an EMT state is enriched in the cancer expansions. Finally, they propose a model whereby TNFa is a state leading to cancer, which is stabilized in an invasive subpopulation with an autocrine loop.

The manuscript makes important and interesting claims. The authors have an elegant and powerful in vivo system for identifying fitness (clonal expansion) effects for specific perturbations. The authors use the results to address an open and important question regarding the relationship between clonal expansion and tumor initiation. The manuscript also has an extensive number of models and draws evidence from numerous experiments.

There however a number of concerns regarding the evidence provided for the claims detailed below.

1. The authors claim that a convergent phenotype of oncogenic mutations is the TNF-alpha signature. The authors provide evidence that the TNF-alpha gene expression program is enriched across different cell types of the epidermis with perturbations relative to control. However, the authors do not show how each perturbation and consequently the degree of clonal expansion relates with the activation of the TNF-alpha gene expression program. The authors may achieve this by ordering Figure 2g-h by their sgRNA representation.

We thank the reviewer for this helpful suggestion.

In accordance with the suggestion, we now revised Figure 2 and included a number of changes. As suggested by the reviewer, we ordered Figure 2e and 2g to mirror the sgRNA representation in Figure 2a.

For Figure 2h, we chose to retain the existing order to better illustrate the hierarchical clustering of TNF- α module genes and their perturbations. However, to further elucidate the relationship between TNF- α gene expression and clonal expansion, we have added a new analysis to more directly link TNF- α gene expression with clonal expansion. We identified four major clusters of perturbations based on similar TNF- α gene expression profiles. Utilizing these clusters, Figure 2i now displays the extent of clonal expansion for each cluster. Notably, we observed that clusters 1 and 2, characterized by higher TNF- α module gene expression, show significantly greater clonal expansion compared to the other clusters. This new analysis in Figures 2h-i is aimed at providing a clearer connection between TNF- α module gene expression and clonal expansion dynamics.

Finally, the formal assessment whether clonal expansion correlates with the activation of the TNF- α gene module is presented in Figure 2j-l and Extended Data Figure 4e (single genes), where we fitted a linear module to assess how the module genes predict clonal expansion, indicating that the TNF- α signaling module explains a substantial portion of the variance in clonal expansion rates.

Additionally, the authors claim that they corroborate their WGCNA TNF-alpha gene modules by two alternate approaches - while the first one included GSEA of differentially expressed genes that indeed supports their findings, the second approach simply shows that groups of perturbations have similar transcriptional responses but does not show how these transcriptionally similar perturbations converge on TNF-alpha signaling. (Extended Figure 5).

The reviewer is correct.

To address this issue raised by the reviewer, we analyzed the groups of perturbations with similar transcriptional response. We found that 4 clusters, including the cluster with perturbations that lead to strong clonal expansions such as *Notch1*, *Notch2*, *Fat1* and *Myh1* (cluster 4), showed differentially expressed genes strongly enriched in TNF- α signaling.

We now include this additional analysis in Extended Data Figure 5a.

The observation that a TNF α state is present in many of the perturbations is very interesting. The authors do not make a case however regarding what is its functional role for the cells.

The reviewer is correct and raises an important point.

1. Regarding the functional role, we primarily focus in this study on proliferative aspects of the TNF- α state. To directly assess the functional role of the TNF- α state, we now experimentally tested the role of select TNF- α module genes in keratinocytes. Previous elegant work from the Khavari lab had already implicated TNF- α downstream signaling in skin hyperproliferation, where they elegantly

demonstrated that TNFR1-dependent JNK activation conferred skin hyperproliferation (Zhang et al, Genes Dev, 2004).

2. To assess the role of ectopic TNF- α and epithelial TNF- α expression on keratinocyte proliferation, we either treated keratinocytes with TNF- α or expressed a lentiviral TNF- α construct in keratinocytes. We then analyzed proliferation rates and observed that both conditions increased proliferation of keratinocytes. We included these data in Extended Data Figure 9c.

3. We now experimentally tested the role of three TNF- α module genes on keratinocyte proliferation. We overexpressed the TNF- α module genes *Jun*, *Fos* and *Ccn1* in keratinocytes using lentiviral constructs and carefully assessed proliferation rates in control infected and TNF- α module gene-expressing cells. We observed *Jun* and *Fos* expression enhanced keratinocyte proliferation. *Ccn1* expression increased proliferative rates, albeit without reaching the significance threshold. These experiments confirm that increased expression of TNF- α module genes can indeed directly confer keratinocyte proliferation. We now included these data in Extended Data Figure 4f.

4. Finally, we further assessed the relevance of TNF- α signalling and contrasted conventional from autocrine TNF- α signaling. To this end, we co-cultured TNF- α -expressing and receiving keratinocytes. Both cell types were equally exposed to TNF- α , being mixed 1:1 on the same plate and subjected to similar TNF- α levels in the media (secreted by the TNF-expressing cells). In a parallel experiment, keratinocytes were treated with an anti-TNF- α antibody. TNF- α -expressing and control keratinocytes were then sorted and subsequently subjected to bulk RNA sequencing. Notably, despite this equal TNF- α exposure and similar levels of TNF receptor 1, the two cell types displayed distinct transcriptional profiles. We identified 498 differentially expressed genes (adjusted p-value < 0.05, log₂ FC >1 or < -1, DESeq2), highlighting the significant impact of the autocrine TNF gene program on keratinocyte behavior, independent of the interaction between TNF- α and TNF receptor 1 (Supplementary Figure 9).

Cells receiving TNF- α showed an upregulation of genes associated with canonical TNF signaling pathways. Conversely, TNF- α -expressing cells demonstrated an upregulation of genes linked to epithelial-mesenchymal transition (EMT), aligning with the invasive characteristics promoted by the autocrine TNF gene program. The comparisons also revealed that while TNF- α signaling can be inhibited by TNF antibody treatment in TNF-receiving cells, such treatment did practically not elicit any changes (only 2 differentially expressed genes with p adj < 0.05) in the gene expression of TNF- α -expressing cells. This suggests that TNF- α antibody treatment is ineffective in inhibiting the autocrine TNF- α gene program. Consequently, these observations suggest that TNF antibody therapies in cancer patients require a more nuanced understanding of their inhibitory effect on conventional versus autocrine TNF- α signaling.

Collectively, these data suggest that a potential TNF- α antibody treatment to block TNF- α signaling during the tumor initiation stage may also be ineffective and that the *in vivo* epithelial TNF- α expression experiments in Figure 5 are currently in our opinion the most compelling experiments to validate the role of the autocrine TNF- α gene program *in vivo*.

This set of experiments is included in Supplementary Figure 9.

2. In support of the claim that TNF-alpha, they repeat the *in vivo* single cell CRISPR in *Tnfr1* heterozygous mice. The *Tnfr1* experiment is important however it is not possible to evaluate its results given the way the authors have reported on it. Extended data figure 3 shows the results for the first experiment however the authors fail to show the analogous results for the *Tnfr1* experiment. Instead the authors restrict analysis to the top 20 most enriched genes from the control. Even for these they do not show the actual guide abundances but only a ratio of two ratios (Fig. 3d). Finally in Fig. 3e they

compare the ratio of ratios (TNF1-dependency) with a ratio only (P4 enrichment). Thus for this crucial experiment, there is no way to evaluate the results.

We agree that the results of these experiments were not properly presented for evaluation. In addition, an additional reviewer criticized that the P4 time point was not the proper time point for this question, given that most analyses were performed for the P60 time point.

Therefore, to directly address these important points, we carried out a new set of experiments to address the role of TNFR1 in clonal expansions at P60 skin. We re-injected our library of 500 sgRNAs into *Cas9; Tnfr1 +/-* knockout animals and collected the skin of P60 animals. We then created single-cell suspensions and performed amplicon sequencing of sgRNAs.

Taking the same approach as for the previous P4 analysis, we calculated TNFR1-dependency for each perturbation and included the P60 TNFR1-dependency data in the main Figure 3d. We now also show the complete set of 150 perturbations and their TNFR-1 dependency as suggested by the reviewer, and not just the top 20.

There are several interesting aspects of the new P60 data set analysis.

1. Our complete P60 data set clearly indicates a pronounced TNFR1-dependency primarily in the enriched perturbations. This is evident from the locally weighted regression curve (red), which distinctly shows the TNFR1-dependency for these enriched perturbations on the right side of the graph. Conversely, for non-enriched perturbations (middle and left side of the graph), the TNFR1-dependency diminishes, returning basically to zero. This stark contrast underscores the specificity of the TNFR1-dependency in relation to perturbation enrichment. We have also noted that for highly depleted perturbations like p63, where cell counts are very low or almost lost, as expected the data becomes less reliable due to increased noise in ratios.

2. As pointed out by the reviewer, clonal dynamics are indeed not modulated in the exact same manner in adult and postnatal stages. While the overall pattern of TNFR1-dependency remains broadly consistent for the enriched perturbations between P4 and P60, there are two notable and important exceptions: *Fat1* and *Notch1* now also exhibit a clear TNFR-1 dependency at P60. This finding at P60, differing from the postnatal stage, underscores the importance of developmental context in the modulation of clonal dynamics. This reinforces the value of including data from the more accurate P60 time point for a comprehensive understanding.

3. To include statistical analyses, we utilized the MAGeCK pipeline, a standard and robust method for CRISPR screen analysis, to scrutinize our control and *Tnfr1* knockout data set. From the top 20 top enriched perturbations, 17 perturbations show a highly significant False Discovery Rates (FDR), underscoring that the majority of the top enriched perturbations at P60 indeed show a strong TNFR1-dependency.

Also for the tumorigenesis experiment described in Figure 4 and 5, the results are not shown for all of the guides (as shown in extended data figure 3 for the first experiment).

1. We now show the results for all the perturbations assessed in the tumorigenesis experiment in Extended Data Table 8.

2. We appreciate the opportunity to clarify any potential misunderstanding regarding the scope and the strategy of our tumorigenesis experiments. These experiments in the original manuscript involved sgRNA amplicon sequencing of 112 tumors (not involving scRNA-seq). In Figure 5c, we highlighted

the selection score between P60 and tumors in the top 40 perturbations (coverage in P60 and tumor). For a comprehensive view, the entire list of perturbations can be found in Extended Data Figure 8.

3. To bolster the robustness of our findings and increase the coverage of our study, we have now expanded the number of tumors processed in these experiments (new total n = 168 tumors). This enhancement allows for a more thorough investigation and substantiates our claims.

4. The assessment of sgRNA representation in scRNA-seq data of tumors was not included in Figure 4. Our methodological choice to focus on sgRNA amplicon sequencing in tumors over single-cell RNA-seq was driven by coverage limitations and the complexity of the clonal/multiclonal nature during tumorigenesis, as outlined below.

Even if we take the potential multiclonal origin of skin tumors into account (Revees et.al. Nat Cell Biol 2018), the coverage is limiting. Therefore, it is critical to assess a high number of tumors. In our previous scRNA-seq experiment of tumors (Figure 4c-d), we included 30 tumors, which is not sufficient to obtain reliable sgRNA representations, hence we did not assess sgRNAs in this experiment. This is the reason why we carried out sgRNA sequencing of individual tumors. In addition, sequencing reads (which are in the millions) by amplicon sequencing allow more granularity than cell numbers determined by scRNA-seq and sgRNA capture.

For our P4 and P60, our coverage (cells/sgRNA) was 240x and 366x, respectively, which is a good coverage that allows us to infer gene function for each perturbation. However, with the tumor experiments, the clonal/multiclonal nature of tumorigenesis changes the coverage calculations and currently not all factors are known to calculate the coverage reliably. For example, if we assume that a tumor is of polyclonal origin (e.g. 20 clones), we technically assess with each tumor 20 initial cells and thus 20 sgRNAs. In addition, sgRNAs are already enriched and depleted in P60 skin, further complicating the coverage aspect. For 112 tumors, to be on the cautious side, we focused on prioritizing the top 20 tumor and P60 enriched perturbations in Figure 5c.

However, with the inclusion of the additional tumors (now 168), we focus on the top 30 tumor and P60 enriched perturbations in Figure 5c. We also show the results of the tumor sgRNA representation, the normalized P60 representation and selection score for all the perturbations in Extended Data Table 8.

3. One of the major claims of the manuscript is that the TNF-alpha switches from an immune cell driven to an autocrine TNF-alpha signaling loop. This is supported by applying CellChat, but the authors do not provide any additional analysis or experimental data to support it.

The reviewer is correct and it is a great suggestion to include experimental data to support it. We now included a large set of immune depletion experiments to support the working model.

To explore the role of immune cells in clonal expansion, we assessed the rate of clonal expansions by the 150 perturbations upon immune depletion experiments. We re-injected animals with the library of 500 sgRNAs into E9.5 embryos and subsequently divided the animals into three experimental groups from the P4 stage on:

Experimental group 1: Macrophage depletion by i.p. injecting CSF1R/CD115 antibodies twice a week.

Experimental group 2: T cell depletion by i.p. injecting CD4/CD8 antibodies.

Experimental group 3: Control arm by i.p. injecting IgG2a isotype control antibodies.

At P60, we harvested back skin from the animals, prepared single-cell suspensions, and conducted amplicon sequencing of the sgRNAs to evaluate their representation in the different experimental groups. This strategy parallels the new set of *Tnfr1* knockout experiments at the P60 stage.

We carried out these experiments in two independent runs, including 14 amplicon sequencing runs from 7 animals in control and macrophage depleted samples and 12 amplicon sequencing runs/6 animals in T cell depleted samples (to allow robust amplification of sgRNAs, we collected and prepared the anterior and posterior back skin separately from each animal).

As a side note, since we can only deplete immune cells between P4-P60, we cannot expect the same effect as *Tnfr1* KO animals as P4 animals already have substantial level of clonal expansion (Extended Data Figure 3b). Rather, we would expect a reduction of the expansion seen in the range of the P60 to P4 comparison as outlined in Extended Data Figure 3c. Moreover, the immune depletion in the skin is not 100% efficient but rather reduces macrophages and T cells to 20-30% (Supplementary Figure 7c).

Amplicon sequencing data from 12-14 replicates were analyzed using DESeq2. First, we normalized the sgRNAs in each sample to the 50 sgRNA control library using DESeq2 and then computed differential representation of the 150 perturbations.

T cell depletion did clearly not reduce clone sizes. However, macrophage depletion led to a small but reproducible shift in the overall distribution of the enriched perturbations (Figure 3g), reminiscent of the *Tnfr1* KO data (Figure 3d).

Focusing on the top 20 enriched perturbation, we found that while T cell depletion did clearly not reduce clone sizes, macrophage depletion had a significant effect on the top 20 enriched perturbations. Again, we want to be careful with the interpretation, but given the high number of replicates, these data suggest that macrophages at least contribute to the TNF- α source in clonal expansion, but are likely complemented by other TNF- α sources such as dermal cells or systemic sources.

These data are included in Figure 3f-h, Supplementary Figure 7a-d and Extended Data Table 5.

4. The authors claim to identify two distinct subpopulations of TSK, that are differentiated by the expression of MMP10. As the authors cite, this was first identified as a single population in Ji et al., (ref.31). It seems intriguing that the two subpopulations of TSKs appear to be in two distinct clusters. The authors do not show enrichment of the TSK gene signature or module score, representing their annotation of cell types in their dataset to claim that both these clusters of cells are TSKs. This would be critical in understanding the difference between the two TSK sub populations.

We thank the reviewer for highlighting this aspect. Indeed, we initially identified a cluster in our mCherry-positive single-cell tumor cells with markers in common with the TSKs as described in the paper by the Khavari lab (Ji et al.). We realized that the cluster could be further divided into two subclusters, clearly differentiated by the expression of *Mmp10*, among other genes.

We now added a module score based on the full list of markers provided in Ji et al.'s Supplementary Table S3. Of the 100 human gene symbols, we could convert 95 to their murine homologs (obtained from the Mouse Genome Informatics hosted by The Jackson Laboratory at jax.org). The full signature helped us better discriminate among the two subpopulations and suggests that the *Mmp10*-positive cluster is clearly closer to the human TSKs, whereas the *Mmp10*-negative ones exhibit a lower score, halfway between that of the *Mmp10*-positive cluster and the epidermal stem cells. Furthermore, the

Mmp10-positive population also shows a higher score for the EMT hallmark signature, as previously also shown by Ji et al.

While *Mmp10*-negative TSK have a higher TSK module score, *Mmp10*-positive TSK therefore more closely resemble the human TSKs as described by the Khavari lab.

These analyses are now included in Extended Data 7f.

Furthermore, the authors project the TSKs to their spatial transcriptomic data, that reveals the MMP10-negative and -positive cell state do not colocalize, while the MMP10-negative appears to be at the edges of the tumor and the positive at the core of the tumor, suggesting an inherent difference between the two cell states. It is critical to understand the differences between these cell states, as the authors then use these annotations to claim the presence of autocrine TNF-alpha signaling in TSKs.

We agree with the reviewer. We now carried out additional spatial transcriptomics experiments and report 3 additional tumor Visium spatial transcriptomics data sets (for a total of 7 tumors). We confirm also with the co-occurrence analysis on the cell-type deconvolution results in the new tumor set that *Mmp10*-positive TSKs and *Mmp10*-negative cells are indeed spatially separated (negative co-occurrence). For example, Figure 4m highlights that, while *Mmp10*-negative cells are within the differentiated, exophytic part of the tumor, *Mmp10*-positive TSKs reside at the invasive front within the base. Co-occurrence analysis of this tumor suggests again that *Mmp10*-positive TSKs and *Mmp10*-negative cells have a negative co-occurrence (Extended Data Figure 7h, right panel). *Mmp10*-positive TSKs co-localize within a fibrovascular niche, as reported before (Ji et al, Cell, 2020) and are associated with basal cells. In contrast, *Mmp10*-negative cells are consistently across the different spatial transcriptomics tumors associated with differentiated cells.

These analyses and spatial transcriptomic data are included in Figure 4m and Extended Data Figure 7.

Minor Comments:

Please correct this reference as the wrong journal is listed: Ji, A. L. et al. Multimodal Analysis of Composition and Spatial Architecture in Human Squamous Cell Carcinoma. J. Clean. Prod. (2020). doi:10.1016/J.CELL.2020.05.039

We apologize for this bug in the citation tool and now corrected the reference.

The authors should clarify in the main text what it means to target a CNV. Are they targeting frequently amplified/deleted regions?

We apologize for this oversight in our initial presentation of Extended Data Figure 1c. To address this, we have now revised the main text and figure legend to provide clearer information. It now specifies that CDKN2B is deleted in human head and neck squamous cell carcinoma (HNSCC) and highlights that the other 15 copy number variations (CNVs) included in our screen are amplifications. This clarification aligns with our observations in Figure 2a, where many of the perturbed CNVs, as expected, show strong depletion – a pattern typically seen when an amplified gene in cancer is targeted.

Revised sentences:

Page 4: To longitudinally monitor the function of cancer genes, we chose the 150 most frequently altered genes in human head and neck (HNSCC) and skin squamous cell carcinomas (sSCC), comprising 134 mutated genes (predominantly tumor suppressors) and 16 copy number variations (CNVs, **15 of which are amplifications**) (Figure 1a-b, Extended Data Figure 1a-c).

Page 5: In contrast, sgRNAs targeting the 15 amplified CNVs in cancer such as *p63* - an essential transcription factor for epidermal development and maintenance¹² - were as expected among the most strongly depleted guides at P60 (Figure 2a).

Figure legend ED 1a: While *CKDN2B* is deleted in HNSCC, the other 15 CNVs represent amplifications.

The legend for Extended Figure 4c is hard to understand.

We agree with the reviewer and now revised the legend to:

Analysis of gene modules in the different cell populations of the P60 skin. The figure depicts 44 gene modules identified within nine distinct cell clusters at P60. Using Weighted Gene Correlation Network Analysis (WGCNA), we explored correlation patterns among genes to identify modules/groups of genes that co-vary within these clusters. The dot plot visualizes the average log₂ fold change for genes within each WGCNA module relative to control sgRNA cells, offering insights into the perturbation effects on different cell types. A notable finding is the identification of a significant TNF- α gene module, which is consistently present across six different cell clusters in P60 skin, comprising largely the same set of genes. Additionally, the plot highlights modules corresponding to specific cell types, such as melanocytes (characterized by genes like *Sox2*, *Dct*, *Tyrp1* and *Kit*) or corneal cells within the suprabasal cluster. The color of each dot indicates the log₂ fold change, while the size of the dot represents the adjusted p-value, calculated using the Benjamini-Hochberg procedure for controlling the false discovery rate. Detailed lists of genes in each module can be found in Extended Data Table 2. The nomenclature for modules follows a Module + cluster number in P60 + color format, referencing the specific modules within each cluster.

The equation to determine TNFR1 dependency (shown in Figure 3) needs to be presented better.

In response, we have revised this figure to more effectively illustrate the TNFR1-dependency in our study. We have updated the schematic to align with our modified experimental approach, which focuses on monitoring TNFR1-dependency in P60 skin. To enhance the figure's readability and to keep it accessible to the reader, we have also moved the equation to the figure legend.

Clarification is required for the difference between Figure 3a and Figure 2g

In Figures 3a and 2g in the original manuscript, we utilized two different approaches to estimate the combined perturbation effect of TNF- α module genes. In the case of Figure 2g, we first calculated the differentially expressed genes for each of 150 perturbations vs control in the different clusters. Next we took the average log₂ fold change of the TNF- α module genes in each of these perturbations to compute the average perturbation effect. We combined the p-values of differentially expressed module genes over control by the Fisher's method using the *metap* R package. The p-values were adjusted by Benjamini-Hochberg method to control the False Discovery Rate.

In the original Figure 3a (now moved to Supplementary Figure 4a), we took a similar approach as described by Jin et al, *Science*, 2020 (Regev and Arlotta labs). We first ran the *moduleEigengenes* function of WGCNA R package to obtain the module eigengenes (i.e. first principal component) of the given module. Next we normalized the values using the *scale* function in R (from the base package).

Afterwards, we performed linear modeling in R to obtain a perturbation score for each perturbation over control with the formula: normalized gene score \sim perturbation. We extracted p-values, standard error and effect sizes from the linear model for plotting as described in Jin et al, Science, 2020. Therefore, the first approach in Figure 2g calculates the perturbation effect of each module gene separately and combines them, while the second approach in Figure 3a (now Supplementary Figure 4a) estimates the perturbation effect of all module genes simultaneously.

We agree with the reviewer that it was confusing to include both figures as main figures. Thus, we decided to move Figure 3a to Supplementary Figure 4 and keep Figure 2g (now reordered according to P60 enrichment) as it highlights also the perturbation effect in different cell types. We believe that this adjustment will facilitate a better understanding of the perturbation effects on the TNF- α module without overburdening the figures with detailed information.

Clarification is required for why the basal tumor cells are compared to P60 EpSc cells in Figure 5.

We apologize for any confusion regarding Figure 5 and now revised and re-structured Figure 5.

Initially, we computed both, P60 EpSCs and total P60 skin sgRNA representation as comparisons. Since EpSCs represent the largest cluster, the comparative results were very similar. But the reviewer raises a valid question and we now present and compare the tumor sgRNA representation to total P60 skin sgRNA representation (dial-out data). This also ensures that both data set are sequencing reads with similar granularity and high sequencing depth (instead of number of cells with sgRNAs).

Reviewer Reports on the First Revision:

Referees' comments:

Referee #1 (Remarks to the Author):

The authors have thoughtfully addressed my comments during their revision of the paper.

The work is clear and extensive but I do agree with some of the comments made by the other reviewers which can be paraphrased as "an impressive use of technology but less overwhelming in terms of biological insights". This work will, however, be a primer for others to perform these experiments.

Referee #2 (Remarks to the Author):

In their revised manuscript, Renz and colleagues have addressed all my initial major concerns by providing additional data or clarifications in the text. Overall, this is an impressive manuscript, and I only have some minor comments the authors may want to consider.

1. Figure 2j, l: It is unclear why two different correlation analyses have been used and no indication of whether the correlations in j (and EDF4e) are significant is provided.
2. The authors state that they collected 168 tumors (both papillomas and SCCs). Since these are very different from each other, the authors need to clarify the ratio benign to malignant tumors they obtained, and state which tumors went into the analyses or whether only papillomas were analyzed in Figure 4c-i.

Referee #3 (Remarks to the Author):

Nature manuscript 2023-06-10719 (revised version)

Title: "In vivo single-cell CRISPR uncovers distinct TNF- α programs in clonal expansion and tumorigenesis" by Renz et. al.

In the revised manuscript and response to reviewers the authors have addressed the most critical issues raised during the first review. The revision is extensive and comprehensive including new molecular and functional approaches that reinforce the robustness of their novel approach, as well as the role of TNF- α as a potential regulator of clonal expansion and competition.

Two minor points:

- 1- The authors indicate that they show additional wholemount images of the P4 and P60 skin in the Supplementary Figure 1. However, there are no wholemount images in that figure. Please include these.
- 2- In the sentence "Therefore, for clonal expansion to occur in the interfollicular epidermis, mutated epidermal stem cells need to either proliferate faster than their neighboring cells or shift the balance towards self-renewal", the authors should consider the possibility to mention that clonal expansion may also be promoted by influencing the behavior of neighboring cells. This is a well-established mechanism of clonal competition, which will ultimately impact cell fate balance and competitive outcome.

Referee #4 (Remarks to the Author):

In this revision, Renz et al performed a number of new analyses and experiments to support their original claims. Specifically, in response to Concern #1, the authors provide two additional pieces of evidence to support their claim that TNF- α is a convergent initial state upon perturbation of driver mutations: (1) Correlate clonal expansion using cell numbers as a proxy with the activation of the TNF- α gene expression program, and (2) Perform enrichment of differentially expressed genes between perturbations with similar transcriptional response. Furthermore, the authors performed genetic and pharmacological perturbations of TNF- α to support the claim that TNF- α overexpression and its downstream targets lead to a hyper-proliferative phenotype in keratinocytes. The authors also provided an appropriately controlled experiment to support the claim that "TNF- α signaling directly confers clonal expansions", by directly comparing the clonal expansion of all the perturbations in contexts where TNF- α signaling is abrogated (Tnfr1 heterozygous mice) versus wildtype mice. Some minor outstanding comments:

1. To mechanistically support the observation that TNF- α is secreted from the immune cell compartment, the authors performed an elegant experimental set up by depleting either macrophages or T-cells. However the data presented in Fig 3g is not presented in a manner that enables evaluation. The experiment, analysis and visuals are not well described.
2. The authors write "Clonal expansion must initiate in the epidermal stem cell compartment, as expansions in differentiated layers are ultimately shed off without long-term maintenance in epithelia." The authors should provide support from the literature for this statement.

Author Rebuttals to First Revision:

We thank all four reviewers for their support and constructive comments!

Referee #1 (Remarks to the Author):

The authors have thoughtfully addressed my comments during their revision of the paper.

The work is clear and extensive but I do agree with some of the comments made by the other reviewers which can be paraphrased as "an impressive use of technology but less overwhelming in terms of biological insights". This work will, however, be a primer for others to perform these experiments.

Referee #2 (Remarks to the Author):

In their revised manuscript, Renz and colleagues have addressed all my initial major concerns by providing additional data or clarifications in the text. Overall, this is an impressive manuscript, and I only have some minor comments the authors may want to consider.

1. Figure 2j, l: It is unclear why two different correlation analyses have been used and no indication of whether the correlations in j (and EDF4e) are significant is provided.

Spearman correlation was used in Fig. 2j to indicate monotonic relationship between $\log_2(\text{cell number})$ and average gene expression, based on ranks. However, the weighted EpSC module expression showed a linear relationship explaining a substantial portion of the variance in clonal expansion. As such, Pearson correlation was used to indicate goodness of fit as the square of Pearson correlation equals the coefficient of determination (R^2) in linear regression.

We now added the p values for all 25 genes in EDF 4e.

2. The authors state that they collected 168 tumors (both papillomas and SCCs). Since these are very different from each other, the authors need to clarify the ratio benign to malignant tumors they obtained, and state which tumors went into the analyses or whether only papillomas were analyzed in Figure 4c-i.

We have observed that our chemical carcinogenesis strategy induces a mix of papillomas and squamous cell carcinomas (SCCs) 12 weeks post-treatment, as evidenced by Hematoxylin and Eosin (H&E) staining of the few tumors that were further analyzed. However, without pathology for each of the collected 168 tumors, we do not feel comfortable to make any claims regarding the ratio of benign to malignant tumors.

Referee #3 (Remarks to the Author):

Nature manuscript 2023-06-10719 (revised version)

Title: "In vivo single-cell CRISPR uncovers distinct TNF- α programs in clonal expansion and tumorigenesis" by Renz et. al.

In the revised manuscript and response to reviewers the authors have addressed the most critical issues raised during the first review. The revision is extensive and comprehensive including new molecular and functional approaches that reinforce the robustness of their novel approach, as well as the role of TNF- α as a potential regulator of clonal expansion and competition.

Two minor points:

1- The authors indicate that they show additional wholemount images of the P4 and P60 skin in the Supplementary Figure 1. However, there are no wholemount images in that figure. Please include these.

We checked again and for us, the submitted Supplementary Figure 1 shows the wholemount images. We are not sure why the wholemounts were not shown for this reviewer, but we made sure that the current version includes the wholemount images (one was moved to ED Figure 2, the other one is still in Supplementary Figure 1).

2- In the sentence "Therefore, for clonal expansion to occur in the interfollicular epidermis, mutated epidermal stem cells need to either proliferate faster than their neighboring cells or shift the balance towards self-renewal", the authors should consider the possibility to mention that clonal expansion may also be promoted by influencing the behavior of neighboring cells. This is a well-established mechanism of clonal competition, which will ultimately impact cell fate balance and competitive outcome.

The reviewer is correct. On page 5, we now have the statement: For clonal expansion to occur, EpSCs must either proliferate faster than their neighbors or shift the balance towards self-renewal. With “proliferate faster than their neighbors”, both scenarios are included. Either EpSCs can increase their proliferative rate or influence the proliferative rate of neighboring cells. Unfortunately, due to space restrictions (we reduced word counts from over 6000 to 3790 words), we cannot further discuss these two possibilities. However, we added 3 references to that sentence that highlight also the mechanisms influencing the behavior of neighboring cells.

Referee #4 (Remarks to the Author):

In this revision, Renz et al performed a number of new analyses and experiments to support their original claims. Specifically, in response to Concern #1, the authors provide two additional pieces of evidence to support their claim that TNF-alpha is a convergent initial state upon perturbation of driver mutations: (1) Correlate clonal expansion using cell numbers as a proxy with the activation of the TNF-alpha gene expression program, and (2) Perform enrichment of differentially expressed genes between perturbations with similar transcriptional response. Furthermore, the authors performed genetic and pharmacological perturbations of TNF-alpha to support the claim that TNF-alpha overexpression and its downstream targets lead to a hyper-proliferative phenotype in keratinocytes. The authors also provided an appropriately controlled experiment to support the claim that “TNF-a signaling directly confers clonal expansions“, by directly comparing the clonal expansion of all the perturbations in contexts where TNF-alpha signaling is abrogated (Tnfr1 heterozygous mice) versus wildtype mice. Some minor outstanding comments:

1. To mechanistically support the observation that TNF-alpha is secreted from the immune cell compartment, the authors performed an elegant experimental set up by depleting either macrophages or T-cells. However the data presented in Fig 3g is not presented in a manner that enables evaluation. The experiment, analysis and visuals are not well described.

We now reorganized Fig. 3g and added additional information and descriptions to enable a better evaluation.

2. The authors write “Clonal expansion must initiate in the epidermal stem cell compartment, as expansions in differentiated layers are ultimately shed off without long-term maintenance in epithelia.” The authors should provide support from the literature for this statement.

We now added the references.